# Towards Understanding Transformers in Learning Random Walks

**Wei Shi**
School of Computing and Data Science
The University of Hong Kong
shiwei1997@connect.hku.hk

**Yuan Cao**
School of Computing and Data Science
The University of Hong Kong
yuancao@hku.hk

## Abstract

Transformers have proven highly effective across various applications, especially in handling sequential data such as natural languages and time series. However, transformer models often lack clear interpretability, and the success of transformers has not been well understood in theory. In this paper, we study the capability and interpretability of transformers in learning a family of classic statistical models, namely random walks on circles. We theoretically demonstrate that, after training with gradient descent, a one-layer transformer model can achieve optimal accuracy in predicting random walks. Importantly, our analysis reveals that the trained model is interpretable: the trained softmax attention serves as a token selector, focusing on the direct parent state; subsequently, the value matrix executes a one-step probability transition to predict the location of the next state based on this parent state. We also show that certain edge cases not covered by our theory are indeed failure cases, demonstrating that our theoretical conditions are tight. By investigating these success and failure cases, it is revealed that gradient descent with small initialization may fail or struggle to converge to a good solution in certain simple tasks even beyond random walks. Experiments are conducted to support our theoretical findings.

## 1 Introduction

In recent years, transformers [31] have revolutionized many fields such as natural language processing [2, 23, 31], computer vision [8, 24], reinforcement learning [5, 13, 15], and have rapidly emerged as a key component in state-of-the-art deep learning models due to their ability to capture complex dependencies in data. While transformers exhibit remarkable practical performance, the underlying mechanisms of transformers are still not well understood due to their complex architecture.

In order to theoretically understand transformers, a number of recent works have investigated their capability in learning from sequential data that follows certain classic statistical models. Specifically, [4, 22] studied sequential data with an underlying causal graph, and theoretically showed how transformers can encode the causal structure with the self-attention mechanism for in-context learning. [6, 9] considered the task of in-context learning of Markov chains, and investigated how two-layer transformers can make predictions for Markov chains according to the context. [12] considered Markov chain forecasting tasks and reveals a connection between a context-conditioned Markov chain and the self-attention mechanism. [21] characterized the loss landscape of a one-layer transformer and demonstrated the existence of global minima and bad local minima in learning Markovian data with vocabularies of size two. Although these works provided valuable insights, there remain many open questions that require further exploration. Notably, the Markov chain data studied in [21] can be essentially understood as a random walk over a space of only two states. Therefore, given the results in [21], a natural question is when and how transformers can learn more general random walks over a larger state space.

39th Conference on Neural Information Processing Systems (NeurIPS 2025).

In this paper, we consider a classic statistical model for random sequences, namely random walks on circles, and study the capability of one-layer transformers to learn from such data and make predictions. We consider a general setting that allows a large number of nodes (possible locations of the walker) on the circle, which is an extension to the two-state setting studied in [21]. We also consider the general setting where a walker moves on the circle clockwisely with probability $p$, and counter-clockwisely with probability $1 - p$, and we build theories that cover the full range of $p \in [0, 1]$. For such a classic, yet relatively general class of random sequences, our goal is to theoretically study the performance of a one-layer transformer trained by gradient descent in predicting the next location, and reveal the interpretability of the obtained transformer model.

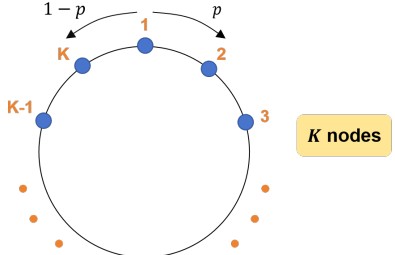

Figure 1: Illustration of random walks on circles with $K$ nodes and transition probability $p$.

The main contributions of this paper are as follows:

- We theoretically demonstrate that a one-layer transformer can be trained to optimally predict the next location of a random walk with $p \in (0, 1)$. Despite non-convexity, we prove that the trained model converges to the optimal prediction function with a rate $\mathcal{O}(T^{-1/2})$, where $T$ is the iteration number of gradient descent. Furthermore, we show that the trained transformer achieves optimal prediction accuracy $\max\{p, 1 - p\}$ after a constant number of iterations.

- Our analysis reveals that the trained transformer model is interpretable, as we precisely delineate the role of each component in the model. First, the trained softmax attention can select the "direct parent" token by assigning it a near-one score. Second, the trained value matrix serves as a one-step transition model that recovers the ground-truth probability transition matrix, which is applied to the "direct parent" token to make an optimal prediction.

- We also identify failure cases when $p = 0$ or $1$. In these cases, we show that starting from zero initialization, the training of the one-layer transformer model with any loss function and any learning rate will always fail, resulting in a transformer model whose performance is no better than a random guess. This negative result is complementary to our positive guarantees for learning random walks with $p \in (0, 1)$, and they together give a comprehensive characterization of the capability of transformers in learning random walks with $p \in [0, 1]$.

- We provide intuitive explanations that the failure cases with $p = 0$ or $1$ are optimization failures caused by zero initialization. Notably, similar optimization failures may also happen beyond the cases of random walks, as we can construct simple question answering tasks that also suffer from similar issues in optimization. We also empirically demonstrate that although training may still take longer, these failure cases can be resolved to a certain extent with random initialization.

## 2 Problem Setup

In this section, we present our problem formulations, including the random walk prediction task, the one-layer transformer model, and the training algorithm.

We study random walks on circles. Specifically, consider $K$ nodes (possible locations) that are arranged on a circle so that each node has two neighbors. Without loss of generality, we suppose that the nodes are assigned with node IDs $1, 2, \ldots, K$ in a clockwise manner. A "walk" on the circle refers to the process where a "walker" moves step-by-step among the nodes of the circle. We suppose that, starting from a random initial location, at each step, the walker moves either clockwise with probability $p$ or counterclockwise with probability $1 - p$, to a neighboring node of its current position, where $p \in (0, 1)$ is a fixed probability. In this way, a random walk of length $N$ generates a sequence of "states" $s_1, \ldots s_N$, where $s_i \in [K]$ denotes the location (node ID) of the walker at the $i$-th step. We aim to address the problem of predicting the walker's next location $s_N$ based on the historical locations $s_1, \ldots s_{N-1}$.

To better formulate this random walk prediction task, we map $s_1, \ldots, s_{N-1}$ to embeddings $\boldsymbol{x}_1, \ldots, \boldsymbol{x}_{N-1} \in \mathbb{R}^K$. Our goal is then to train a model to predict the target $y = s_N$ based on $\boldsymbol{x}_1, \ldots, \boldsymbol{x}_{N-1}$. In the following, we give a detailed definition and discuss some basic properties.

**Random walk on circles.** Suppose that there are $K$ nodes on a circle and the transition probability is $p$. $\boldsymbol{x}_1, \ldots, \boldsymbol{x}_{N-1}, y$ are generated as follows:

1. Draw $s_1 \sim \text{Unif}([K])$.
2. For $i = 2, \ldots, N$, sample either $s_i = \langle s_{i-1} + 1 \rangle_K$ with probability $p$ or $s_i = \langle s_{i-1} - 1 \rangle_K$ with probability $1 - p$.
3. Set $\boldsymbol{x}_i = \boldsymbol{e}_{s_i}$, $i \in [N-1]$, and $y = s_N$.

Here, $\langle s \rangle_K$ is defined as the integer satisfying $\langle s \rangle_K \in [K]$ and $\langle s \rangle_K \equiv s \pmod{K}$. It is clear that the sequence $\boldsymbol{x}_1, \ldots, \boldsymbol{x}_{N-1}, \boldsymbol{e}_y$ form a Markov chain, and $\mathbb{P}(y|\boldsymbol{x}_1, \ldots, \boldsymbol{x}_{N-1}) = \mathbb{P}(y|\boldsymbol{x}_{N-1})$. Moreover, the transition matrix of the Markov model is $\boldsymbol{\Pi} = (\pi_{i,j})_{K \times K}$, where $\pi_{i,j} = p \cdot \mathbb{1}\{i \equiv j - 1 (\text{mod } K)\} + (1 - p) \cdot \mathbb{1}\{i \equiv j + 1 (\text{mod } K)\}$. A visualization of $\boldsymbol{\Pi}$ is given in Figure 2. The Markov property indicates that the optimal predictor of $y$ is given by

$$f^{\text{OPT}}(\boldsymbol{x}_1, \ldots, \boldsymbol{x}_{N-1}) = \boldsymbol{\Pi}^\top \boldsymbol{x}_{N-1},$$

and the optimal prediction accuracy any predictor can achieve is $\text{OPT} = \max\{p, 1 - p\}$. Based on the formulation of $f^{\text{OPT}}(\cdot)$, it is clear that a simple autoregressive model $f(\boldsymbol{X}) = \boldsymbol{V} \boldsymbol{x}_{N-1}$ can already solve this random walk prediction task. However, the goal of this work is not to identify the optimal model for solving random walk tasks. Instead, we aim to understand and analyze the capability and interpretability of transformers when learning such classic statistical tasks from scratch. Therefore, in the following, we introduce a simple transformer model.

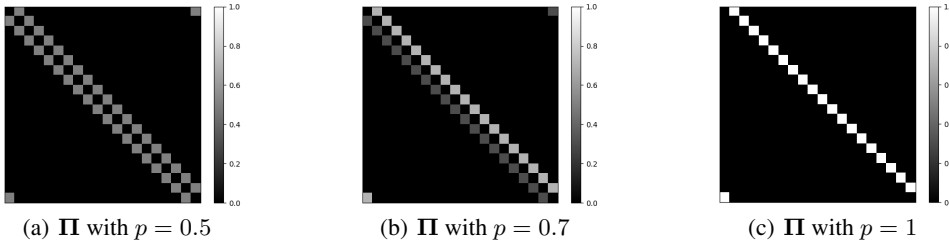

(a) $\boldsymbol{\Pi}$ with $p = 0.5$        (b) $\boldsymbol{\Pi}$ with $p = 0.7$        (c) $\boldsymbol{\Pi}$ with $p = 1$

Figure 2: Visualization of the transition matrices $\boldsymbol{\Pi}$ with $p = 0.5, 0.7, 1.0$ respectively.

We consider learning the random walk prediction task with a simple one-layer transformer model. By naturally treating the one-hot vectors $\boldsymbol{x}_1, \ldots, \boldsymbol{x}_{N-1}$ as token embeddings, the task to predict the next position $y$ is a problem of next token prediction. Therefore, we define the data matrix $\boldsymbol{X} = [\boldsymbol{x}_1, \boldsymbol{x}_2, \ldots, \boldsymbol{x}_{N-1}, \boldsymbol{0}] \in \mathbb{R}^{K \times N}$. We also employ a positional embedding matrix $\boldsymbol{P} = [\boldsymbol{p}_1, \boldsymbol{p}_2, \ldots, \boldsymbol{p}_N] \in \mathbb{R}^{M \times N}$, where $M$ is the embedding dimension with $M = \Omega(N^{3/2})$ and $\boldsymbol{p}_i \in \mathbb{R}^M$ is defined as

$$\boldsymbol{p}_i = \left[ \sin\left(\frac{i\pi}{M+1}\right), \sin\left(\frac{2i\pi}{M+1}\right), \ldots, \sin\left(\frac{Mi\pi}{M+1}\right) \right]^\top$$

for $i = 1, 2, \ldots, N$. The positional embeddings above are inspired by the fact that $\langle \boldsymbol{p}_i, \boldsymbol{p}_j \rangle = 0$ for all $i \neq j$, which significantly helps to simplify our theoretical analysis (see Lemma E.5 in the appendix). Additionally, orthogonal positional embeddings are commonly considered in existing theoretical studies [3, 22, 32, 33]. Then, we define the matrix $\widetilde{\boldsymbol{X}}$ by concatenating the input matrix $\boldsymbol{X}$ and the positional embedding matrix $\boldsymbol{P}$ as

$$\widetilde{\boldsymbol{X}} = \begin{bmatrix} \boldsymbol{X} \\ \boldsymbol{P} \end{bmatrix} = \begin{bmatrix} \boldsymbol{x}_1 & \boldsymbol{x}_2 & \cdots & \boldsymbol{x}_{N-1} & \boldsymbol{0} \\ \boldsymbol{p}_1 & \boldsymbol{p}_2 & \cdots & \boldsymbol{p}_{N-1} & \boldsymbol{p}_N \end{bmatrix} := [\widetilde{\boldsymbol{x}}_1, \widetilde{\boldsymbol{x}}_2, \ldots, \widetilde{\boldsymbol{x}}_N] \in \mathbb{R}^{(K+M) \times N}.$$

We consider a one-layer transformer model to make a prediction on a given input matrix $\boldsymbol{X}$. The transformer is defined as follows:

$$f_\theta(\boldsymbol{X}) = \boldsymbol{V} \boldsymbol{X} \mathcal{S}(\widetilde{\boldsymbol{X}}^\top \boldsymbol{W} \widetilde{\boldsymbol{x}}_N), \tag{2.1}$$

where $\boldsymbol{V} \in \mathbb{R}^{K \times K}$, $\boldsymbol{W} \in \mathbb{R}^{(K+M) \times (K+M)}$ are the trainable parameter matrices, $\mathcal{S} : \mathbb{R}^N \to \mathbb{R}^N$ is the softmax function defined by $[\mathcal{S}(\boldsymbol{z})]_i = \frac{\exp(z_i)}{\sum_{j=1}^N \exp(z_j)}$, and $\theta = (\boldsymbol{V}, \boldsymbol{W})$ denotes the collection of all the trainable parameters. In this definition, we consider a reparameterization where we

use a single matrix $\boldsymbol{W}$ to denote the product of the key and query parameter matrices in self-attention. Such kind of reparameterizations are widely considered in theoretical studies of transformer models [11, 12, 14, 16, 22, 29, 32, 34]. In addition, we omit the softmax scaling factor, which is mainly for simplicity. Such omission has also been considered in most of the theoretical studies [7, 17, 18, 22, 25, 26].

Note that by (2.1), given any input matrix $\boldsymbol{X}$, the transformer model outputs a $K$-dimensional vector. This follows the standard practice of $K$-class classification: for $i \in [K]$, $[f_\theta(\boldsymbol{X})]_i$ can be treated as a predicted "score" of the $i$-th class. More specifically, we define the prediction rule as follows.

**Definition 2.1.** For any predictor $f(\boldsymbol{X}) : \mathbb{R}^{K \times N} \to \mathbb{R}^K$, the predicted label is given as

$$\mathrm{Pred}(f(\boldsymbol{X})) := \min \left\{ j \in [K] : [f(\boldsymbol{X})]_j = \max_{i \in [K]}\{[f(\boldsymbol{X})]_i\} \right\}.$$

The definition above matches the common practice to predict the label that corresponds to the entry in $f(\boldsymbol{X})$ with the maximum function value. It also gives a naive way to handle ties – when $f(\boldsymbol{X})$ contains multiple dimensions with the same (and maximum) function value, we always predict the dimension corresponding to the smallest label. We remark that this definition to handle ties is just to exclude ambiguity, and the detailed rule to handle ties is not essential. Our result works for all reasonable methods to handle ties.

We train the transformer model defined in (2.1) by gradient descent, minimizing the loss function

$$L(\theta) = \mathbb{E}_{(\boldsymbol{X},y)} \left[ \ell \left( e_y^\top f_\theta(\boldsymbol{X}) \right) \right], \tag{2.2}$$

where $\ell(\cdot)$ is a loss function. In terms of the specific choice of $\ell(\cdot)$, our analysis will cover learning random walks on a circle by minimizing the log-loss $\ell(z) = -\log(z + \epsilon)$, which has been considered in a series of recent works [12, 16, 21, 28].

We consider gradient descent with zero initialization $\boldsymbol{V}^{(0)} = \boldsymbol{0}^{K \times K}$, $\boldsymbol{W}^{(0)} = \boldsymbol{0}^{(K+M) \times (K+M)}$ to train the model. The update rule for the parameter matrices $\boldsymbol{W}, \boldsymbol{V}$ are as follows:

$$\boldsymbol{W}^{(t+1)} = \boldsymbol{W}^{(t)} - \eta \nabla_{\boldsymbol{W}} L(\theta^{(t)}), \quad \boldsymbol{V}^{(t+1)} = \boldsymbol{V}^{(t)} - \eta \nabla_{\boldsymbol{V}} L(\theta^{(t)}), \tag{2.3}$$

where $\eta > 0$ is the learning rate and $t \geq 0$ is the iteration number. Note that the log-loss $\ell(z) = -\log(z + \epsilon)$ is well-defined and does not blow up at zero initialization due to the stability constant $\epsilon > 0$. Gradient descent with appropriate learning rate further ensures that it does not blow up during training.

# 3 Main Results

In this section, we present our main theoretical results on learning random walks with a self-attention layer. In our result, we can choose any $T^* = \mathrm{poly}(\eta, \epsilon^{-1}, K, N, M)$ as the maximum admissible number of iterations, and only consider the training period $0 \leq t \leq T^*$. This technical assumption regarding a polynomially large maximum admissible number of iterations prevents training from becoming exponentially long and is a mild assumption since exponentially long training is impractical.

Our main results are given in the following theorem.

**Theorem 3.1.** Suppose that $0 < p < 1$, $K > 4$, and $N \geq C_p \cdot \mathrm{poly}(K)$ for some constant $C_p$ that only depends on $p$. Further suppose that the transformer is trained by gradient descent (2.3) to minimize the loss (2.2) with $\ell(z) = -\log(z + \epsilon)$, and $\eta, \epsilon = \Theta(1)$. Then there exists $T_0 = \Theta(1)$, such that for all $T_0 \leq T \leq T^*$, it holds that:

1. The trained transformer achieves optimal prediction accuracy:

$$\mathbb{P}\big[\mathrm{Pred}(f_{\theta^{(T)}}(\boldsymbol{X})) = y\big] = \mathrm{OPT} = \max\{p, 1 - p\}.$$

2. The transformer converges to the optimal predictor:

$$\left\| \frac{f_{\theta^{(T)}}(\boldsymbol{X})}{\|f_{\theta^{(T)}}(\boldsymbol{X})\|_2} - f^{\mathrm{OPT}}(\boldsymbol{X}) \right\|_2 = \mathcal{O}\left( \frac{1}{\sqrt{T}} \right).$$

3. The value matrix converges to the true transition matrix in direction:

$$\left\| \frac{\boldsymbol{V}^{(T)}}{\|\boldsymbol{V}^{(T)}\|_F} - \frac{\boldsymbol{\Pi}^\top}{\|\boldsymbol{\Pi}^\top\|_F} \right\|_F = \mathcal{O}\left(\frac{1}{\sqrt{T}}\right).$$

4. Softmax attention selects the "direct parent" of $y$:

$$\left[\mathcal{S}(\widetilde{\boldsymbol{X}}^\top \boldsymbol{W}^{(T)}\widetilde{\boldsymbol{x}}_N)\right]_{N-1} \geq 1 - \exp(-\Omega(N)),$$
$$\left[\mathcal{S}(\widetilde{\boldsymbol{X}}^\top \boldsymbol{W}^{(T)}\widetilde{\boldsymbol{x}}_N)\right]_j \leq \exp(-\Omega(N)) \text{ for all } j \neq N-1.$$

The first result in Theorem 3.1 states that the transformer trained by gradient descent for a constant number of iterations can achieve a prediction accuracy $\max\{p, 1-p\}$, which matches the optimal accuracy OPT. The second result in Theorem 3.1 further gives a more detailed characterization of the trained transformer, and demonstrates that the normalized model converges to the optimal prediction model $f^{\mathrm{OPT}}(\boldsymbol{X}) = \boldsymbol{\Pi}^\top \boldsymbol{x}_{N-1}$.

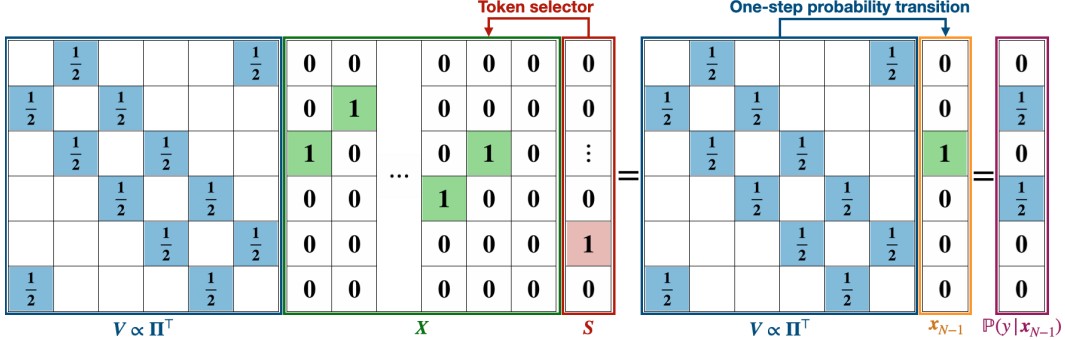

Figure 3: Illustration of how the trained one-layer transformer performs optimal prediction in an example with $p = 1/2$, $K = 6$. Here we denote $\boldsymbol{S} = \mathcal{S}(\widetilde{\boldsymbol{X}}^\top \boldsymbol{W}^{(T)}\widetilde{\boldsymbol{x}}_N)$.

The third and fourth results in Theorem 3.1 further back up the first two results by a precise characterization on how the self-attention mechanism works in predicting random walks. Specifically, the third result demonstrates that in direction, **the value matrix $\boldsymbol{V}^{(T)}$ serves as a one-step transition model** by recovering the ground-truth one-step transition matrix, and the last result indicates that **softmax attention performs optimal token selection** by assigning a near-one score to the $(N-1)$-th token (the direct parent of $y$). These two results show that, when learning to predict random walks, the one-layer transformer obtained through gradient descent is **interpretable**: the trained one-layer transformer model makes predictions by (i) selecting the correct parent token $\boldsymbol{x}_{N-1}$ of $y$ by assigning a softmax weighting close to 1 to it, and (ii) predicting $y$ by applying a one-step transition model to $\boldsymbol{x}_{N-1}$ through the linear mapping defined by $\boldsymbol{V}$. An illustration is given in Figure 3.

A recent related work [21] studied how one-layer transformers can learn Markov chains with a state space of size two, and analyzed the loss landscape by identifying global minima and bad local minima. We remark that we consider a different parameterization of one-layer transformers, which makes our results not rigorously comparable. However, by studying random walks over state spaces of arbitrary sizes and establishing theoretical guarantees directly on transformers trained by gradient descent, our work explores a setting that is not covered in [21].

Notably, Theorem 3.1 is based on the assumption that the transition probability satisfies $0 < p < 1$, excluding the edge cases when $p = 0, 1$. In Section 4, we will show a negative result showing that when $p = 0, 1$, gradient descent with zero initialization fails to achieve good prediction accuracy. Therefore, the assumption $0 < p < 1$ in Theorem 3.1 is inevitable.

Here we informally explain how Theorem 3.1 is proved. The idea of the proof can be shown by investigating the first several gradient descent steps:

**Step 1.** After the first gradient descent step, due to the zero initialization and the Toeplitz property of the transition matrix $\boldsymbol{\Pi}$, it can be shown that $\boldsymbol{V}^{(1)}$ is also a Toeplitz matrix whose largest entries appear exactly on the locations of the largest entries of $\boldsymbol{\Pi}^\top$. $\boldsymbol{W}^{(1)}$ is still a zero matrix due to the fact that $\boldsymbol{V}^{(0)} = \boldsymbol{0}$.

**Step 2.** With the same analysis, we can also show that $\boldsymbol{V}^{(2)}$ is a Toeplitz matrix whose largest entries appear exactly on the locations of the largest entries of $\boldsymbol{\Pi}^{\top}$. Moreover, the locations of the largest entries in $\boldsymbol{V}^{(1)}$ encourage $\boldsymbol{W}^{(2)}$ to be updated so that higher softmax weightings are put upon the "direct parent" token $\boldsymbol{x}_{N-1}$ (see Lemma C.5 in the appendix).

**Step 3.** The higher weighting on $\boldsymbol{x}_{N-1}$ given by $\boldsymbol{W}^{(2)}$ further encourages $\boldsymbol{V}^{(3)}$ to be updated towards $\boldsymbol{\Pi}^{\top}$ in direction. And $\boldsymbol{V}^{(2)}$ obtained in Step 2 continues to encourage $\boldsymbol{W}^{(3)}$ to place a even higher weighting on $\boldsymbol{x}_{N-1}$ (see Lemma C.7 in the appendix).

From the three gradient descent steps discussed above, it is clear that $\boldsymbol{V}^{(t)}$ will converge to the direction of $\boldsymbol{\Pi}^{\top}$, and $\boldsymbol{W}^{(t)}$ will consistently place a high weighting on the direct parent token $\boldsymbol{x}_{N-1}$. This is our key intuition for proving Theorem 3.1, and in our formal proof (given in the appendix), we use an induction to characterize the whole training procedure.

# 4 "Deterministic Walks" with $p = 0, 1$

In Section 3, we demonstrate that a one-layer transformer can be trained to optimally predict random walks under the assumption $0 < p < 1$. In this section, we justify the necessity of this assumption and analyze the edge cases $p = 0, 1$, which lead to a failure for learning random walks. Random walks with $p = 0, 1$ are essentially "deterministic walks", and for them we have the following theorem.

**Theorem 4.1.** Suppose that $N = rK + 1$ with $r \geq 1$, and $p = 0$ or $1$. Further suppose that the transformer is trained by gradient descent (2.3) to minimize the loss (2.2). Then for any loss function $\ell(\cdot)$, any learning rate $\eta > 0$, and any $T \geq 0$, it holds that

$$\mathbb{P}\big(\mathrm{Pred}(f_{\theta^{(T)}}(\boldsymbol{X})) = y\big) = \frac{1}{K}.$$

Moreover, with probability 1, for all $T \geq 0$, it holds that

$$\boldsymbol{V}^{(T)} \propto \mathbf{1}_{K \times K}, \quad \Big[\mathcal{S}(\widetilde{\boldsymbol{X}}^{\top} \boldsymbol{W}^{(T)} \widetilde{\boldsymbol{x}}_N)\Big]_1 = \cdots = \Big[\mathcal{S}(\widetilde{\boldsymbol{X}}^{\top} \boldsymbol{W}^{(T)} \widetilde{\boldsymbol{x}}_N)\Big]_{N-1}.$$

Theorem 4.1 shows that the prediction accuracy of the trained transformer on deterministic walks is $1/K$, equal to the accuracy of a random guess. Moreover, the characterizations of the softmax scores and the value matrix $\boldsymbol{V}^{(T)}$ further demonstrate that the transformer takes average over all tokens, and then gives the same prediction scores for all possible values of $y$. Notably, these results hold for any choice of the loss function and any learning rate, indicating that this failure case of the transformer cannot be resolved by simply adjusting these training setups.

Theorems 3.1 and 4.1 together provide a comprehensive characterization of the transformer's performance when learning random walks with transition probabilities $p \in [0, 1]$. Based on Theorem 4.1, it is clear that the assumption in Theorem 3.1 that $0 < p < 1$ is necessary and cannot be further relaxed. Theorems 3.1 and 4.1 also point out an interesting fact, that compared to random walks with $0 < p < 1$, the seemingly easier task of predicting deterministic walks with $p = 0, 1$ may be more challenging for a transformer to learn. Here, we would like to remark that the failure of learning deterministic walks is an optimization failure, and is highly due to zero initialization. To see the reason, we can consider the two initial gradient descent steps:

**Step 1.** Since the initial softmax weightings on all tokens are the same, $\boldsymbol{V}^{(1)}$ is essentially trained based on the averaged token $\overline{\boldsymbol{x}} = \frac{1}{N-1} \sum_{i=1}^{N-1} \boldsymbol{x}_i$. Importantly, we can see that

$$\overline{\boldsymbol{x}} \text{ is a constant vector that does not depend on } y.$$

This means that $\overline{\boldsymbol{x}}$ does not provide any helpful information in predicting $y$, and is therefore "uninformative". As a result, it can be shown that all entries in $\boldsymbol{V}^{(1)}$ are equal (see Lemma D.1 in the appendix). Additionally, $\boldsymbol{W}^{(1)}$ is still a zero matrix since $\boldsymbol{V}^{(0)} = \mathbf{0}$.

**Step 2.** With the same analysis as Step 1, we can show that all entries in $\boldsymbol{V}^{(2)}$ are equal. Moreover, due to the fact that $\boldsymbol{V}^{(1)}$ is proportional to the all-one matrix, it can be further shown that $\boldsymbol{W}^{(2)}$ is updated so that the softmax weightings on all tokens $\boldsymbol{x}_1, \ldots, \boldsymbol{x}_{N-1}$ remain equal.

In our formal proof, we inductively show that throughout training, the value matrix $\boldsymbol{V}^{(t)}$ is always proportional to the all-one matrix, and the softmax weights on all tokens are always the same.

From the discussion above, we can observe that transformers struggle to learn deterministic walks due to the unbreakable symmetry in the training dynamics, arising from the "uninformative" token average $\overline{\boldsymbol{x}}$ given by the zero initialization. We remark that if random initialization is used, the symmetry will be broken, and transformers may succeed in learning the optimal predictor. However, if the random initialization is too small in scale, we can still expect that the optimization for learning deterministic walks to be more challenging compared to that for random walks. While our theoretical analysis does not cover random initialization, we present empirical studies in Sections 5 and 6 to verify this claim. We believe that theoretically analyzing the impact of random initializations can be an important future work direction.

# 5   Experiment

In this section, we present experimental results to support our theoretical analysis. We consider two cases: the first one is the zero initialization case that aligns with the setting used in our theoretical analysis, and the second one is the random initialization case, which helps verify our discussion about the optimization failure caused by zero initialization. In all experiments introduced in this section, we set the number of nodes $K = 6$ and the length of each sequence $N = 97$. We utilize the transformer model introduced in Section 2 and utilize the gradient method to train the model. The prediction accuracy is calculated based on 1000 test data.

**Zero initialization.** In this case, we set the length of the positional embedding $M = 1000$, the initialization $\boldsymbol{V}^{(0)} = \boldsymbol{0}_{K \times K}$, $\boldsymbol{W}^{(0)} = \boldsymbol{0}_{(K+M) \times (K+M)}$, and the learning rate $\eta = 1$. The constant $\epsilon$ in the log-loss is set as $\epsilon = 0.1$. For both tasks, we generate 1000 sequences to train the model.

Figure 4 and Figure 5 illustrate the results of the experiment for $p = 0.5$ and $p = 1$ respectively: Figure 4(a) and Figure 5(a) present the prediction accuracy; Figure 4(b) and Figure 5(b) visualize the value matrix $\boldsymbol{V}^{(T)}$ after 50 iterations; Figure 4(c) and Figure 5(c) display the attention scores attached to each token after 50 iterations. To clearly observe the results, we also provide Figure 4(d) that represents the part of Figure 4(c).

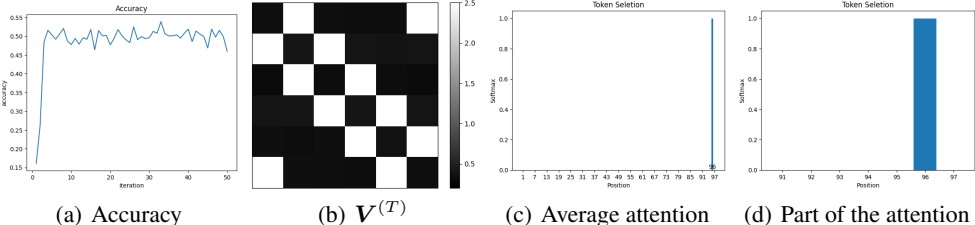

(a) Accuracy       (b) $\boldsymbol{V}^{(T)}$       (c) Average attention       (d) Part of the attention

Figure 4: The results of the experiments for $p = 0.5$ with zero initialization: (a) is the test accuracy; (b) is the visualization of $\boldsymbol{V}$; (c) and (d) present the average attention of the test data with $x$-axis representing the position of the token and $y$-axis representing the attention score.

We can observe that these experimental results for $p = 0.5$ provide strong support for Theorem 3.1. Figure 4(a) shows that the prediction accuracy is close to the optimal accuracy (50%) within constant iterations. Figure 4(b) indicates that $\boldsymbol{V}^{(T)}$ can recover the transition matrix $\boldsymbol{\Pi}^{\top}$ as shown in Figure 2(a). Figure 4(c) presents that the $(N - 1)$-th attention score is the highest and close to 1, indicating that the self-attention layer is able to select the true parent token. All of these experimental results demonstrate the performance of transformers in learning random walks.

In addition, we can find that the experimental outcomes for $p = 1$ match the theoretical results stated in Theorem 4.1. We obtain an accuracy close to 0.167 from Figure 5(a), which suggests that the prediction accuracy for learning deterministic walks is approximately equal to $1/K$, far away from the optimal accuracy (100%) and no better than a random guess. Figure 5(b) indicates that $\boldsymbol{V}^{(T)}$ is approximately proportional to $\boldsymbol{1}_{K \times K}$. Figure 5(c) shows that the attention scores attached to all tokens are identical, which proves that the self-attention layer cannot select any of the tokens when learning deterministic walks. These experimental results demonstrate that the self-attention mechanism struggles in learning deterministic walks with $p = 0, 1$.

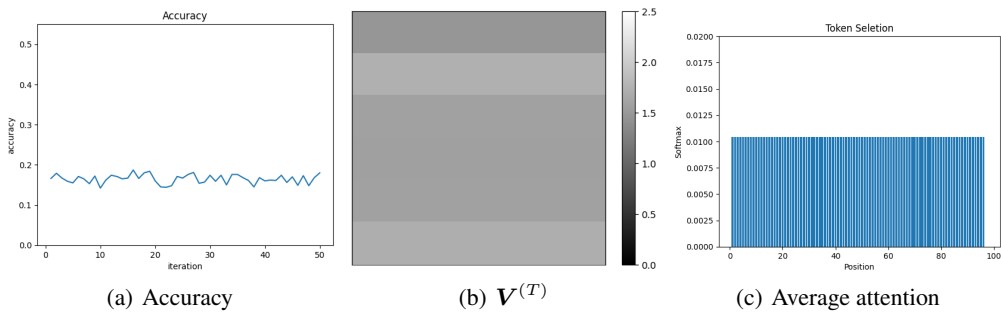

(a) Accuracy           (b) $\boldsymbol{V}^{(T)}$           (c) Average attention

Figure 5: Experiments for $p = 1$ with zero initialization. (a) gives the prediction accuracy along training. (b) is the visualization of $\boldsymbol{V}$. (c) is the average attention of the test data with $x$-axis representing the position of the token and $y$-axis representing the attention score.

**Random initialization.** In this case, we set the length of the positional embedding $M = 1000$, the initialization $\boldsymbol{V}_{ij}^{(0)}, \boldsymbol{W}_{ij}^{(0)} \sim N(0, \sigma^2)$ with $\sigma = 0.01$, and the learning rate $\eta = 0.01$. The constant $\epsilon$ in the log-loss is set as $\epsilon = 0.1$. For both tasks, we generate 1000 sequences to train the model. To ensure numerical stability, we normalize $\widetilde{\boldsymbol{x}}_i$'s to unit length in the softmax attention.

Figure 6 illustrates the results of the experiment for $p = 0.5$ and $p = 1$. Figure 6(a) and Figure 6(c) show the prediction accuracy within 1000 iterations, respectively. In Figure 6(b) and Figure 6(d), we first normalize the output of the trained transformer model to get a $K$-dimensional vector, which can be regarded as the prediction distribution of $K$ locations. The KL-divergence between this prediction distribution and the true distribution of $y|\boldsymbol{x}_{N-1}$ is illustrated in Figure 6(b) and Figure 6(d).

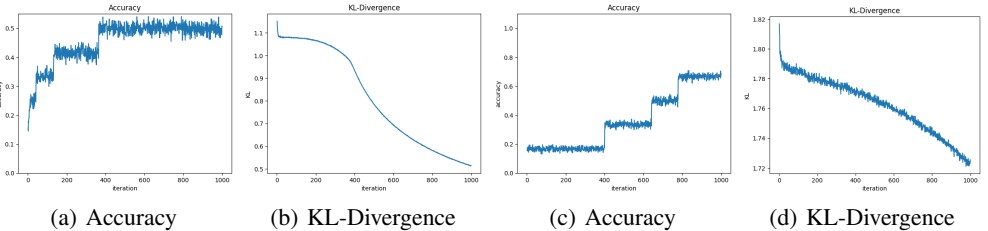

(a) Accuracy      (b) KL-Divergence      (c) Accuracy      (d) KL-Divergence

Figure 6: The results of the synthetic experiment with random initialization: (a) and (b) correspond to the experiment for $p = 0.5$; (c) and (d) correspond to the experiment for $p = 1$. (a) and (c) present the prediction accuracy. In (b) and (d), we first normalize the output of the trained transformer model to get a $K$-dimensional vector, representing the prediction distribution of $K$ locations. Then, we display the KL-divergence between this prediction distribution and the true distribution of $y|\boldsymbol{x}_{N-1}$.

Figure 6(a) clearly shows that in the experiment for $p = 0.5$, the accuracy is close to the optimal accuracy (50%) after around 400 iterations. However, as shown in Figure 6(c), for $p = 1$, the prediction accuracy cannot reach the optimal accuracy (100%) within 1000 iterations. Based on the plots of KL-divergence, we can also see that the transformer learns the true prediction distribution of random walks much faster than learning that of deterministic walks. Note that these results are for training with random initialization, and hence the results do not perfectly match our theory for zero initialization in Section 3. However, the experiment results still show that the optimization task is more challenging, even with small random initialization.

## 6 Beyond Random Walks

In Section 4, we intuitively explain that one-layer transformers may suffer from optimization issues when learning deterministic walks with $p = 0, 1$ due to the fact that zero initialization produces a token average which is "uninformative". In this section, we briefly discuss other tasks beyond random walks where transformers with zero/small initialization may face similar optimization challenges.

We construct two question answering tasks. The detailed descriptions are given as follows.

**Task 1.** We consider a simple question answering task. Possible input questions are of the form:

*Based on the list 'apple, orange, apple, apple, orange', which type of fruit appears most frequently?*

Here, the list stated in the question can be any combination of 'apple' and 'orange' with a fixed length of 5. Therefore, there are a total of 32 possible questions the model may see, and each of these questions occurs with probability $1/32$. Ignoring punctuation marks, each input sample is assumed to be 16 words involving the list and other words in the inquiry sentence. The correct response (the 'label' for classification) is the fruit that appears most frequently in the list. For example, for the question *"Based on the list 'apple, orange, apple, apple, orange', which type of fruit appears most frequently?"*, the correct response is *apple*.

**Task 2.** We again consider a simple question answering task with only two possible questions:

> *Based on the sentence 'I prefer an apple to an orange', which type of fruit do I prefer?*
> *Based on the sentence 'I prefer an orange to an apple', which type of fruit do I prefer?*

Here, each of the two questions above occurs with probability $1/2$. Similar to Task 1, we ignore the punctuation marks, and the input is the 18 words in the sentence. The correct response (the "label" for classification) is *apple* for the first question above, and *orange* for the second question above.

Combining all the words appearing in two tasks, we attain a vocabulary with a length of 19: {'apple', 'orange', 'Based', 'on', 'the', 'which', 'type', 'of', 'fruit', 'list', 'appears', 'most', 'frequently', 'sentence', 'I', 'prefer', 'an', 'to', 'do'}. We embed this sequence as a matrix $\boldsymbol{E} = [\boldsymbol{e}_1, \boldsymbol{e}_2, ..., \boldsymbol{e}_{19}] \in \mathbb{R}^{19 \times 19}$. Then, we know that the length of the vocabulary $K$ and the length of each input sequence $N$ are set as $(K, N) = (19, 17), (19, 19)$ for Task 1 and Task 2 respectively.

In the experiments, we consider a similar transformer model as we introduced in our theoretical analysis. To train the model, we consider Gaussian random initialization $\boldsymbol{V}_{ij}^{(0)}, \boldsymbol{W}_{ij}^{(0)} \sim N(0, \sigma^2)$ with $\sigma = 0.01$, and we use gradient descent with learning rate $\eta = 0.1$ to train the model. The constant $\epsilon$ in the log-loss is set as $\epsilon = 0.1$. Both the training and test datasets contain 1000 samples.

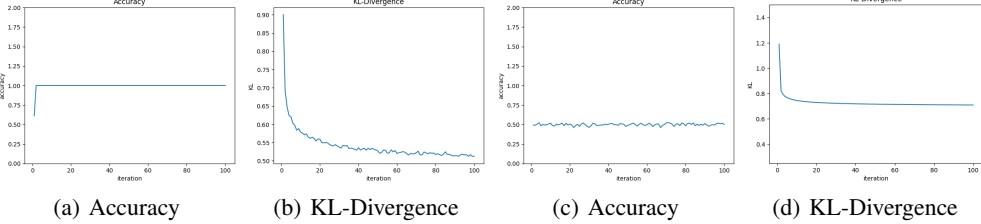

| (a) Accuracy | (b) KL-Divergence | (c) Accuracy | (d) KL-Divergence |

Figure 7: The results of the experiment for Task 1 and Task 2: (a) and (b) correspond to the experiment for Task 1; (c) and (d) correspond to the experiment for Task 2.

Figure 7 shows the experiment results for Task 1 and Task 2. Figure 7(a) and Figure 7(c) present the test accuracy. In Figure 7(b) and Figure 7(d), we first normalize the output of the trained transformer model to get a $K$-dimensional vector, representing the prediction distribution of $K$ words. Then, we report the KL-divergence between this prediction distribution and the true distribution of $y|\boldsymbol{x}_1, \boldsymbol{x}_2, ..., \boldsymbol{x}_{N-1}$ in Figure 7(b) and Figure 7(d). The experiment results show a clear difference between the performances of the transformer model in the two tasks. In Task 1, the trained transformer model can successfully approach the optimal accuracy (100%) within 100 iterations. However, in Task 2, the test accuracy remains around 50% within 100 iterations.

The results can be explained following the discussion in Section 4. Specifically, in Task 1, the average of the word embeddings $\overline{\boldsymbol{x}}$ in a question can help the model to find the correct response, while in Task 2, the two questions give the *same* average of word embeddings $\overline{\boldsymbol{x}}$, and therefore, it causes inefficient optimization.

The results for these two tasks demonstrate that our theories and explanations for random walks can also guide the construction of various other learning tasks and predict the performance of a transformer model in these tasks. This confirms the validity of our theories and explanations and highlights the insights provided by our study.

# 7 Conclusion

This paper investigates the capability of a one-layer transformer model to learn random walks on circles. We demonstrate that transformers successfully learn such walks for the transition probability $0 < p < 1$, achieving the optimal prediction accuracy. We also show that the trained model is

interpretable: the softmax attention mechanism effectively selects the correct "parent" token, while the value matrix recovers a one-step transition model that applies to the selected "parent" token for optimal prediction. In addition, we identify that the edge cases ($p = 0, 1$) are failure cases, thereby proving the necessity of the assumption $0 < p < 1$. Motivated by the analysis of success and failure in learning random walks, we design simple question answering tasks that exhibit similar optimization challenges, showing the broader applicability of our analysis to other tasks beyond random walks. We also provide experimental results to validate our theoretical findings.

In future works, it is important to theoretically study the impact of random initialization in learning random walks. Moreover, an interesting future work direction is to extend the results and study the performance of deeper transformer architectures, which may require more advanced theoretical tools. Moreover, extending the finding to more complicated learning tasks, such as random sequences generated by general Markov chains or Bayesian networks, is also an important future work direction.

## Acknowledgments

We thank the anonymous reviewers and area chairs for their valuable and constructive comments. Yuan Cao is partially supported by NSFC 12301657 and Hong Kong ECS award 27308624.

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

## A  Additional Related Work

In this section, we give an overview of some additional related works.

**Token selection.** Our work reveals that a one-layer transformer can learn to perform the optimal token selection by focusing on the direct parent in a random walk. A line of recent works has studied the token selection of the self-attention mechanism from different perspectives. [26, 27] propose an equivalence between the optimization dynamics of one self-attention layer and an SVM problem and prove the global convergence under certain assumptions. [16] shows that when training a self-attention layer, the priority in token selection is determined by a directed graph extracted from the training data. [32] demonstrates that transformer models can learn the sparse token selection task effectively while fully connected networks fail in the worst case. [18] shows that a self-attention layer can be trained to perform proper token selection so that the model acts as a one-nearest neighbor classifier in context.

**Next-token prediction.** [28] explores the implicit bias of next-token prediction employing a related SVM formulation. [19] demonstrates that transformers fail to solve the Partially Observable Markov Decision Processes problem (POMDP) even with sufficient data. [10] observes a phenomenon of next-token prediction in LLM that each layer contributes equally to enhancing the prediction accuracy. [29] studies the SGD training dynamics of a transformer with one self-attention layer and one decoder layer for next-token prediction, restricted to some specific assumptions like no positional encoding, long input sequences, and the fact that the decoder layer learns faster than the self-attention layer.

**Training dynamics of transformers.** [20, 34] investigate the training dynamics of in-context learning in transformers with a single self-attention layer trained through gradient flow on linear regression tasks. [11] solves in-context linear regression with the orthogonal input data by gradient descent on a single softmax attention layer. [14] demonstrates that the position-position block of a single attention layer in a vision transformer can encode spatial structure by dealing with a binary classification task. [30] delves into the training process of transformers with multi-layers by analyzing the dynamics of the MLP layers. [3] analyzes a synthetic in-context learning task and emphasizes the significance of weight matrices as associative memories. [1] shows incremental learning dynamics in transformers with diagonal attention matrices.

## B  Gradient Calculation

Recall that the population loss is

$$L(\theta) = \mathbb{E}[\ell(\theta)] = \mathbb{E}[-\log(e_y^\top f_\theta(\boldsymbol{X}) + \epsilon)] = \mathbb{E}[-\log(e_y^\top \boldsymbol{V}\boldsymbol{X}\mathcal{S}(\widetilde{\boldsymbol{X}}^\top \boldsymbol{W}\widetilde{\boldsymbol{x}}_N) + \epsilon)].$$

The following lemma calculates the gradients of the population function.

**Lemma B.1.** The gradients regarding $\boldsymbol{V}$ and $\boldsymbol{W}$ are

$$\nabla_{\boldsymbol{V}}\ell(\theta) = \ell' \cdot e_y \sum_{i=1}^{N-1} \mathcal{S}_i \boldsymbol{x}_i^\top = -\frac{1}{e_y^\top \boldsymbol{V}\boldsymbol{X}\mathcal{S} + \epsilon} \cdot e_y \sum_{i=1}^{N-1} \mathcal{S}_i \boldsymbol{x}_i^\top,$$

$$\nabla_{\boldsymbol{W}}\ell(\theta) = \ell' \cdot \begin{bmatrix} \boldsymbol{0} & \left(\sum_{i=1}^{N-1}\mathcal{S}_i\boldsymbol{x}_i\boldsymbol{x}_i^\top - \sum_{i=1}^{N-1}\mathcal{S}_i\boldsymbol{x}_i \cdot \sum_{i=1}^{N-1}\mathcal{S}_i\boldsymbol{x}_i^\top\right)\boldsymbol{V}^\top e_y \boldsymbol{p}_N^\top \\ \boldsymbol{0} & \left(\sum_{i=1}^{N-1}\mathcal{S}_i\boldsymbol{p}_i\boldsymbol{x}_i^\top - \sum_{i=1}^{N}\mathcal{S}_i\boldsymbol{p}_i \cdot \sum_{i=1}^{N-1}\mathcal{S}_i\boldsymbol{x}_i^\top\right)\boldsymbol{V}^\top e_y \boldsymbol{p}_N^\top \end{bmatrix}$$

$$= -\frac{1}{e_y^\top \boldsymbol{V}\boldsymbol{X}\mathcal{S} + \epsilon} \cdot \begin{bmatrix} \boldsymbol{0} & \left(\sum_{i=1}^{N-1}\mathcal{S}_i\boldsymbol{x}_i\boldsymbol{x}_i^\top - \sum_{i=1}^{N-1}\mathcal{S}_i\boldsymbol{x}_i \cdot \sum_{i=1}^{N-1}\mathcal{S}_i\boldsymbol{x}_i^\top\right)\boldsymbol{V}^\top e_y \boldsymbol{p}_N^\top \\ \boldsymbol{0} & \left(\sum_{i=1}^{N-1}\mathcal{S}_i\boldsymbol{p}_i\boldsymbol{x}_i^\top - \sum_{i=1}^{N}\mathcal{S}_i\boldsymbol{p}_i \cdot \sum_{i=1}^{N-1}\mathcal{S}_i\boldsymbol{x}_i^\top\right)\boldsymbol{V}^\top e_y \boldsymbol{p}_N^\top \end{bmatrix},$$

where $\mathcal{S} = \mathcal{S}(\widetilde{\boldsymbol{X}}^\top \boldsymbol{W}\widetilde{\boldsymbol{x}}_N)$, and $\mathcal{S}_i$ is the $i$-th element of $\mathcal{S}$.

***Proof of Lemma B.1.*** For $\boldsymbol{V}$, we have

$$\nabla_{\boldsymbol{V}}\ell(\theta) = -\frac{1}{e_y^\top f_\theta(\boldsymbol{X}) + \epsilon} \cdot \frac{\partial e_y^\top \boldsymbol{V}\boldsymbol{X}\mathcal{S}}{\partial \boldsymbol{V}}$$

$$= -\frac{1}{e_y^\top \boldsymbol{V}\boldsymbol{X}\mathcal{S} + \epsilon} \cdot e_y \mathcal{S}^\top \boldsymbol{X}^\top$$

$$= -\frac{1}{e_y^\top V X \mathcal{S} + \epsilon} \cdot e_y \sum_{i=1}^{N-1} \mathcal{S}_i x_i^\top.$$

For $W$, we have

$$\nabla_W \ell(\theta) = -\frac{1}{e_y^\top f_\theta(X) + \epsilon} \cdot \frac{\partial e_y^\top V X \mathcal{S}(\widetilde{X}^\top W \widetilde{x}_N)}{\partial W}$$

$$= -\frac{1}{e_y^\top V X \mathcal{S} + \epsilon} \cdot \widetilde{X} \mathcal{S}'(\widetilde{X}^\top W \widetilde{x}_N) X^\top V^\top e_y \widetilde{x}_N^\top$$

$$= -\frac{1}{e_y^\top V X \mathcal{S} + \epsilon} \cdot \widetilde{X} [\mathrm{diag}(\mathcal{S}) - \mathcal{S}\mathcal{S}^\top] X^\top V^\top e_y \widetilde{x}_N^\top$$

$$= -\frac{1}{e_y^\top V X \mathcal{S} + \epsilon} \cdot \begin{bmatrix} \sum_{i=1}^{N-1} \mathcal{S}_i x_i x_i^\top - \sum_{i=1}^{N-1} \mathcal{S}_i x_i \cdot \sum_{i=1}^{N-1} \mathcal{S}_i x_i^\top \\ \sum_{i=1}^{N} \mathcal{S}_i p_i x_i^\top - \sum_{i=1}^{N} \mathcal{S}_i p_i \cdot \sum_{i=1}^{N-1} \mathcal{S}_i x_i^\top \end{bmatrix} \cdot \begin{bmatrix} \mathbf{0} & V^\top e_y p_N^\top \end{bmatrix}$$

$$= -\frac{1}{e_y^\top V X \mathcal{S} + \epsilon} \cdot \begin{bmatrix} \mathbf{0} & (\sum_{i=1}^{N-1} \mathcal{S}_i x_i x_i^\top - \sum_{i=1}^{N-1} \mathcal{S}_i x_i \cdot \sum_{i=1}^{N-1} \mathcal{S}_i x_i^\top) V^\top e_y p_N^\top \\ \mathbf{0} & (\sum_{i=1}^{N-1} \mathcal{S}_i p_i x_i^\top - \sum_{i=1}^{N} \mathcal{S}_i p_i \cdot \sum_{i=1}^{N-1} \mathcal{S}_i x_i^\top) V^\top e_y p_N^\top \end{bmatrix},$$

where we use the fact that $\mathcal{S}'(\widetilde{X}^\top W \widetilde{x}_N) = [\mathrm{diag}(\mathcal{S}) - \mathcal{S}\mathcal{S}^\top]$ and

$$\mathrm{diag}(\mathcal{S}) := \begin{bmatrix} \mathcal{S}_1 & & & \\ & \mathcal{S}_2 & & \\ & & \ddots & \\ & & & \mathcal{S}_N \end{bmatrix}.$$

$\square$

To simplify the notation, we denote

$$A := \left( \sum_{i=1}^{N-1} \mathcal{S}_i x_i x_i^\top - \sum_{i=1}^{N-1} \mathcal{S}_i x_i \cdot \sum_{i=1}^{N-1} \mathcal{S}_i x_i^\top \right) V^\top e_y p_N^\top,$$

$$B := \left( \sum_{i=1}^{N-1} \mathcal{S}_i p_i x_i^\top - \sum_{i=1}^{N} \mathcal{S}_i p_i \cdot \sum_{i=1}^{N-1} \mathcal{S}_i x_i^\top \right) V^\top e_y p_N^\top.$$

With this notations, by Lemma B.1, we have

$$\nabla_W \ell(\theta) = -\frac{1}{e_y^\top V X \mathcal{S} + \epsilon} \cdot \begin{bmatrix} \mathbf{0} & A \\ \mathbf{0} & B \end{bmatrix}. \tag{B.1}$$

We also denote $A^{(t)}, B^{(t)}$ the corresponding matrices at the $t$-th iteration of gradient descent. Moreover, we can observe that $W = \begin{bmatrix} W_{11} & W_{12} \\ W_{21} & W_{22} \end{bmatrix}$, where $W_{11} \in \mathbb{R}^{K \times K}$, $W_{12} \in \mathbb{R}^{K \times M}$, $W_{21} \in \mathbb{R}^{M \times K}$, and $W_{22} \in \mathbb{R}^{M \times M}$. By (B.1), we know that $W_{11}^{(t)} = \mathbf{0}_{K \times K}$ and $W_{21}^{(t)} = \mathbf{0}_{M \times K}$ for all $t \geq 1$.

By the definition of the random walks on circles, we can write the transition matrix $\Pi$ as $\Pi = p\Pi_0^\top + (1-p)\Pi_0$, where

$$\Pi_0 = \begin{bmatrix} 0 & & & & 1 \\ 1 & 0 & & & \\ & 1 & 0 & & \\ & & \ddots & \ddots & \\ & & & 1 & 0 \end{bmatrix}.$$

## C   Proof of Theorem 3.1

In this section, we analyze random walks with the transition probability $p$ satisfying $0 < p < 1$. Without loss of generality, we assume $\frac{1}{2} \leq p < 1$. We consider gradient descent starting from zero initialization. The following lemma presents the result of the first iteration.

**Lemma C.1.** Under the same conditions as Theorem 3.1, it holds that

$$\boldsymbol{V}^{(1)} = \frac{\eta}{\epsilon N K} \sum_{i=1}^{N-1} (\boldsymbol{\Pi}^\top)^{N-i} \text{ and } \boldsymbol{W}^{(1)} = \mathbf{0}_{(K+M)\times(K+M)}.$$

***Proof of Lemma C.1.*** By Lemma B.1, we have

$$\mathbb{E}[\nabla_{\boldsymbol{V}} \ell(\theta^{(0)})] = -\frac{1}{\epsilon N} \sum_{i=1}^{N-1} \mathbb{E}[\boldsymbol{e}_y \boldsymbol{x}_i^\top]$$

$$= -\frac{1}{\epsilon N} \sum_{i=1}^{N-1} \mathbb{E}[(\boldsymbol{\Pi}^\top)^{N-i} \boldsymbol{x}_i \boldsymbol{x}_i^\top]$$

$$= -\frac{1}{\epsilon N K} \sum_{i=1}^{N-1} (\boldsymbol{\Pi}^\top)^{N-i}$$

where the first equation is by the initialization of $\boldsymbol{V}^{(0)}$ and $\boldsymbol{W}^{(0)}$, the second equation is by the sampling method, and the third equation is by $\mathbb{E}[\boldsymbol{x}_i \boldsymbol{x}_i^\top] = \frac{1}{K} \mathbf{I}_K$ for $i \in [N-1]$ since $\boldsymbol{x}_i$ is uniformly distributed in $\boldsymbol{E}$. Thus, by the update, we can get

$$\boldsymbol{V}^{(1)} = \boldsymbol{V}^{(0)} - \eta \mathbb{E}[\nabla_{\boldsymbol{V}} \ell(\theta^{(0)})]$$

$$= \frac{\eta}{\epsilon N K} \sum_{i=1}^{N-1} (\boldsymbol{\Pi}^\top)^{N-i}.$$

Since $\boldsymbol{V}^{(0)} = \mathbf{0}_{K\times K}$ and $\boldsymbol{W}^{(0)} = \mathbf{0}_{(K+M)\times(K+M)}$, we can get $\mathbb{E}[\nabla_{\boldsymbol{W}} \ell(\theta^{(0)})] = \mathbf{0}_{(K+M)\times(K+M)}$. Thus,

$$\boldsymbol{W}^{(1)} = \boldsymbol{W}^{(0)} - \eta \mathbb{E}[\nabla_{\boldsymbol{W}} \ell(\theta^{(0)})] = \mathbf{0}_{(K+M)\times(K+M)}.$$

$\square$

Lemma C.1 gives explicit calculations of $\boldsymbol{V}^{(1)}$. Based on these, we can further derive some properties of $\boldsymbol{V}^{(1)}$, given in Lemma C.2 below. In this lemma, for all matrix indices, we consider simplified notations following the rule that if an index $i$ is not in the set $\{1, \ldots, K\}$, we treat it as $i' \in \{1, \ldots, K\}$ with $i' \equiv i \pmod{K}$.

**Lemma C.2.** Under the same conditions as Theorem 3.1, it holds that $[\boldsymbol{V}^{(1)}]_{i_1,j_2} = [\boldsymbol{V}^{(1)}]_{i_2,j_2}$ for $i_1 - j_1 \equiv i_2 - j_2 \pmod{K}$.

***Proof of Lemma C.2.*** We use induction to prove that for any $R \in \mathbb{N}$, $[\boldsymbol{\Pi}^R]_{i_1,j_1} = [\boldsymbol{\Pi}^R]_{i_2,j_2}$ for $i_1 - j_1 \equiv i_2 - j_2 \pmod{K}$. The result is obvious at $R = 1$. Suppose that the results hold for $\boldsymbol{\Pi}^R$. We aim to prove the results hold for $\boldsymbol{\Pi}^{R+1}$. By the definition of $\boldsymbol{\Pi}$, for any $i, j$, we obtain that

$$[\boldsymbol{\Pi}^{R+1}]_{i,j} = p \cdot [\boldsymbol{\Pi}^R]_{i,j-1} + (1-p) \cdot [\boldsymbol{\Pi}^R]_{i,j+1}. \tag{C.1}$$

By induction, for any $i_1, j_1, i_2, j_2$ satisfying $i_1 - j_1 \equiv i_2 - j_2 \pmod{K}$, we have $[\boldsymbol{\Pi}^R]_{i_1,j_1-1} = [\boldsymbol{\Pi}^R]_{i_2,j_2-1}$ and $[\boldsymbol{\Pi}^R]_{i_1,j_1+1} = [\boldsymbol{\Pi}^R]_{i_2,j_2+1}$. Thus, by (C.1), we can easily get $[\boldsymbol{\Pi}^{R+1}]_{i_1,j_1} = [\boldsymbol{\Pi}^{R+1}]_{i_2,j_2}$, which completes the induction.

From Lemma C.1, we know $(\boldsymbol{V}^{(1)})^\top = \frac{\eta}{\epsilon N K} \sum_{i=1}^{N-1} \boldsymbol{\Pi}^{N-i}$ holds the result. Therefore, $\boldsymbol{V}^{(1)}$ also has the same property. $\square$

**Lemma C.3.** Under the same conditions as Theorem 3.1, it holds that $[\boldsymbol{V}^{(1)}]_{2,1} = \|\boldsymbol{V}^{(1)}\|_{\max}$.

***Proof of Lemma C.3.*** Without loss of generality, we assume $\frac{1}{2} \leq p < 1$. From Lemma C.1, we know $\boldsymbol{V}^{(1)} = \frac{\eta}{\epsilon N K} \sum_{i=1}^{N-1} (\boldsymbol{\Pi}^\top)^{N-i}$. We can observe that since $p < 1$, with increasing $R$, $(\boldsymbol{\Pi}^\top)^R$ will be closer to $\frac{1}{K} \mathbf{1} \mathbf{1}^\top$, as stated in Lemma E.2 and E.3. Thus, the location of the largest entries in $\boldsymbol{V}^{(1)}$ is mainly determined by the first several terms. In $\boldsymbol{\Pi}^\top$, there are two locations with non-zero

values $p, 1-p$. In $(\boldsymbol{\Pi}^\top)^2$, there are three locations with non-zero values $p^2, 2p(1-p), (1-p)^2$. We can easily check that $p$ is larger than $p^2$ and $2p(1-p)$. Thus, we know the location of the value $p$ in $\boldsymbol{\Pi}^\top$ is also the largest in $\boldsymbol{V}^{(1)}$, i.e. $[\boldsymbol{V}^{(1)}]_{2,1}$. In detail, we can first get that

$$\sum_{i=0}^{N-1} (\boldsymbol{\Pi}^\top)^{N-i} = (\boldsymbol{\Pi}^\top - (\boldsymbol{\Pi}^\top)^N)(\boldsymbol{I} - \boldsymbol{\Pi}^\top)^{-1}$$

matrix of interest:

$$\boldsymbol{e}_1^\top \sum_{i=0}^{N-2} (\boldsymbol{\Pi}^\top)^i (\boldsymbol{\Pi}^\top \boldsymbol{\Pi} \boldsymbol{e}_1).$$

$$\boldsymbol{e}_1^\top \sum_{i=0}^{N-2} (\boldsymbol{\Pi}^\top)^i (\boldsymbol{\Pi}^\top \boldsymbol{\Pi}^j \boldsymbol{e}_1).$$

Denote

$$\boldsymbol{\Gamma}(N) = \sum_{i=0}^{N-2} (\boldsymbol{\Pi}^\top)^i.$$

Claim: for all $N$, the diagonal entries in $\boldsymbol{\Gamma}(N)$ are larger than or equal to the other entries.

$$\boldsymbol{\Gamma}(N) = \boldsymbol{\Pi}^\top \boldsymbol{\Gamma}(N-1) + \boldsymbol{I}.$$

Induction hypothesis:

$$\boldsymbol{\Gamma}(N)_{1,1} \geq \boldsymbol{\Gamma}(N)_{i,j} + c_p.$$

Suppose the induction hypothesis holds for $\boldsymbol{\Gamma}(N-1)$.

For $i, j$ with $|i-j| \geq 1$,

$$\boldsymbol{\Gamma}(N)_{ij} = p \cdot \boldsymbol{\Gamma}(N-1)_{i,j-1} + (1-p) \cdot (\boldsymbol{\Gamma}(N-1))_{i,j+1} \leq (\boldsymbol{\Gamma}(N-1))_{1,1} + c_p.$$

$$\boldsymbol{\Gamma}(N)_{1,1} \geq \boldsymbol{\Gamma}(N-1)_{1,1}$$

$\sum_{i=1}^{N-1}(\boldsymbol{\Pi}^\top)^{N-i} = (\boldsymbol{\Pi}^\top - (\boldsymbol{\Pi}^\top)^N) C_p \sum_{k=0}^{\infty} \left(\frac{1-p}{p}\right)^k (\boldsymbol{\Pi}_0^\top)^k$, where $C_p$ is a positive constant regarding $p$. Considering the terms $\boldsymbol{\Pi}^\top (\boldsymbol{\Pi}_0^\top)^k$ with $0 \leq k \leq K-1$, we can easily observe that the coefficient of $\boldsymbol{\Pi}^\top (\boldsymbol{\Pi}_0^\top)^0$ is much larger than that of other terms. Since $\boldsymbol{\Pi}_0^\top$ is a cyclic shift matrix and the entries in $(\boldsymbol{\Pi}^\top)^N$ are almost same (Lemma E.2 and E.3), the location of the largest entry in $\boldsymbol{\Pi}^\top$ is also that in $\boldsymbol{V}^{(1)}$. Therefore, it holds that $[\boldsymbol{V}^{(1)}]_{2,1} = \|\boldsymbol{V}^{(1)}\|_{\max}$. $\qquad\square$

Further, the following two lemmas provide some properties of the weights for the second iteration.

**Lemma C.4.** Under the same conditions as Theorem 3.1, it holds that $\|\boldsymbol{V}^{(2)}\|_{\max} \leq \frac{\eta}{\epsilon K} + 2\epsilon K^2$.

*Proof of Lemma C.4.* First, we have

$$\boldsymbol{V}^{(2)} = \boldsymbol{V}^{(1)} - \eta \mathbb{E}[\nabla_{\boldsymbol{V}} \ell(\theta^{(1)})] = \boldsymbol{V}^{(1)} + \eta \mathbb{E}\left[\frac{\boldsymbol{e}_y \sum_{i=1}^{N-1} \mathcal{S}_i^{(1)} \boldsymbol{x}_i^\top}{\boldsymbol{e}_y^\top \boldsymbol{V}^{(1)} \sum_{i=1}^{N-1} \mathcal{S}_i^{(1)} \boldsymbol{x}_i + \epsilon}\right].$$

Thus,

$$\|\boldsymbol{V}^{(2)}\|_{\max} \leq \left\|\boldsymbol{V}^{(1)}\right\|_{\max} + \left\|\eta \mathbb{E}\left[\frac{\frac{1}{N} \boldsymbol{e}_y \sum_{i=1}^{N-1} \boldsymbol{x}_i^\top}{\frac{1}{N} \boldsymbol{e}_y^\top \boldsymbol{V}^{(1)} \sum_{i=1}^{N-1} \boldsymbol{x}_i + \epsilon}\right]\right\|_{\max}$$

$$\leq \left\|\boldsymbol{V}^{(1)}\right\|_{\max} + \frac{\eta}{\frac{N-1}{N}\min_{i,j}[\boldsymbol{V}^{(1)}]_{i,j}} \cdot \left\|\mathbb{E}\left[\frac{1}{N}\boldsymbol{e}_y\sum_{i=1}^{N-1}\boldsymbol{x}_i^\top\right]\right\|_{\max}$$

$$\leq \left\|\frac{\eta}{\epsilon NK}\sum_{i=1}^{N-1}\boldsymbol{\Pi}^{N-i}\right\|_{\max} + \frac{\eta}{\frac{N-1}{N}\min_{i,j}\left[\frac{\eta}{\epsilon NK}\sum_{i=1}^{N-1}\boldsymbol{\Pi}^{N-i}\right]_{i,j}} \cdot \frac{N-1}{N}$$

$$\leq \frac{\eta}{\epsilon K} + 2\epsilon K^2,$$

where the second inequality is by $\boldsymbol{e}_y^\top\boldsymbol{V}^{(1)}\boldsymbol{x}_i \geq \min_{i,j}[\boldsymbol{V}^{(1)}]_{i,j}$, and the last inequality is by Lemma E.2 and E.3. $\qquad\square$

**Lemma C.5.** Under the same conditions as Theorem 3.1, it holds that $\mathcal{S}_{N-1}^{(2)} \geq \mathcal{S}_j^{(2)}\exp(\Omega(N))$ for $j \neq N-1$. Further, $\mathcal{S}_{N-1}^{(2)} \geq 1 - \exp(-\Omega(N))$ and $\mathcal{S}_j^{(2)} \leq \exp(-\Omega(N))$ for $j \neq N-1$.

*Proof of Lemma C.5.* By Lemma B.1, we have

$$\mathbb{E}[\boldsymbol{A}^{(1)}]$$

$$= \mathbb{E}\left[\left(\sum_{i=1}^{N-1}\mathcal{S}_i^{(1)}\boldsymbol{x}_i\boldsymbol{x}_i^\top(\boldsymbol{V}^{(1)})^\top\boldsymbol{e}_y - \sum_{i_1=1}^{N-1}\sum_{i_2=1}^{N-1}\mathcal{S}_{i_1}^{(1)}\mathcal{S}_{i_2}^{(1)}\boldsymbol{x}_{i_1}\boldsymbol{x}_{i_2}^\top(\boldsymbol{V}^{(1)})^\top\boldsymbol{e}_y\right)\boldsymbol{p}_N^\top\right]$$

$$= \mathbb{E}\left[\left(\frac{\eta}{\epsilon N^2 K}\sum_{i=1}^{N-1}\boldsymbol{x}_i\boldsymbol{x}_i^\top\sum_{i'=1}^{N-1}\boldsymbol{\Pi}^{N-i'}\boldsymbol{e}_y - \frac{\eta}{\epsilon N^3 K}\sum_{i_1=1}^{N-1}\sum_{i_2=1}^{N-1}\boldsymbol{x}_{i_1}\boldsymbol{x}_{i_2}^\top\sum_{i'=1}^{N-1}\boldsymbol{\Pi}^{N-i'}\boldsymbol{e}_y\right)\boldsymbol{p}_N^\top\right]$$

$$= \mathbb{E}\left[\frac{\eta}{\epsilon N^2 K}\sum_{i=1}^{N-1}\boldsymbol{x}_i\boldsymbol{x}_i^\top\sum_{i'=1}^{N-1}\boldsymbol{\Pi}^{N-i'}(\boldsymbol{\Pi}^\top)^{N-i}\boldsymbol{x}_i\cdot\boldsymbol{p}_N^\top\right]$$

$$\quad - \mathbb{E}\left[\frac{\eta}{\epsilon N^3 K}\sum_{i_1=1}^{N-1}\sum_{i_2=1}^{N-1}\boldsymbol{x}_{i_1}\boldsymbol{x}_{i_1}^\top\boldsymbol{\Pi}^{i_2-i_1}\sum_{i'=1}^{N-1}\boldsymbol{\Pi}^{N-i'}(\boldsymbol{\Pi}^\top)^{N-i_1}\boldsymbol{x}_{i_1}\cdot\boldsymbol{p}_N^\top\right]$$

$$= \mathbb{E}\left[\frac{\eta}{\epsilon N^2 K}\sum_{i=1}^{N-1}\sum_{i'=1}^{N-1}\boldsymbol{x}_i\boldsymbol{x}_i^\top\boldsymbol{\Pi}^{N-i'}(\boldsymbol{\Pi}^\top)^{N-i}\boldsymbol{x}_i\right]\boldsymbol{p}_N^\top$$

$$\quad - \mathbb{E}\left[\frac{\eta}{\epsilon N^3 K}\sum_{i_1=1}^{N-1}\sum_{i_2=1}^{N-1}\sum_{i'=1}^{N-1}\boldsymbol{x}_{i_1}\boldsymbol{x}_{i_1}^\top\boldsymbol{\Pi}^{N-i'+i_2-i_1}(\boldsymbol{\Pi}^\top)^{N-i_1}\boldsymbol{x}_{i_1}\cdot\boldsymbol{p}_N^\top\right]$$

$$= \frac{\eta}{\epsilon N^2 K^2}\sum_{i=1}^{N-1}\sum_{i'=1}^{N-1}\operatorname{tr}\left(\boldsymbol{\Pi}^{N-i'}(\boldsymbol{\Pi}^\top)^{N-i}\right)\boldsymbol{1}_K\boldsymbol{p}_N^\top$$

$$\quad - \frac{\eta}{\epsilon N^3 K^2}\sum_{i_1=1}^{N-1}\sum_{i_2=1}^{N-1}\sum_{i'=1}^{N-1}\operatorname{tr}\left(\boldsymbol{\Pi}^{N-i'+i_2-i_1}(\boldsymbol{\Pi}^\top)^{N-i_1}\right)\boldsymbol{1}_K\boldsymbol{p}_N^\top,$$

where the second equation is by Lemma C.1, the third equation is by the sampling method, and the fifth equation is by the fact that all the $\boldsymbol{x}_i$ is uniformly distributed in $\boldsymbol{E}$ for $i \in [N-1]$. Then, $\boldsymbol{W}_{12}^{(2)} = \boldsymbol{W}_{12}^{(1)} - \eta\mathbb{E}[\nabla_{\boldsymbol{W}}\ell(\theta^{(1)})]_{12} \propto \boldsymbol{1}_K\boldsymbol{p}_N^\top$. Thus, We also have

$$\mathbb{E}[\boldsymbol{B}^{(1)}]$$

$$= \mathbb{E}\left[\left(\sum_{i=1}^{N-1}\mathcal{S}_i^{(1)}\boldsymbol{p}_i\boldsymbol{x}_i^\top(\boldsymbol{V}^{(1)})^\top\boldsymbol{e}_y - \sum_{i=1}^{N}\mathcal{S}_i^{(1)}\boldsymbol{p}_i\cdot\sum_{i=1}^{N-1}\mathcal{S}_i^{(1)}\boldsymbol{x}_i^\top(\boldsymbol{V}^{(1)})^\top\boldsymbol{e}_y\right)\boldsymbol{p}_N^\top\right]$$

$$= \mathbb{E}\left[\left(\frac{\eta}{\epsilon N^2 K}\sum_{i=1}^{N-1}\boldsymbol{p}_i\boldsymbol{x}_i^\top\sum_{i'=1}^{N-1}\boldsymbol{\Pi}^{N-i'}\boldsymbol{e}_y - \frac{\eta}{\epsilon N^3 K}\sum_{i=1}^{N}\boldsymbol{p}_i\cdot\sum_{i=1}^{N-1}\boldsymbol{x}_i^\top\sum_{i'=1}^{N-1}\boldsymbol{\Pi}^{N-i'}\boldsymbol{e}_y\right)\boldsymbol{p}_N^\top\right]$$

$$= \mathbb{E}\left[\frac{\eta}{\epsilon N^2 K}\sum_{i=1}^{N-1}\boldsymbol{p}_i\boldsymbol{x}_i^\top\sum_{i'=1}^{N-1}\boldsymbol{\Pi}^{N-i'}(\boldsymbol{\Pi}^\top)^{N-i}\boldsymbol{x}_i\cdot\boldsymbol{p}_N^\top\right]$$

$$-\,\mathbb{E}\left[\frac{\eta}{\epsilon N^3 K}\sum_{i=1}^{N}\boldsymbol{p}_i\cdot\sum_{i=1}^{N-1}\boldsymbol{x}_i^\top\sum_{i'=1}^{N-1}\boldsymbol{\Pi}^{N-i'}(\boldsymbol{\Pi}^\top)^{N-i}\boldsymbol{x}_i\cdot\boldsymbol{p}_N^\top\right]$$

$$=\frac{\eta}{\epsilon N^2 K^2}\sum_{i=1}^{N-1}\sum_{i'=1}^{N-1}\boldsymbol{p}_i\,\mathrm{tr}\left(\boldsymbol{\Pi}^{N-i'}(\boldsymbol{\Pi}^\top)^{N-i}\right)\boldsymbol{p}_N^\top$$

$$-\frac{\eta}{\epsilon N^3 K^2}\sum_{i=1}^{N}\boldsymbol{p}_i\cdot\sum_{i=1}^{N-1}\sum_{i'=1}^{N-1}\mathrm{tr}\left(\boldsymbol{\Pi}^{N-i'}(\boldsymbol{\Pi}^\top)^{N-i}\right)\boldsymbol{p}_N^\top,$$

where the second equation is by Lemma C.1, the third equation is by the sampling method, and the last equation is by the fact that all the $\boldsymbol{x}_i$ is uniformly distributed in $\boldsymbol{E}$ for $i\in[N-1]$. Since $\left[\widetilde{\boldsymbol{X}}\boldsymbol{W}^{(2)}\widetilde{\boldsymbol{x}}_N\right]_N=\boldsymbol{p}_N^\top\boldsymbol{W}_{22}^{(2)}\boldsymbol{p}_N$ and $\left[\widetilde{\boldsymbol{X}}\boldsymbol{W}^{(2)}\widetilde{\boldsymbol{x}}_N\right]_j=\boldsymbol{x}_j^\top\boldsymbol{W}_{12}^{(2)}\boldsymbol{p}_N+\boldsymbol{p}_j^\top\boldsymbol{W}_{22}^{(2)}\boldsymbol{p}_N$ for $j\in\{1,2,\ldots,N-1\}$, we can obtain that

$$\left[\widetilde{\boldsymbol{X}}\boldsymbol{W}^{(2)}\widetilde{\boldsymbol{x}}_N\right]_N=\boldsymbol{p}_N^\top\boldsymbol{W}_{22}^{(2)}\boldsymbol{p}_N$$

$$=\mathbb{E}\left[\boldsymbol{p}_N^\top\frac{\eta}{\boldsymbol{e}_y^\top\boldsymbol{V}\boldsymbol{X}\mathcal{S}+\epsilon}\boldsymbol{B}^{(1)}\boldsymbol{p}_N\right]$$

$$=\mathbb{E}\left[\frac{\eta}{\boldsymbol{e}_y^\top\boldsymbol{V}\boldsymbol{X}\mathcal{S}+\epsilon}\left(-\mathcal{S}_N^{(1)}\boldsymbol{p}_N^\top\boldsymbol{p}_N\cdot\sum_{i=1}^{N-1}\mathcal{S}_i^{(1)}\boldsymbol{x}_i^\top(\boldsymbol{V}^{(1)})^\top\boldsymbol{e}_y\right)\boldsymbol{p}_N^\top\boldsymbol{p}_N\right]$$

$$<0,$$

where the third equation is by $\boldsymbol{p}_i^\top\boldsymbol{p}_j=0$ for $i\neq j$. And for $j\in\{1,2,\ldots,N-2\}$, we can get

$$\left[\widetilde{\boldsymbol{X}}\boldsymbol{W}^{(2)}\widetilde{\boldsymbol{x}}_N\right]_{N-1}-\left[\widetilde{\boldsymbol{X}}\boldsymbol{W}^{(2)}\widetilde{\boldsymbol{x}}_N\right]_j$$

$$=\boldsymbol{x}_{N-1}^\top\boldsymbol{W}_{12}^{(2)}\boldsymbol{p}_N+\boldsymbol{p}_{N-1}^\top\boldsymbol{W}_{22}^{(2)}\boldsymbol{p}_N-\boldsymbol{x}_j^\top\boldsymbol{W}_{12}^{(2)}\boldsymbol{p}_N-\boldsymbol{p}_j^\top\boldsymbol{W}_{22}^{(2)}\boldsymbol{p}_N$$

$$\overset{(i)}{=}\boldsymbol{p}_{N-1}^\top\boldsymbol{W}_{22}^{(2)}\boldsymbol{p}_N-\boldsymbol{p}_j^\top\boldsymbol{W}_{22}^{(2)}\boldsymbol{p}_N$$

$$\overset{(ii)}{=}\mathbb{E}\left[\frac{\eta}{\boldsymbol{e}_y^\top\boldsymbol{V}^{(1)}\boldsymbol{X}\mathcal{S}^{(1)}+\epsilon}\cdot\frac{1}{N}\left(\boldsymbol{p}_{N-1}^\top\boldsymbol{p}_{N-1}\boldsymbol{x}_{N-1}^\top(\boldsymbol{V}^{(1)})^\top\boldsymbol{e}_y-\boldsymbol{p}_j^\top\boldsymbol{p}_j\boldsymbol{x}_j^\top(\boldsymbol{V}^{(1)})^\top\boldsymbol{e}_y\right)\boldsymbol{p}_N^\top\boldsymbol{p}_N\right]$$

$$\overset{(iii)}{=}\mathbb{E}\left[\frac{\eta}{\boldsymbol{e}_y^\top\boldsymbol{V}^{(1)}\boldsymbol{X}\mathcal{S}^{(1)}+\epsilon}\cdot\frac{1}{N}\left(\boldsymbol{p}_{N-1}^\top\boldsymbol{p}_{N-1}\boldsymbol{x}_{N-1}^\top(\boldsymbol{V}^{(1)})^\top\boldsymbol{\Pi}^\top\boldsymbol{x}_{N-1}-\boldsymbol{p}_j^\top\boldsymbol{p}_j\boldsymbol{x}_j^\top(\boldsymbol{V}^{(1)})^\top(\boldsymbol{\Pi}^\top)^{N-j}\boldsymbol{x}_j\right)\boldsymbol{p}_N^\top\boldsymbol{p}_N\right]$$

$$\overset{(iv)}{=}\mathbb{E}\left[\frac{\eta}{\boldsymbol{e}_y^\top\boldsymbol{V}^{(1)}\boldsymbol{X}\mathcal{S}^{(1)}+\epsilon}\cdot\frac{(\boldsymbol{p}_N^\top\boldsymbol{p}_N)^2}{N}\left(\left[(\boldsymbol{V}^{(1)})^\top\boldsymbol{\Pi}^\top\right]_{1,1}-\left[(\boldsymbol{V}^{(1)})^\top(\boldsymbol{\Pi}^\top)^{N-j}\right]_{1,1}\right)\right]$$

$$\overset{(v)}{\geq}\frac{\eta}{\max_{i,j}[\boldsymbol{V}^{(1)}]_{i,j}+\epsilon}\cdot\frac{(\boldsymbol{p}_N^\top\boldsymbol{p}_N)^2}{N}\mathbb{E}\left[\left[(\boldsymbol{V}^{(1)})^\top\boldsymbol{\Pi}^\top\right]_{1,1}-\left[(\boldsymbol{V}^{(1)})^\top(\boldsymbol{\Pi}^\top)^{N-j}\right]_{1,1}\right]$$

$$\overset{(vi)}{\geq}\frac{\eta}{\frac{\eta}{\epsilon K}-\epsilon}\cdot\frac{\eta(\boldsymbol{p}_N^\top\boldsymbol{p}_N)^2}{\epsilon N^2 K}C_p$$

$$\geq\Omega\left(\frac{\eta M^2}{N^2}\right)$$

$$\geq\Omega(N),$$

where $(i)$ is by $\boldsymbol{W}_{12}^{(2)}\propto\mathbf{1}_K\boldsymbol{p}_N^\top$, $(ii)$ is by $\boldsymbol{p}_i^\top\boldsymbol{p}_{i'}=0$ for $i\neq i'$, $(iii)$ is by the sampling methods, $(iv)$ is by the fact that all the $\boldsymbol{x}_i$ is uniformly distributed in $\boldsymbol{E}$ for $i\in[N-1]$, $(v)$ and $(vi)$ are by Lemma E.4. Therefore, we have $\mathcal{S}_{N-1}^{(2)}/\mathcal{S}_j^{(2)}=\exp\left(\left[\widetilde{\boldsymbol{X}}\boldsymbol{W}^{(2)}\widetilde{\boldsymbol{x}}_N\right]_{N-1}-\left[\widetilde{\boldsymbol{X}}\boldsymbol{W}^{(2)}\widetilde{\boldsymbol{x}}_N\right]_j\right)\geq\exp(\Omega(N))$ for $j\neq N-1$. Further,

$$\mathcal{S}_{N-1}^{(2)}=1-\sum_{j\neq N-1}\mathcal{S}_j^{(2)}\geq 1-(N-1)\exp(-\Omega(N))\mathcal{S}_{N-1}^{(2)},$$

which implies that

$$\mathcal{S}_{N-1}^{(2)} \geq \frac{1}{1 + (N-1)\exp(-\Omega(N))} = 1 - \frac{N-1}{\exp(\Omega(N)) + N - 1} = 1 - \exp(-\Omega(N)).$$

Then, we have $\mathcal{S}_j^{(2)} \leq 1 - \mathcal{S}_{N-1}^{(2)} \leq \exp(-\Omega(N))$ for $j \neq N-1$. $\qquad\square$

Then, we can derive the bounds of entries in $\boldsymbol{V}^{(t)}$.

**Lemma C.6.** Under the same conditions as Theorem 3.1, it holds for $t \geq 3$ that

$$\min_{i,j}[\boldsymbol{V}^{(t)}]_{i,j} \geq \frac{\eta}{2\epsilon K^2} \text{ and } \|\boldsymbol{V}^{(t)}\|_{\max} \leq \frac{\eta}{\epsilon K} + (t-2)\cdot 2\epsilon K^2.$$

*Proof of Lemma C.6.* First, we have

$$\min_{i,j}[\boldsymbol{V}^{(t)}]_{i,j} \geq \min_{i,j}[\boldsymbol{V}^{(1)}]_{i,j}$$

$$\geq \min_{i,j}\left[\frac{\eta}{\epsilon N K}\sum_{i'=1}^{N-1}(\boldsymbol{\Pi}^\top)^{N-i'}\right]_{i,j}$$

$$\geq \frac{\eta}{\epsilon N K}\cdot\frac{N}{2K}$$

$$= \frac{\eta}{2\epsilon K^2},$$

where the third inequality is by Lemma E.2 and E.3. Then, we can get that

$$\|\boldsymbol{V}^{(t)}\|_{\max} \leq \|\boldsymbol{V}^{(t-1)}\|_{\max} + \left\|\mathbb{E}\left[\frac{\eta\boldsymbol{e}_y\sum_{i=1}^{N-1}\mathcal{S}_i^{(t-1)}\boldsymbol{x}_i^\top}{\boldsymbol{e}_y^\top\boldsymbol{V}^{(t-1)}\sum_{i=1}^{N-1}\mathcal{S}_i^{(t-1)}\boldsymbol{x}_i + \epsilon}\right]\right\|_{\max}$$

$$\leq \|\boldsymbol{V}^{(t-1)}\|_{\max} + \mathbb{E}\left[\frac{\eta\left\|\boldsymbol{e}_y\sum_{i=1}^{N-1}\mathcal{S}_i^{(t-1)}\boldsymbol{x}_i^\top\right\|_{\max}}{\min\left[\boldsymbol{e}_y^\top\boldsymbol{V}^{(t-1)}\sum_{i=1}^{N-1}\mathcal{S}_i^{(t-1)}\boldsymbol{x}_i\right]}\right]$$

$$\leq \|\boldsymbol{V}^{(t-1)}\|_{\max} + \frac{\eta}{\min_{i,j}[\boldsymbol{V}^{(t-1)}]_{i,j}}$$

$$\leq \|\boldsymbol{V}^{(t-1)}\|_{\max} + 2\epsilon K^2$$

$$\leq \|\boldsymbol{V}^{(2)}\|_{\max} + (t-3)\cdot 2\epsilon K^2$$

$$\leq \frac{\eta}{\epsilon K} + (t-2)\cdot 2\epsilon K^2,$$

where the third inequality is by $\boldsymbol{e}_y^\top\boldsymbol{V}\boldsymbol{x}_i \geq \min_{i,j}[\boldsymbol{V}]_{i,j}$, and the last inequality is by Lemma C.4. $\quad\square$

Next, we analyze the training dynamics over multiple iterations.

**Lemma C.7.** Assume the same conditions as Theorem 3.1. For $2 \leq t \leq T^*$, it holds that $\mathcal{S}_{N-1}^{(t)} \geq 1 - \exp(-\Omega(N))$ and $\boldsymbol{V}^{(t)} = \beta^{(t)}\boldsymbol{\Pi}^\top + \widetilde{\boldsymbol{V}}^{(t)}$ where $\left\|\widetilde{\boldsymbol{V}}^{(t)}\right\|_{\max} \leq \gamma^{(t)}$. Here, $\beta^{(t)} \geq \sqrt{\eta t} - \frac{2\eta}{\epsilon K}$ and $\gamma^{(t)} \leq \frac{2\eta}{\epsilon K} + 2(t-1)\epsilon K^2 N\exp(-\Omega(N))$.

*Proof of Lemma C.7.* We use induction to prove the results that

$$\beta^{(t)} \geq \sqrt{\eta t} - \frac{2\eta}{\epsilon K},$$

$$\gamma^{(t)} \leq \frac{2\eta}{\epsilon K} + 2(t-1)\epsilon K^2 N\exp(-\Omega(N)),$$

$$\left[\widetilde{\boldsymbol{X}}\boldsymbol{W}^{(t)}\widetilde{\boldsymbol{x}}_N\right]_{N-1} - \left[\widetilde{\boldsymbol{X}}\boldsymbol{W}^{(t)}\widetilde{\boldsymbol{x}}_N\right]_j \geq \Omega(N),$$

$$\mathcal{S}_{N-1}^{(t)} \geq 1 - \exp(-\Omega(N)).$$

It can be easily checked that the results hold for $t = 2$. Suppose that the results hold for the $t$-th iteration. We aim to prove that the results hold for $t + 1$.

For $\boldsymbol{V}^{(t+1)}$, we can get

$$
\begin{aligned}
\boldsymbol{V}^{(t+1)} &= \boldsymbol{V}^{(t)} - \eta \mathbb{E}[\nabla_{\boldsymbol{V}} \ell(\theta^{(t)})] \\
&= \boldsymbol{V}^{(t)} + \eta \mathbb{E}\left[ \frac{\boldsymbol{e}_y \sum_{i=1}^{N-1} \mathcal{S}_i^{(t)} \boldsymbol{x}_i^\top}{\boldsymbol{e}_y^\top \boldsymbol{V}^{(t)} \sum_{i=1}^{N-1} \mathcal{S}_i^{(t)} \boldsymbol{x}_i + \epsilon} \right] \\
&= \boldsymbol{V}^{(t)} + \mathbb{E}\left[ \frac{\eta \mathcal{S}_{N-1}^{(t)} \boldsymbol{e}_y \boldsymbol{x}_{N-1}^\top}{\boldsymbol{e}_y^\top \boldsymbol{V}^{(t)} \sum_{i=1}^{N-1} \mathcal{S}_i^{(t)} \boldsymbol{x}_i + \epsilon} \right] + \mathbb{E}\left[ \frac{\eta \boldsymbol{e}_y \sum_{i=1}^{N-2} \mathcal{S}_i^{(t)} \boldsymbol{x}_i^\top}{\boldsymbol{e}_y^\top \boldsymbol{V}^{(t)} \sum_{i=1}^{N-1} \mathcal{S}_i^{(t)} \boldsymbol{x}_i + \epsilon} \right] \\
&= \boldsymbol{V}^{(t)} + \mathbb{E}\left[ \frac{\eta \mathcal{S}_{N-1}^{(t)} \boldsymbol{x}_{N-1} \boldsymbol{x}_{N-1}^\top}{\boldsymbol{e}_y^\top \boldsymbol{V}^{(t)} \sum_{i=1}^{N-1} \mathcal{S}_i^{(t)} \boldsymbol{x}_i + \epsilon} \right] \boldsymbol{\Pi}^\top + \mathbb{E}\left[ \frac{\eta \boldsymbol{e}_y \sum_{i=1}^{N-2} \mathcal{S}_i^{(t)} \boldsymbol{x}_i^\top}{\boldsymbol{e}_y^\top \boldsymbol{V}^{(t)} \sum_{i=1}^{N-1} \mathcal{S}_i^{(t)} \boldsymbol{x}_i + \epsilon} \right]
\end{aligned}
$$

Then, we have

$$
\begin{aligned}
\left[ \mathbb{E}\left[ \frac{\eta \mathcal{S}_{N-1}^{(t)} \boldsymbol{x}_{N-1} \boldsymbol{x}_{N-1}^\top}{\boldsymbol{e}_y^\top \boldsymbol{V}^{(t)} \sum_{i=1}^{N-1} \mathcal{S}_i^{(t)} \boldsymbol{x}_i + \epsilon} \right] \right]_{1,1} &\geq \frac{\eta \mathcal{S}_{N-1}^{(t)}}{\|\boldsymbol{V}^{(t)}\|_{\max} + \epsilon} \\
&\geq \frac{\eta[1 - \exp(-\Omega(N))]}{\beta^{(t)} + \gamma^{(t)} + \epsilon} \\
&\geq \frac{\eta}{2 \left[ \beta^{(t)} + \frac{\eta}{\epsilon K} + 2\epsilon K^2 + 2t\epsilon K^2 N \exp(-\Omega(N)) + \epsilon \right]} \\
&\geq \frac{\eta}{2 \left( \beta^{(t)} + \frac{2\eta}{\epsilon K} \right)}, \quad \text{(C.2)}
\end{aligned}
$$

where the first inequality is by $\boldsymbol{e}_y^\top \boldsymbol{V}^{(t)} \boldsymbol{x}_i \leq \|\boldsymbol{V}^{(t)}\|_{\max}$, the second inequality is by induction, and the third inequality is by the assumption of $\epsilon$. And, we have

$$
\begin{aligned}
\left\| \mathbb{E}\left[ \frac{\eta \boldsymbol{e}_y \sum_{i=1}^{N-2} \mathcal{S}_i^{(t)} \boldsymbol{x}_i^\top}{\boldsymbol{e}_y^\top \boldsymbol{V}^{(t)} \sum_{i=1}^{N-1} \mathcal{S}_i^{(t)} \boldsymbol{x}_i + \epsilon} \right] \right\|_{\max} &\leq \frac{\eta \exp(-\Omega(N))}{\min_{i,j}[\boldsymbol{V}^{(t)}]_{i,j}} \cdot \left\| \mathbb{E}\left[ \sum_{i=1}^{N-2} \boldsymbol{e}_y \boldsymbol{x}_i^\top \right] \right\|_{\max} \\
&\leq 2\epsilon K^2 \exp(-\Omega(N)) \cdot N, \quad \text{(C.3)}
\end{aligned}
$$

where the first inequality is by induction and $\boldsymbol{e}_y^\top \boldsymbol{V}^{(t)} \boldsymbol{x}_i \geq \min_{i,j}[\boldsymbol{V}^{(t)}]_{i,j}$, and the second inequality is by Lemma C.6. Thus, we can get that

$$
\begin{aligned}
\beta^{(t+1)} &\geq \beta^{(t)} + \left[ \mathbb{E}\left[ \frac{\eta \mathcal{S}_{N-1}^{(t)} \boldsymbol{x}_{N-1} \boldsymbol{x}_{N-1}^\top}{\boldsymbol{e}_y^\top \boldsymbol{V}^{(t)} \sum_{i=1}^{N-1} \mathcal{S}_i^{(t)} \boldsymbol{x}_i + \epsilon} \right] \right]_{1,1} \\
&\geq \beta^{(t)} + \frac{\eta}{2\beta^{(t)} + \frac{4\eta}{\epsilon K}} \\
&\geq \sqrt{\eta t} - \frac{2\eta}{\epsilon K} + \frac{\eta}{2\sqrt{\eta t}} \\
&\geq \sqrt{\eta t} - \frac{2\eta}{\epsilon K} + \frac{\sqrt{\eta}}{\sqrt{t+1} + \sqrt{t}} \\
&= \sqrt{\eta(t+1)} - \frac{2\eta}{\epsilon K}
\end{aligned}
$$

where the second inequality is by (C.2), and the third inequality is by induction and the fact that $x + \frac{\eta}{2x + \frac{4\eta}{\epsilon K}}$ is monotonically increasing for $x \geq \frac{\sqrt{\eta}}{\sqrt{2}} - \frac{2\eta}{\epsilon K}$. And, we can get

$$
\begin{aligned}
\gamma^{(t+1)} &\leq \gamma^{(t)} + \left\| \mathbb{E}\left[ \frac{\eta \boldsymbol{e}_y \sum_{i=1}^{N-2} \mathcal{S}_i^{(t)} \boldsymbol{x}_i^\top}{\boldsymbol{e}_y^\top \boldsymbol{V}^{(t)} \sum_{i=1}^{N-1} \mathcal{S}_i^{(t)} \boldsymbol{x}_i + \epsilon} \right] \right\|_{\max} \\
&\leq \gamma^{(t)} + 2\epsilon K^2 N \exp(-\Omega(N))
\end{aligned}
$$

$$\leq \frac{2\eta}{\epsilon K} + 2t\epsilon K^2 N \exp(-\Omega(N)),$$

where the second inequality is by (C.3), and the third inequality is by induction.

Next, we consider $\mathcal{S}^{(t+1)}$. Recall that

$$\boldsymbol{W}_{12}^{(t+1)} = \boldsymbol{W}_{12}^{(t)} + \eta \mathbb{E}\left[\frac{\boldsymbol{A}^{(t)}}{\boldsymbol{e}_y^\top \boldsymbol{V} \boldsymbol{X} \mathcal{S} + \epsilon}\right] \text{ and } \boldsymbol{W}_{22}^{(t+1)} = \boldsymbol{W}_{22}^{(t)} + \eta \mathbb{E}\left[\frac{\boldsymbol{B}^{(t)}}{\boldsymbol{e}_y^\top \boldsymbol{V} \boldsymbol{X} \mathcal{S} + \epsilon}\right],$$

where

$$\boldsymbol{A}^{(t)} = \left(\sum_{i=1}^{N-1} \mathcal{S}_i^{(t)} \boldsymbol{x}_i \boldsymbol{x}_i^\top (\boldsymbol{V}^{(t)})^\top \boldsymbol{e}_y - \sum_{i_1=1}^{N-1}\sum_{i_2=1}^{N-1} \mathcal{S}_{i_1}^{(t)} \mathcal{S}_{i_2}^{(t)} \boldsymbol{x}_{i_1} \boldsymbol{x}_{i_2}^\top (\boldsymbol{V}^{(t)})^\top \boldsymbol{e}_y\right) \boldsymbol{p}_N^\top,$$

$$\boldsymbol{B}^{(t)} = \left(\sum_{i=1}^{N-1} \mathcal{S}_i^{(t)} \boldsymbol{p}_i \boldsymbol{x}_i^\top (\boldsymbol{V}^{(t)})^\top \boldsymbol{e}_y - \sum_{i=1}^{N} \mathcal{S}_i^{(t)} \boldsymbol{p}_i \cdot \sum_{i=1}^{N-1} \mathcal{S}_i^{(t)} \boldsymbol{x}_i^\top (\boldsymbol{V}^{(t)})^\top \boldsymbol{e}_y\right) \boldsymbol{p}_N^\top.$$

We also have $\left[\widetilde{\boldsymbol{X}} \boldsymbol{W}^{(t)} \widetilde{\boldsymbol{x}}_N\right]_N = \boldsymbol{p}_N^\top \boldsymbol{W}_{22}^{(t)} \boldsymbol{p}_N$ and $\left[\widetilde{\boldsymbol{X}} \boldsymbol{W}^{(t)} \widetilde{\boldsymbol{x}}_N\right]_j = \boldsymbol{x}_j^\top \boldsymbol{W}_{12}^{(t)} \boldsymbol{p}_N + \boldsymbol{p}_j^\top \boldsymbol{W}_{22}^{(t)} \boldsymbol{p}_N$ for $j \in \{1, 2, \ldots, N-1\}$. Then, for $j = N$, we have

$$\boldsymbol{p}_N^\top \boldsymbol{W}_{22}^{(t+1)} \boldsymbol{p}_N = \boldsymbol{p}_N^\top \boldsymbol{W}_{22}^{(t)} \boldsymbol{p}_N + \eta \mathbb{E}\left[\frac{\boldsymbol{p}_N^\top \boldsymbol{B}^{(t)} \boldsymbol{p}_N}{\boldsymbol{e}_y^\top \boldsymbol{V}^{(t)} \sum_{i=1}^{N-1} \mathcal{S}_i^{(t)} \boldsymbol{x}_i + \epsilon}\right]$$

$$= \boldsymbol{p}_N^\top \boldsymbol{W}_{22}^{(t)} \boldsymbol{p}_N - \eta M \mathcal{S}_N^{(t)} \mathbb{E}\left[\frac{\sum_{i=1}^{N-1} \mathcal{S}_i^{(t)} \boldsymbol{x}_i^\top \boldsymbol{V}^{(t)} \boldsymbol{e}_y}{\boldsymbol{e}_y^\top \boldsymbol{V}^{(t)} \sum_{i=1}^{N-1} \mathcal{S}_i^{(t)} \boldsymbol{x}_i + \epsilon}\right]$$

$$\leq \boldsymbol{p}_N^\top \boldsymbol{W}_{22}^{(t)} \boldsymbol{p}_N.$$

For $j \in \{1, 2, \ldots, N-2\}$, we have

$$\boldsymbol{x}_j^\top \boldsymbol{W}_{12}^{(t+1)} \boldsymbol{p}_N = \boldsymbol{x}_j^\top \boldsymbol{W}_{12}^{(t)} \boldsymbol{p}_N + \eta \mathbb{E}\left[\frac{\boldsymbol{x}_j^\top \boldsymbol{A}^{(t)} \boldsymbol{p}_N}{\boldsymbol{e}_y^\top \boldsymbol{V}^{(t)} \sum_{i=1}^{N-1} \mathcal{S}_i^{(t)} \boldsymbol{x}_i + \epsilon}\right]$$

$$= \boldsymbol{x}_j^\top \boldsymbol{W}_{12}^{(t)} \boldsymbol{p}_N + \eta \sqrt{M} \mathcal{S}_j^{(t)} \mathbb{E}\left[\frac{\boldsymbol{x}_j^\top \boldsymbol{V}^{(t)} \boldsymbol{e}_y - \sum_{i_2=1}^{N-1} \mathcal{S}_{i_2}^{(t)} \boldsymbol{x}_{i_2}^\top \boldsymbol{V}^{(t)} \boldsymbol{e}_y}{\boldsymbol{e}_y^\top \boldsymbol{V}^{(t)} \sum_{i=1}^{N-1} \mathcal{S}_i^{(t)} \boldsymbol{x}_i + \epsilon}\right]$$

$$\leq \boldsymbol{x}_j^\top \boldsymbol{W}_{12}^{(t)} \boldsymbol{p}_N + \eta \sqrt{M} \mathcal{S}_j^{(t)} \frac{\|\boldsymbol{V}^{(t)}\|_{\max}}{\min_{i,j}[\boldsymbol{V}^{(t)}]_{i,j}}$$

$$\leq \boldsymbol{x}_j^\top \boldsymbol{W}_{12}^{(t)} \boldsymbol{p}_N + \eta \sqrt{M} \exp(-\Omega(N)) \left(2K + \frac{4(t-2)\epsilon^2 K^4}{\eta}\right),$$

where the first inequality is by $\boldsymbol{e}_y^\top \boldsymbol{V}^{(t)} \boldsymbol{x}_{N-1} = \|\boldsymbol{V}^{(t)}\|_{\max}$, and the second inequality is by induction and Lemma C.6. And,

$$\boldsymbol{p}_j^\top \boldsymbol{W}_{22}^{(t+1)} \boldsymbol{p}_N = \boldsymbol{p}_j^\top \boldsymbol{W}_{22}^{(t)} \boldsymbol{p}_N + \eta \mathbb{E}\left[\frac{\boldsymbol{p}_j^\top \boldsymbol{B}^{(t)} \boldsymbol{p}_N}{\boldsymbol{e}_y^\top \boldsymbol{V}^{(t)} \sum_{i=1}^{N-1} \mathcal{S}_i^{(t)} \boldsymbol{x}_i + \epsilon}\right]$$

$$= \boldsymbol{p}_j^\top \boldsymbol{W}_{22}^{(t)} \boldsymbol{p}_N + \eta M \mathcal{S}_j^{(t)} \mathbb{E}\left[\frac{\boldsymbol{x}_j^\top \boldsymbol{V}^{(t)} \boldsymbol{e}_y - \sum_{i=1}^{N-1} \mathcal{S}_i^{(t)} \boldsymbol{x}_i^\top \boldsymbol{V}^{(t)} \boldsymbol{e}_y}{\boldsymbol{e}_y^\top \boldsymbol{V}^{(t)} \sum_{i=1}^{N-1} \mathcal{S}_i^{(t)} \boldsymbol{x}_i + \epsilon}\right]$$

$$\leq \boldsymbol{p}_j^\top \boldsymbol{W}_{22}^{(t)} \boldsymbol{p}_N + \eta M \mathcal{S}_j^{(t)} \frac{\|\boldsymbol{V}^{(t)}\|_{\max}}{\min_{i,j}[\boldsymbol{V}^{(t)}]_{i,j}}$$

$$\leq \boldsymbol{p}_j^\top \boldsymbol{W}_{22}^{(t)} \boldsymbol{p}_N + \eta M \exp(-\Omega(N)) \left(2K + \frac{4(t-2)\epsilon^2 K^4}{\eta}\right),$$

where the first inequality is by $\boldsymbol{e}_y^\top \boldsymbol{V}^{(t)} \boldsymbol{x}_{N-1} = \|\boldsymbol{V}^{(t)}\|_{\max}$, and the second inequality is by induction and Lemma C.6. For $j = N - 1$, we have

$$\boldsymbol{x}_{N-1}^\top \boldsymbol{W}_{12}^{(t+1)} \boldsymbol{p}_N = \boldsymbol{x}_{N-1}^\top \boldsymbol{W}_{12}^{(t)} \boldsymbol{p}_N + \eta \mathbb{E}\left[\frac{\boldsymbol{x}_{N-1}^\top \boldsymbol{A}^{(t)} \boldsymbol{p}_N}{\boldsymbol{e}_y^\top \boldsymbol{V}^{(t)} \sum_{i=1}^{N-1} \mathcal{S}_i^{(t)} \boldsymbol{x}_i + \epsilon}\right]$$

$$= \boldsymbol{x}_{N-1}^\top \boldsymbol{W}_{12}^{(t)} \boldsymbol{p}_N + \eta \sqrt{M} \mathcal{S}_{N-1}^{(t)} \mathbb{E} \left[ \frac{\boldsymbol{x}_{N-1}^\top \boldsymbol{V}^{(t)} \boldsymbol{e}_y - \sum_{i_2=1}^{N-1} \mathcal{S}_{i_2}^{(t)} \boldsymbol{x}_{i_2}^\top \boldsymbol{V}^{(t)} \boldsymbol{e}_y}{\boldsymbol{e}_y^\top \boldsymbol{V}^{(t)} \sum_{i=1}^{N-1} \mathcal{S}_i^{(t)} \boldsymbol{x}_i + \epsilon} \right]$$

$$\geq \boldsymbol{x}_{N-1}^\top \boldsymbol{W}_{12}^{(t)} \boldsymbol{p}_N + \eta \sqrt{M} \mathbb{E} \left[ \frac{-\sum_{i_2=1}^{N-2} \mathcal{S}_{i_2}^{(t)} \boldsymbol{x}_{i_2}^\top \boldsymbol{V}^{(t)} \boldsymbol{e}_y}{\boldsymbol{e}_y^\top \boldsymbol{V}^{(t)} \sum_{i=1}^{N-1} \mathcal{S}_i^{(t)} \boldsymbol{x}_i + \epsilon} \right]$$

$$\geq \boldsymbol{x}_{N-1}^\top \boldsymbol{W}_{12}^{(t)} \boldsymbol{p}_N - \eta \sqrt{M} \frac{\sum_{i_2=1}^{N-2} \mathcal{S}_{i_2}^{(t)} \|\boldsymbol{V}^{(t)}\|_{\max}}{\min_{i,j} [\boldsymbol{V}^{(t)}]_{i,j}}$$

$$\geq \boldsymbol{x}_{N-1}^\top \boldsymbol{W}_{12}^{(t)} \boldsymbol{p}_N - \eta \sqrt{M} N \exp(-\Omega(N)) \left( 2K + \frac{4(t-2)\epsilon^2 K^4}{\eta} \right),$$

where the second inequality is by $\boldsymbol{e}_y^\top \boldsymbol{V}^{(t)} \boldsymbol{x}_{N-1} = \|\boldsymbol{V}^{(t)}\|_{\max}$, and the third inequality is by induction and Lemma C.6. And,

$$\boldsymbol{p}_{N-1}^\top \boldsymbol{W}_{22}^{(t+1)} \boldsymbol{p}_N = \boldsymbol{p}_{N-1}^\top \boldsymbol{W}_{22}^{(t)} \boldsymbol{p}_N + \eta \mathbb{E} \left[ \frac{\boldsymbol{p}_{N-1}^\top \boldsymbol{B}^{(t)} \boldsymbol{p}_N}{\boldsymbol{e}_y^\top \boldsymbol{V}^{(t)} \sum_{i=1}^{N-1} \mathcal{S}_i^{(t)} \boldsymbol{x}_i + \epsilon} \right]$$

$$= \boldsymbol{p}_{N-1}^\top \boldsymbol{W}_{22}^{(t)} \boldsymbol{p}_N + \eta M \mathcal{S}_{N-1}^{(t)} \mathbb{E} \left[ \frac{\boldsymbol{x}_{N-1}^\top \boldsymbol{V}^{(t)} \boldsymbol{e}_y - \sum_{i=1}^{N-1} \mathcal{S}_i^{(t)} \boldsymbol{x}_i^\top \boldsymbol{V}^{(t)} \boldsymbol{e}_y}{\boldsymbol{e}_y^\top \boldsymbol{V}^{(t)} \sum_{i=1}^{N-1} \mathcal{S}_i^{(t)} \boldsymbol{x}_i + \epsilon} \right]$$

$$\geq \boldsymbol{p}_{N-1}^\top \boldsymbol{W}_{22}^{(t)} \boldsymbol{p}_N + \eta M \mathbb{E} \left[ \frac{-\sum_{i=1}^{N-2} \mathcal{S}_i^{(t)} \boldsymbol{x}_i^\top \boldsymbol{V}^{(t)} \boldsymbol{e}_y}{\boldsymbol{e}_y^\top \boldsymbol{V}^{(t)} \sum_{i=1}^{N-1} \mathcal{S}_i^{(t)} \boldsymbol{x}_i + \epsilon} \right]$$

$$\geq \boldsymbol{p}_{N-1}^\top \boldsymbol{W}_{22}^{(t)} \boldsymbol{p}_N - \eta M \frac{\sum_{i_2=1}^{N-2} \mathcal{S}_{i_2}^{(t)} \|\boldsymbol{V}^{(t)}\|_{\max}}{\min_{i,j} [\boldsymbol{V}^{(t)}]_{i,j}}$$

$$\geq \boldsymbol{p}_{N-1}^\top \boldsymbol{W}_{22}^{(t)} \boldsymbol{p}_N - \eta M N \exp(-\Omega(N)) \left( 2K + \frac{4(t-2)\epsilon^2 K^4}{\eta} \right),$$

where the second inequality is by $\boldsymbol{e}_y^\top \boldsymbol{V}^{(t)} \boldsymbol{x}_{N-1} = \|\boldsymbol{V}^{(t)}\|_{\max}$, and the third inequality is by induction and Lemma C.6. Therefore, we can get that for $j \neq N-1$,

$$\left[ \widetilde{\boldsymbol{X}} \boldsymbol{W}^{(t+1)} \widetilde{\boldsymbol{x}}_N \right]_{N-1} - \left[ \widetilde{\boldsymbol{X}} \boldsymbol{W}^{(t+1)} \widetilde{\boldsymbol{x}}_N \right]_j$$

$$= \boldsymbol{x}_{N-1}^\top \boldsymbol{W}_{12}^{(t+1)} \boldsymbol{p}_N + \boldsymbol{p}_{N-1}^\top \boldsymbol{W}_{22}^{(t+1)} \boldsymbol{p}_N - \boldsymbol{x}_j^\top \boldsymbol{W}_{12}^{(t+1)} \boldsymbol{p}_N - \boldsymbol{p}_j^\top \boldsymbol{W}_{22}^{(t+1)} \boldsymbol{p}_N$$

$$\geq \boldsymbol{x}_{N-1}^\top \boldsymbol{W}_{12}^{(t)} \boldsymbol{p}_N + \boldsymbol{p}_{N-1}^\top \boldsymbol{W}_{22}^{(t)} \boldsymbol{p}_N - \boldsymbol{x}_j^\top \boldsymbol{W}_{12}^{(t)} \boldsymbol{p}_N - \boldsymbol{p}_j^\top \boldsymbol{W}_{22}^{(t)} \boldsymbol{p}_N$$

$$\quad - 4\eta M N \exp(-\Omega(N)) \left( 2K + \frac{4(t-2)\epsilon^2 K^4}{\eta} \right)$$

$$= \left[ \widetilde{\boldsymbol{X}} \boldsymbol{W}^{(t)} \widetilde{\boldsymbol{x}}_N \right]_{N-1} - \left[ \widetilde{\boldsymbol{X}} \boldsymbol{W}^{(t)} \widetilde{\boldsymbol{x}}_N \right]_j - \exp(-\Omega(N))$$

$$\geq \Omega(N),$$

where the last inequality is by induction. Thus, $\mathcal{S}_{N-1}^{(t+1)} / \mathcal{S}_j^{(t+1)} \geq \exp(\Omega(N))$ for $j \neq N-1$, which implies that $\mathcal{S}_{N-1}^{(t+1)} \geq 1 - \exp(-\Omega(N))$. Therefore, we prove that the results hold for $t+1$, which completes the proof. $\qquad\square$

The next two lemmas show the convergence rates of $\boldsymbol{V}^{(T)} / \|\boldsymbol{V}^{(T)}\|_F$ and $f_{\theta^T}(\boldsymbol{X}) / \|f_{\theta^T}(\boldsymbol{X})\|_2$.

**Lemma C.8.** Assume the same conditions as Theorem 3.1. For $\Omega(\eta \epsilon^{-2} K^{-2}) \leq T \leq T^*$, it holds that

$$\left\| \frac{\boldsymbol{V}^{(T)}}{\|\boldsymbol{V}^{(T)}\|_F} - \frac{\boldsymbol{\Pi}^\top}{\|\boldsymbol{\Pi}^\top\|_F} \right\|_F \leq \mathcal{O}\left( \frac{1}{\sqrt{T}} \right).$$

*Proof of Lemma C.8.* By Lemma C.7, we can get that

$$\left\| \frac{\boldsymbol{V}^{(T)}}{\|\boldsymbol{V}^{(T)}\|_F} - \frac{\boldsymbol{\Pi}^\top}{\|\boldsymbol{\Pi}^\top\|_F} \right\|_F = \left\| \frac{\beta^{(T)} \boldsymbol{\Pi}^\top + \widetilde{\boldsymbol{V}}^{(T)}}{\|\boldsymbol{V}^{(T)}\|_F} - \frac{\boldsymbol{\Pi}^\top}{\|\boldsymbol{\Pi}^\top\|_F} \right\|_F$$

$$\leq \left\|\left(\frac{\beta^{(T)}}{\|\boldsymbol{V}^{(T)}\|_F} - \frac{1}{\|\boldsymbol{\Pi}^\top\|_F}\right)\boldsymbol{\Pi}^\top\right\|_F + \left\|\frac{\widetilde{\boldsymbol{V}}^{(T)}}{\|\boldsymbol{V}^{(T)}\|_F}\right\|_F.$$

For the first part, we have

$$
\begin{aligned}
\left\|\left(\frac{\beta^{(T)}}{\|\boldsymbol{V}^{(T)}\|_F} - \frac{1}{\|\boldsymbol{\Pi}^\top\|_F}\right)\boldsymbol{\Pi}^\top\right\|_F &= \left|\frac{\beta^{(T)}\|\boldsymbol{\Pi}^\top\|_F}{\|\beta^{(T)}\boldsymbol{\Pi}^\top + \widetilde{\boldsymbol{V}}^{(T)}\|_F} - 1\right| \\
&\leq 1 - \frac{\beta^{(T)}\|\boldsymbol{\Pi}^\top\|_F}{\beta^{(T)}\|\boldsymbol{\Pi}^\top\|_F + \|\widetilde{\boldsymbol{V}}^{(T)}\|_F} \\
&\overset{(i)}{=} \frac{\|\widetilde{\boldsymbol{V}}^{(T)}\|_F}{\sqrt{K(p^2 + (1-p)^2)}\beta^{(T)} + \|\widetilde{\boldsymbol{V}}^{(T)}\|_F} \\
&\leq \frac{K\gamma^{(T)}}{\sqrt{K(p^2 + (1-p)^2)}\beta^{(T)} + K\gamma^{(T)}} \\
&\overset{(ii)}{\leq} \frac{\frac{2\eta}{\epsilon} + 2T\epsilon K^3 N \exp(-\Omega(N))}{\frac{\sqrt{2K}}{2}\left(\sqrt{\eta T} - \frac{2\eta}{\epsilon K}\right) + \frac{2\eta}{\epsilon} + 2T\epsilon K^3 N \exp(-\Omega(N))} \\
&\leq \mathcal{O}\left(\frac{1}{\sqrt{T}}\right),
\end{aligned}
$$

where $(i)$ is by $\|\boldsymbol{\Pi}^\top\|_F = \sqrt{K(p^2 + (1-p)^2)}$, and $(ii)$ is by Lemma C.7. For the second part, we have

$$
\begin{aligned}
\left\|\frac{\widetilde{\boldsymbol{V}}^{(T)}}{\|\boldsymbol{V}^{(T)}\|_F}\right\|_F &= \frac{\|\widetilde{\boldsymbol{V}}^{(T)}\|_F}{\|\beta^{(T)}\boldsymbol{\Pi}^\top + \widetilde{\boldsymbol{V}}^{(T)}\|_F} \\
&\leq \frac{\|\widetilde{\boldsymbol{V}}^{(T)}\|_F}{\|\beta^{(T)}\boldsymbol{\Pi}^\top\|_F + \|\widetilde{\boldsymbol{V}}^{(T)}\|_F} \\
&\overset{(i)}{\leq} \frac{K\gamma^{(T)}}{\frac{\sqrt{2K}}{2}\beta^{(T)} + K\gamma^{(T)}} \\
&\overset{(ii)}{\leq} \frac{\frac{2\eta}{\epsilon} + 2T\epsilon K^3 N \exp(-\Omega(N))}{\frac{\sqrt{2K}}{2}\left(\sqrt{\eta T} - \frac{2\eta}{\epsilon K}\right) + \frac{2\eta}{\epsilon} + 2T\epsilon K^3 N \exp(-\Omega(N))} \\
&\leq \mathcal{O}\left(\frac{1}{\sqrt{T}}\right),
\end{aligned}
$$

where $(i)$ is by $\|\boldsymbol{\Pi}^\top\|_F = \sqrt{K(p^2 + (1-p)^2)} \geq \sqrt{2K}/2$, and $(ii)$ is by Lemma C.7. Therefore, we can obtain that

$$\left\|\frac{\boldsymbol{V}^{(T)}}{\|\boldsymbol{V}^{(T)}\|_F} - \frac{\boldsymbol{\Pi}^\top}{\|\boldsymbol{\Pi}^\top\|_F}\right\|_F \leq \mathcal{O}\left(\frac{1}{\sqrt{T}}\right).$$

$\square$

**Lemma C.9.** Assume the same conditions as Theorem 3.1. For $\Omega(\eta\epsilon^{-2}K^{-2}) \leq T \leq T^*$, it holds that

$$\left\|\frac{f_{\theta^{(T)}}(\boldsymbol{X})}{\|f_{\theta^{(T)}}(\boldsymbol{X})\|_2} - \boldsymbol{\Pi}^\top \boldsymbol{x}_{N-1}\right\|_2 \leq \mathcal{O}\left(\frac{1}{\sqrt{T}}\right).$$

***Proof of Lemma C.9.*** The output with $\theta = \theta^{(T)}$ is $f_{\theta^{(T)}}(\boldsymbol{X}) = \boldsymbol{V}^{(T)}\boldsymbol{X}\mathcal{S}(\widetilde{\boldsymbol{X}}^\top \boldsymbol{W}^{(T)}\widetilde{\boldsymbol{x}}_N) = \boldsymbol{V}^{(T)}\sum_{i=1}^{N-1}\mathcal{S}_i^{(T)}\boldsymbol{x}_i$. Then, we can get that

$$\left\|\frac{f_{\theta^{(T)}}(\boldsymbol{X})}{\|f_{\theta^{(T)}}(\boldsymbol{X})\|_2} - \boldsymbol{\Pi}^\top \boldsymbol{x}_{N-1}\right\|_2$$

$$= \left\| \frac{\left(\beta^{(T)}\mathbf{\Pi}^\top + \widetilde{\mathbf{V}}^{(T)}\right)\sum_{i=1}^{N-1}\mathcal{S}_i^{(T)}\boldsymbol{x}_i}{\left\|\mathbf{V}^{(T)}\sum_{i=1}^{N-1}\mathcal{S}_i^{(T)}\boldsymbol{x}_i\right\|_2} - \mathbf{\Pi}^\top\boldsymbol{x}_{N-1} \right\|_2$$

$$\le \left\| \left( \frac{\beta^{(T)}\mathcal{S}_{N-1}^{(T)}}{\left\|\mathbf{V}^{(T)}\sum_{i=1}^{N-1}\mathcal{S}_i^{(T)}\boldsymbol{x}_i\right\|_2} - 1 \right) \mathbf{\Pi}^\top\boldsymbol{x}_{N-1} \right\|_2$$

$$+ \left\| \frac{\beta^{(T)}\mathbf{\Pi}^\top\sum_{i=1}^{N-2}\mathcal{S}_i^{(T)}\boldsymbol{x}_i + \widetilde{\mathbf{V}}^{(T)}\sum_{i=1}^{N-1}\mathcal{S}_i^{(T)}\boldsymbol{x}_i}{\left\|\mathbf{V}^{(T)}\sum_{i=1}^{N-1}\mathcal{S}_i^{(T)}\boldsymbol{x}_i\right\|_2} \right\|_2.$$

For the first part, we have

$$\left\| \left( \frac{\beta^{(T)}\mathcal{S}_{N-1}^{(T)}}{\left\|\mathbf{V}^{(T)}\sum_{i=1}^{N-1}\mathcal{S}_i^{(T)}\boldsymbol{x}_i\right\|_2} - 1 \right) \mathbf{\Pi}^\top\boldsymbol{x}_{N-1} \right\|_2$$

$$\le \left\| \left( 1 - \frac{\beta^{(T)}\mathcal{S}_{N-1}^{(T)}}{\left\|\beta^{(T)}\mathbf{\Pi}^\top\sum_{i=1}^{N-1}\mathcal{S}_i^{(T)}\boldsymbol{x}_i\right\|_2 + \left\|\widetilde{\mathbf{V}}^{(T)}\sum_{i=1}^{N-1}\mathcal{S}_i^{(T)}\boldsymbol{x}_i\right\|_2} \right) \mathbf{\Pi}^\top\boldsymbol{x}_{N-1} \right\|_2$$

$$\le \left\| \left( 1 - \frac{\beta^{(T)}[1 - \exp(-\Omega(N))]}{\frac{\sqrt{2}}{2}\beta^{(T)} + \sqrt{K}\gamma^{(T)}} \right) \mathbf{\Pi}^\top\boldsymbol{x}_{N-1} \right\|_2$$

$$\le \left( 1 - \frac{\left(\sqrt{\eta T} - \frac{2\eta}{\epsilon K}\right)[1 - \exp(-\Omega(N))]}{\left(\sqrt{\eta T} - \frac{2\eta}{\epsilon K}\right) + \sqrt{K}\left(\frac{2\eta}{\epsilon K} + 2T\epsilon K^2 N \exp(-\Omega(N))\right)} \right) \cdot \sqrt{p^2 + (1-p)^2}$$

$$\le \mathcal{O}\left( \frac{\sqrt{K}\left(\frac{2\eta}{\epsilon K} + 2T\epsilon K^2 N \exp(-\Omega(N))\right)}{\sqrt{\eta T} + 2T\epsilon K^2 N \exp(-\Omega(N))} \right)$$

$$\le \mathcal{O}\left( \frac{1}{\sqrt{T}} \right),$$

where the first inequality is by Lemma C.7, the second inequality is by Lemma C.7 and $\|\mathbf{\Pi}^\top\boldsymbol{x}_i\|_2 = \sqrt{p^2 + (1-p)^2}$, and the third inequality is by Lemma C.7. For the second part, we have

$$\left\| \frac{\beta^{(T)}\mathbf{\Pi}^\top\sum_{i=1}^{N-2}\mathcal{S}_i^{(T)}\boldsymbol{x}_i + \widetilde{\mathbf{V}}^{(T)}\sum_{i=1}^{N-1}\mathcal{S}_i^{(T)}\boldsymbol{x}_i}{\left\|\mathbf{V}^{(T)}\sum_{i=1}^{N-1}\mathcal{S}_i^{(T)}\boldsymbol{x}_i\right\|_2} \right\|_2$$

$$\le \frac{\left\|\beta^{(T)}\mathbf{\Pi}^\top\sum_{i=1}^{N-2}\mathcal{S}_i^{(T)}\boldsymbol{x}_i\right\|_2 + \left\|\widetilde{\mathbf{V}}^{(T)}\sum_{i=1}^{N-1}\mathcal{S}_i^{(T)}\boldsymbol{x}_i\right\|_2}{\left\|\beta^{(T)}\mathbf{\Pi}^\top\sum_{i=1}^{N-1}\mathcal{S}_i^{(T)}\boldsymbol{x}_i\right\|_2}$$

$$\le \frac{\exp(-\Omega(N))\|\mathbf{V}^{(T)}\|_{\max} + \sqrt{K}\gamma^{(T)}}{\sqrt{p^2 + (1-p)^2}\beta^{(T)}}$$

$$\le \frac{\exp(-\Omega(N))\left(\frac{\eta}{\epsilon K} + 2T\epsilon K^2\right) + \sqrt{K}\left(\frac{2\eta}{\epsilon K} + 2T\epsilon K^2 N \exp(-\Omega(N))\right)}{\sqrt{p^2 + (1-p)^2} \cdot \left(\sqrt{\eta T} - \frac{2\eta}{\epsilon K}\right)}$$

$$\le \mathcal{O}\left( \frac{1}{\sqrt{T}} \right)$$

where the second inequality is by Lemma C.7 and $\|\mathbf{\Pi}^\top\boldsymbol{x}_i\|_2 = \sqrt{p^2 + (1-p)^2}$, and the third inequality is by Lemma C.7. Therefore, we can obtain that

$$\left\| \frac{f_{\theta^{(T)}}(\boldsymbol{X})}{\|f_{\theta^{(T)}}(\boldsymbol{X})\|_2} - \mathbf{\Pi}^\top\boldsymbol{x}_{N-1} \right\|_2 \le \mathcal{O}\left( \frac{1}{\sqrt{T}} \right).$$

$\square$

# D Proof of Theorem 4.1

In this section, we analyze "deterministic walks" with $p = 0, 1$. Without loss of generality, we set $p = 1$, which means $\mathbf{\Pi} = \mathbf{\Pi}_0^\top$. The following lemma shows the results of the first iteration.

**Lemma D.1.** Under the same condition as Theorem 4.1, for any loss function $\ell(\cdot)$, it holds that

$$\boldsymbol{V}^{(1)} = -\ell'(\theta^{(0)}) \cdot \frac{\eta r}{NK} \mathbf{1}_{K \times K} \text{ and } \boldsymbol{W}^{(1)} = \mathbf{0}_{(K+M) \times (K+M)}.$$

*Proof of Lemma D.1.* By Lemma B.1, we have

$$\mathbb{E}[\nabla_{\boldsymbol{V}} \ell(\theta^{(0)})] = \ell'(\theta^{(0)}) \cdot \frac{1}{N} \sum_{i=1}^{N-1} \mathbb{E}[\boldsymbol{e}_y \boldsymbol{x}_i^\top]$$

$$= \ell'(\theta^{(0)}) \cdot \frac{1}{N} \sum_{i=1}^{N-1} \mathbb{E}[(\mathbf{\Pi}^\top)^{N-i} \boldsymbol{x}_i \boldsymbol{x}_i^\top]$$

$$= \ell'(\theta^{(0)}) \cdot \frac{1}{NK} \sum_{i=1}^{N-1} (\mathbf{\Pi}_0)^{N-i}$$

$$= \ell'(\theta^{(0)}) \cdot \frac{r}{NK} \mathbf{1}_{K \times K},$$

where the first equation is by the initialization of $\boldsymbol{V}^{(0)}$ and $\boldsymbol{W}^{(0)}$, the second equation is by the sampling method, the third equation is by $\mathbb{E}[\boldsymbol{x}_i \boldsymbol{x}_i^\top] = \frac{1}{K} \mathbf{I}_K$ for $i \in [N-1]$, and the last equation is by Lemma E.1. Thus, by the update, we can get

$$\boldsymbol{V}^{(1)} = \boldsymbol{V}^{(0)} - \eta \mathbb{E}[\nabla_{\boldsymbol{V}} \ell(\theta^{(0)})] = -\ell'(\theta^{(0)}) \cdot \frac{\eta r}{NK} \mathbf{1}_{K \times K}.$$

Since $\boldsymbol{V}^{(0)} = \mathbf{0}_{K \times K}$ and $\boldsymbol{W}^{(0)} = \mathbf{0}_{(K+M) \times (K+M)}$, we can get $\mathbb{E}[\nabla_{\boldsymbol{W}} \ell(\theta^{(0)})] = \mathbf{0}_{(K+M) \times (K+M)}$. Thus,

$$\boldsymbol{W}^{(1)} = \boldsymbol{W}^{(0)} - \eta \mathbb{E}[\nabla_{\boldsymbol{W}} \ell(\theta^{(0)})] = \mathbf{0}_{(K+M) \times (K+M)}.$$

$\square$

The following lemma states the results of the second iteration.

**Lemma D.2.** If $\mathbf{\Pi} = \mathbf{\Pi}_2$, then it holds that

$$\boldsymbol{V}^{(2)} = -(\ell'(\theta^{(0)}) + \ell'(\theta^{(1)})) \cdot \frac{\eta r}{NK} \mathbf{1}_{K \times K},$$

$$\boldsymbol{W}_{12}^{(2)} = \ell'(\theta^{(1)}) \ell'(\theta^{(0)}) \frac{\eta^2 r^2}{N^3 K} \mathbf{1}_K \boldsymbol{p}_N^\top,$$

$$\boldsymbol{W}_{22}^{(2)} = \ell'(\theta^{(1)}) \ell'(\theta^{(0)}) \left( \frac{\eta^2 r}{N^3 K} \sum_{i=1}^{N-1} \boldsymbol{p}_i - \frac{\eta^2 r^2}{N^3} \boldsymbol{p}_N \right) \boldsymbol{p}_N^\top.$$

*Proof of Lemma D.2.* By Lemma B.1, we have

$$\mathbb{E}[\nabla_{\boldsymbol{V}} \ell(\theta^{(1)})] = \ell'(\theta^{(1)}) \cdot \frac{1}{N} \sum_{i=1}^{N-1} \mathbb{E}[\boldsymbol{e}_y \boldsymbol{x}_i^\top]$$

$$= \ell'(\theta^{(1)}) \cdot \frac{1}{N} \sum_{i=1}^{N-1} \mathbb{E}[(\mathbf{\Pi}^\top)^{N-i} \boldsymbol{x}_i \boldsymbol{x}_i^\top]$$

$$= \ell'(\theta^{(1)}) \cdot \frac{1}{NK} \sum_{i=1}^{N-1} (\mathbf{\Pi}_0)^{N-i}$$

$$= \ell'(\theta^{(1)}) \cdot \frac{r}{NK} \mathbf{1}_{K \times K},$$

where the second equation is by the sampling method, the third equation is by $\mathbb{E}[\boldsymbol{x}_i \boldsymbol{x}_i^\top] = \frac{1}{K}\mathbf{I}_K$ for $i \in [N-1]$, and the last equation is by Lemma E.1. Thus, we can get

$$\boldsymbol{V}^{(2)} = \boldsymbol{V}^{(1)} - \eta \mathbb{E}[\nabla_{\boldsymbol{V}} \ell(\theta^{(1)})]$$
$$= -\ell'(\theta^{(0)}) \cdot \frac{\eta r}{NK} \mathbf{1}_{K \times K} - \ell'(\theta^{(1)}) \cdot \frac{\eta r}{NK} \mathbf{1}_{K \times K}.$$

By Lemma B.1, we have

$$\mathbb{E}[\boldsymbol{A}^{(1)}] = \mathbb{E}\left[ \left( \sum_{i=1}^{N-1} \mathcal{S}_i^{(1)} \boldsymbol{x}_i \boldsymbol{x}_i^\top (\boldsymbol{V}^{(1)})^\top \boldsymbol{e}_y - \sum_{i_1=1}^{N-1} \sum_{i_2=1}^{N-1} \mathcal{S}_{i_1}^{(1)} \mathcal{S}_{i_2}^{(1)} \boldsymbol{x}_{i_1} \boldsymbol{x}_{i_2}^\top (\boldsymbol{V}^{(1)})^\top \boldsymbol{e}_y \right) \boldsymbol{p}_N^\top \right]$$

$$= \mathbb{E}\left[ -\ell'(\theta^{(0)}) \left( \frac{\eta r}{N^2 K} \sum_{i=1}^{N-1} \boldsymbol{x}_i \boldsymbol{x}_i^\top \mathbf{1}_K - \frac{\eta r}{N^3 K} \sum_{i_1=1}^{N-1} \sum_{i_2=1}^{N-1} \boldsymbol{x}_{i_1} \boldsymbol{x}_{i_2}^\top \mathbf{1}_K \right) \boldsymbol{p}_N^\top \right]$$

$$= \mathbb{E}\left[ -\ell'(\theta^{(0)}) \left( \frac{\eta r}{N^2 K} \sum_{i=1}^{N-1} \boldsymbol{x}_i - \frac{\eta r}{N^3 K} \sum_{i_1=1}^{N-1} \sum_{i_2=1}^{N-1} \boldsymbol{x}_{i_1} \right) \boldsymbol{p}_N^\top \right]$$

$$= -\ell'(\theta^{(0)}) \left( \frac{\eta r^2}{N^2 K} \mathbf{1}_K - \frac{\eta r^2 (N-1)}{N^3 K} \mathbf{1}_K \right) \boldsymbol{p}_N^\top$$

$$= -\ell'(\theta^{(0)}) \frac{\eta r^2}{N^3 K} \mathbf{1}_K \boldsymbol{p}_N^\top,$$

where the second equation is by Lemma D.1, and the fourth equation is by the fact that all the $\boldsymbol{x}_i$ is uniformly distributed in $\boldsymbol{E}$. We also have

$$\mathbb{E}[\boldsymbol{B}^{(1)}] = \mathbb{E}\left[ \left( \sum_{i=1}^{N-1} \mathcal{S}_i^{(1)} \boldsymbol{p}_i \boldsymbol{x}_i^\top (\boldsymbol{V}^{(1)})^\top \boldsymbol{e}_y - \sum_{i=1}^{N} \mathcal{S}_i^{(1)} \boldsymbol{p}_i \cdot \sum_{i=1}^{N-1} \mathcal{S}_i^{(1)} \boldsymbol{x}_i^\top (\boldsymbol{V}^{(1)})^\top \boldsymbol{e}_y \right) \boldsymbol{p}_N^\top \right]$$

$$= \mathbb{E}\left[ -\ell'(\theta^{(0)}) \left( \frac{\eta r}{N^2 K} \sum_{i=1}^{N-1} \boldsymbol{p}_i \boldsymbol{x}_i^\top \mathbf{1}_K - \frac{\eta r}{N^3 K} \sum_{i=1}^{N} \boldsymbol{p}_i \cdot \sum_{i=1}^{N-1} \boldsymbol{x}_i^\top \mathbf{1}_K \right) \boldsymbol{p}_N^\top \right]$$

$$= -\ell'(\theta^{(0)}) \left( \frac{\eta r}{N^2 K} \sum_{i=1}^{N-1} \boldsymbol{p}_i - \frac{\eta r (N-1)}{N^3 K} \sum_{i=1}^{N} \boldsymbol{p}_i \right) \boldsymbol{p}_N^\top$$

$$= -\ell'(\theta^{(0)}) \left( \frac{\eta r}{N^3 K} \sum_{i=1}^{N-1} \boldsymbol{p}_i - \frac{\eta r^2}{N^3} \boldsymbol{p}_N \right) \boldsymbol{p}_N^\top,$$

where the second equation is by Lemma D.1. Thus, we can get that

$$\boldsymbol{W}_{12}^{(2)} = \boldsymbol{W}_{12}^{(1)} - \eta \mathbb{E}[\nabla_{\boldsymbol{W}} \ell(\theta^{(1)})]_{12}$$
$$= -\mathbb{E}\left[ \eta \ell'(\theta^{(1)}) \cdot \boldsymbol{A}^{(1)} \right]$$
$$= \ell'(\theta^{(1)}) \ell'(\theta^{(0)}) \frac{\eta^2 r^2}{N^3 K} \mathbf{1}_K \boldsymbol{p}_N^\top,$$

and

$$\boldsymbol{W}_{22}^{(2)} = \boldsymbol{W}_{22}^{(1)} - \eta \mathbb{E}[\nabla_{\boldsymbol{W}} \ell(\theta^{(1)})]_{22}$$
$$= -\mathbb{E}\left[ \eta \ell'(\theta^{(1)}) \cdot \boldsymbol{B}^{(1)} \right]$$
$$= \ell'(\theta^{(1)}) \ell'(\theta^{(0)}) \left( \frac{\eta^2 r}{N^3 K} \sum_{i=1}^{N-1} \boldsymbol{p}_i - \frac{\eta^2 r^2}{N^3} \boldsymbol{p}_N \right) \boldsymbol{p}_N^\top.$$

$\square$

Next, we can analyze the gradient descent dynamics over multiple iterations.

**Lemma D.3.** If $\mathbf{\Pi} = \mathbf{\Pi}_2$, then for any $t \geq 0$ and any sequence of learning rates $\{\eta_t\}$, it holds that

$$\boldsymbol{V}^{(t)} \propto \mathbf{1}_{K \times K}, \quad \text{and} \quad \mathcal{S}_1^{(t)} = \mathcal{S}_2^{(t)} = \cdots = \mathcal{S}_{N-1}^{(t)}.$$

***Proof of Lemma D.3.*** We use induction to prove that for some scalar $\alpha_1^{(t)}, \alpha_2^{(t)}, \alpha_3^{(t)}, \alpha_4^{(t)}$, it holds that for $t \geq 2$, $\boldsymbol{V}^{(t)} = \alpha_1^{(t)} \mathbf{1}_{K \times K}$, $\boldsymbol{W}_{12}^{(t)} = \alpha_2^{(t)} \mathbf{1}_K \boldsymbol{p}_N^\top$, and $\boldsymbol{W}_{22}^{(t)} = \left(\alpha_3^{(t)} \sum_{i=1}^{N-1} \boldsymbol{p}_i - \alpha_4^{(t)} \boldsymbol{p}_N\right) \boldsymbol{p}_N^\top$. By Lemma D.2, we know that the hypothesis holds for $t = 2$. Suppose that the hypothesis holds for $t = t'$. We aim to prove that the hypothesis holds for $t = t' + 1$. We have

$$
\widetilde{\boldsymbol{X}} \boldsymbol{W}^{(t')} \widetilde{\boldsymbol{x}}_N = 
\begin{bmatrix}
\boldsymbol{x}_1^\top \boldsymbol{W}_{12}^{(t')} \boldsymbol{p}_N + \boldsymbol{p}_1^\top \boldsymbol{W}_{22}^{(t')} \boldsymbol{p}_N \\
\boldsymbol{x}_2^\top \boldsymbol{W}_{12}^{(t')} \boldsymbol{p}_N + \boldsymbol{p}_2^\top \boldsymbol{W}_{22}^{(t')} \boldsymbol{p}_N \\
\vdots \\
\boldsymbol{x}_{N-1}^\top \boldsymbol{W}_{12}^{(t')} \boldsymbol{p}_N + \boldsymbol{p}_{N-1}^\top \boldsymbol{W}_{22}^{(t')} \boldsymbol{p}_N \\
\boldsymbol{p}_N^\top \boldsymbol{W}_{22}^{(t')} \boldsymbol{p}_N
\end{bmatrix}
$$

$$
= 
\begin{bmatrix}
\alpha_2^{(t')} \boldsymbol{p}_N^\top \boldsymbol{p}_N + \alpha_3^{(t')} \boldsymbol{p}_1^\top \boldsymbol{p}_1 \boldsymbol{p}_N^\top \boldsymbol{p}_N \\
\alpha_2^{(t')} \boldsymbol{p}_N^\top \boldsymbol{p}_N + \alpha_3^{(t')} \boldsymbol{p}_2^\top \boldsymbol{p}_2 \boldsymbol{p}_N^\top \boldsymbol{p}_N \\
\vdots \\
\alpha_2^{(t')} \boldsymbol{p}_N^\top \boldsymbol{p}_N + \alpha_3^{(t')} \boldsymbol{p}_{N-1}^\top \boldsymbol{p}_{N-1} \boldsymbol{p}_N^\top \boldsymbol{p}_N \\
-\alpha_4^{(t')} (\boldsymbol{p}_N^\top \boldsymbol{p}_N)^2
\end{bmatrix}.
\tag{D.1}
$$

Since $\boldsymbol{p}_1^\top \boldsymbol{p}_1 = \boldsymbol{p}_2^\top \boldsymbol{p}_2 = \cdots = \boldsymbol{p}_N^\top \boldsymbol{p}_N$, we have $[\widetilde{\boldsymbol{X}} \boldsymbol{W}^{(t')} \widetilde{\boldsymbol{x}}_N]_1 = [\widetilde{\boldsymbol{X}} \boldsymbol{W}^{(t')} \widetilde{\boldsymbol{x}}_N]_2 = \cdots = [\widetilde{\boldsymbol{X}} \boldsymbol{W}^{(t')} \widetilde{\boldsymbol{x}}_N]_{N-1}$. Thus, we can get that $\mathcal{S}_1^{(t')} = \mathcal{S}_2^{(t')} = \cdots = \mathcal{S}_{N-1}^{(t')} := s^{(t')}$. Then, we have

$$
\mathbb{E}[\nabla_{\boldsymbol{V}} \ell(\theta^{(t')})] = \mathbb{E}\left[\ell'(\theta^{(t')}) \cdot \boldsymbol{e}_y \sum_{i=1}^{N-1} \mathcal{S}_i^{(t')} \boldsymbol{x}_i^\top\right]
$$

$$
= \ell'(\theta^{(t')}) s^{(t')} \sum_{i=1}^{N-1} \mathbb{E}[\boldsymbol{e}_y \boldsymbol{x}_i^\top]
$$

$$
= \ell'(\theta^{(t')}) s^{(t')} \sum_{i=1}^{N-1} \mathbb{E}[(\mathbf{\Pi}^\top)^{N-i} \boldsymbol{x}_i \boldsymbol{x}_i^\top]
$$

$$
= \ell'(\theta^{(t')}) \frac{s^{(t')}}{K} \sum_{i=1}^{N-1} (\mathbf{\Pi}_0)^{N-i}
$$

$$
= \ell'(\theta^{(t')}) \frac{s^{(t')} r}{K} \mathbf{1}_{K \times K},
$$

where the second equation is by the induction, the third equation is by the sampling method, the fourth equation is by the fact that $\boldsymbol{x}_i$ is uniformly distributed in $\boldsymbol{E}$, and the last equation is by Lemma E.1. Thus, we can get $\boldsymbol{V}^{(t'+1)} = \boldsymbol{V}^{(t')} - \eta^{(t')} \mathbb{E}[\nabla_{\boldsymbol{V}} \ell(\theta^{(t')})] \propto \mathbf{1}_{K \times K}$. We also have

$$
\mathbb{E}[\boldsymbol{A}^{(t')}] = \mathbb{E}\left[\left(\sum_{i=1}^{N-1} \mathcal{S}_i^{(t')} \boldsymbol{x}_i \boldsymbol{x}_i^\top (\boldsymbol{V}^{(t')})^\top \boldsymbol{e}_y - \sum_{i_1=1}^{N-1} \sum_{i_2=1}^{N-1} \mathcal{S}_{i_1}^{(t')} \mathcal{S}_{i_2}^{(t')} \boldsymbol{x}_{i_1} \boldsymbol{x}_{i_2}^\top (\boldsymbol{V}^{(t')})^\top \boldsymbol{e}_y\right) \boldsymbol{p}_N^\top\right]
$$

$$
= \mathbb{E}\left[\left(\alpha_1^{(t')} s^{(t')} \sum_{i=1}^{N-1} \boldsymbol{x}_i \boldsymbol{x}_i^\top \mathbf{1}_K - \alpha_1^{(t')} (s^{(t')})^2 \sum_{i_1=1}^{N-1} \sum_{i_2=1}^{N-1} \boldsymbol{x}_{i_1} \boldsymbol{x}_{i_2}^\top \mathbf{1}_K\right) \boldsymbol{p}_N^\top\right]
$$

$$
= \mathbb{E}\left[\left(\alpha_1^{(t')} s^{(t')} \sum_{i=1}^{N-1} \boldsymbol{x}_i - \alpha_1^{(t')} (s^{(t')})^2 \sum_{i_1=1}^{N-1} \sum_{i_2=1}^{N-1} \boldsymbol{x}_{i_1}\right) \boldsymbol{p}_N^\top\right]
$$

$$
= \left(\frac{\alpha_1^{(t')} s^{(t')} (N-1)}{K} \mathbf{1}_K - \frac{\alpha_1^{(t')} (s^{(t')})^2 (N-1)^2}{K} \mathbf{1}_K\right) \boldsymbol{p}_N^\top,
$$

where the second equation is by the induction, and the fourth equation is by the fact that all the $x_i$ is uniformly distributed in $E$. And,

$$
\begin{aligned}
\mathbb{E}[\boldsymbol{B}^{(t')}] &= \mathbb{E}\left[\left(\sum_{i=1}^{N-1}\mathcal{S}_i^{(t')}\boldsymbol{p}_i\boldsymbol{x}_i^\top(\boldsymbol{V}^{(t')})^\top\boldsymbol{e}_y - \sum_{i=1}^{N}\mathcal{S}_i^{(t')}\boldsymbol{p}_i\cdot\sum_{i=1}^{N-1}\mathcal{S}_i^{(t')}\boldsymbol{x}_i^\top(\boldsymbol{V}^{(t')})^\top\boldsymbol{e}_y\right)\boldsymbol{p}_N^\top\right] \\
&= \mathbb{E}\left[\left(\alpha_1^{(t')}s^{(t')}\sum_{i=1}^{N-1}\boldsymbol{p}_i\boldsymbol{x}_i^\top\mathbf{1}_K - \alpha_1^{(t')}(s^{(t')})^2\sum_{i=1}^{N}\boldsymbol{p}_i\cdot\sum_{i=1}^{N-1}\boldsymbol{x}_i^\top\mathbf{1}_K\right)\boldsymbol{p}_N^\top\right] \\
&= \left(\alpha_1^{(t')}s^{(t')}\sum_{i=1}^{N-1}\boldsymbol{p}_i - \alpha_1^{(t')}(s^{(t')})^2(N-1)\sum_{i=1}^{N}\boldsymbol{p}_i\right)\boldsymbol{p}_N^\top,
\end{aligned}
$$

where the second equation is by the induction. Therefore, we can get

$$
\begin{aligned}
\boldsymbol{W}_{12}^{(t'+1)} &= \boldsymbol{W}_{12}^{(t')} - \eta\mathbb{E}[\nabla_{\boldsymbol{W}}\ell(\theta^{(t')})]_{12} \\
&= \alpha_2^{(t')}\mathbf{1}_K\boldsymbol{p}_N^\top - \eta\ell'(\theta^{(t')})\mathbb{E}[\boldsymbol{A}^{(t')}] \\
&= \alpha_2^{(t')}\mathbf{1}_K\boldsymbol{p}_N^\top - \eta\ell'(\theta^{(t')})\left(\frac{\alpha_1^{(t')}s^{(t')}(N-1)}{K} - \frac{\alpha_1^{(t')}(s^{(t')})^2(N-1)^2}{K}\right)\mathbf{1}_K\boldsymbol{p}_N^\top \\
&:= \alpha_2^{(t'+1)}\mathbf{1}_K\boldsymbol{p}_N^\top,
\end{aligned}
$$

and

$$
\begin{aligned}
\boldsymbol{W}_{22}^{(t'+1)} &= \boldsymbol{W}_{22}^{(t')} - \eta\mathbb{E}[\nabla_{\boldsymbol{W}}\ell(\theta^{(t')})]_{22} \\
&= \left(\alpha_3^{(t')}\sum_{i=1}^{N-1}\boldsymbol{p}_i - \alpha_4^{(t')}\boldsymbol{p}_N\right)\boldsymbol{p}_N^\top - \eta\ell'(\theta^{(t')})\mathbb{E}[\boldsymbol{B}^{(t')}] \\
&= \left(\alpha_3^{(t')}\sum_{i=1}^{N-1}\boldsymbol{p}_i - \alpha_4^{(t')}\boldsymbol{p}_N\right)\boldsymbol{p}_N^\top \\
&\quad - \eta\ell'(\theta^{(t')})\left(\alpha_1^{(t')}s^{(t')}\sum_{i=1}^{N-1}\boldsymbol{p}_i - \alpha_1^{(t')}(s^{(t')})^2(N-1)\sum_{i=1}^{N}\boldsymbol{p}_i\right)\boldsymbol{p}_N^\top \\
&:= \left(\alpha_3^{(t'+1)}\sum_{i=1}^{N-1}\boldsymbol{p}_i - \alpha_4^{(t'+1)}\boldsymbol{p}_N\right)\boldsymbol{p}_N^\top.
\end{aligned}
$$

Therefore, by induction, we can conclude that for all $t \geq 2$, $\boldsymbol{V}^{(t)} = \alpha_1^{(t)}\mathbf{1}_{K\times K}$, $\boldsymbol{W}_{12}^{(t)} = \alpha_2^{(t)}\mathbf{1}_K\boldsymbol{p}_N^\top$, and $\boldsymbol{W}_{22}^{(t)} = \left(\alpha_3^{(t)}\sum_{i=1}^{N-1}\boldsymbol{p}_i - \alpha_4^{(t)}\boldsymbol{p}_N\right)\boldsymbol{p}_N^\top$. Similar to (D.1), we have $[\widetilde{\boldsymbol{X}}\boldsymbol{W}^{(t)}\widetilde{\boldsymbol{x}}_N]_1 = [\widetilde{\boldsymbol{X}}\boldsymbol{W}^{(t)}\widetilde{\boldsymbol{x}}_N]_2 = \cdots = [\widetilde{\boldsymbol{X}}\boldsymbol{W}^{(t)}\widetilde{\boldsymbol{x}}_N]_{N-1}$, which implies that $\mathcal{S}_1^{(t)} = \mathcal{S}_2^{(t)} = \cdots = \mathcal{S}_{N-1}^{(t)}$. $\qquad\square$

## E   Auxiliary Lemmas

In this section, we present some auxiliary lemmas. The following lemma states the properties of $\boldsymbol{\Pi}_0$.

**Lemma E.1.** By the definition of $\boldsymbol{\Pi}_0$, it holds that $\boldsymbol{\Pi}_0^K = \boldsymbol{I}_K$, $\boldsymbol{\Pi}_0\boldsymbol{\Pi}_0^\top = \boldsymbol{I}_K$, and $\sum_{k=1}^{K}\boldsymbol{\Pi}_0^k = \mathbf{1}_{K\times K}$.

***Proof of Lemma E.1.*** In this proof, the index $i$ larger than $K$ represents $i - K$. For $\boldsymbol{\Pi}_0$, only $[\boldsymbol{\Pi}_0]_{i+1,i} = 1$ for $i \in [K]$ and other elements are 0. We can get that for $\boldsymbol{\Pi}_0^k$, only $[\boldsymbol{\Pi}_0^k]_{i+k,i} = 1$ for $i \in [K]$ and other elements are 0. By this observation, we can derive that $\boldsymbol{\Pi}_0^K = \boldsymbol{I}_K$ and $\sum_{k=1}^{K}\boldsymbol{\Pi}_0^k = \mathbf{1}_{K\times K}$. Also, we have $\boldsymbol{\Pi}_0^\top = \boldsymbol{\Pi}_0^{K-1}$, so we can get $\boldsymbol{\Pi}_0\boldsymbol{\Pi}_0^\top = \boldsymbol{\Pi}_0^K = \boldsymbol{I}_K$. $\qquad\square$

Further, the following three lemmas show some properties of $\boldsymbol{\Pi}$ with transition probability $p$ satisfying $0 < p < 1$.

**Lemma E.2.** Assume that $K$ is odd. It holds that

$$\left| \left[ \mathbf{\Pi}^R \right]_{i,j} - \frac{1}{K} \right| \leq \exp\left( -\frac{8p(1-p)R}{K^2} \right).$$

***Proof of Lemma E.2.*** $\mathbf{\Pi}$ has eigenvalues $\lambda_0, ..., \lambda_{K-1}$, where

$$\lambda_k = pe^{-\frac{2\pi \mathrm{i} k}{K}} + (1-p)e^{\frac{2\pi \mathrm{i} k}{K}} = \cos\left( \frac{2\pi k}{K} \right) + \mathrm{i}(1-2p)\sin\left( \frac{2\pi k}{K} \right)$$

with

$$|\lambda_k| = \sqrt{1 - 4p(1-p)\sin^2\left( \frac{2\pi k}{K} \right)} \leq \sqrt{1 - \frac{16p(1-p)}{K^2}} \leq 1 - \frac{8p(1-p)}{K^2}$$

for $k \neq 0$, where the first inequality is by $\sin(2\pi k/K) \geq \sin(\pi/K) \geq 2/K$. The eigendecomposition of each entry in $\mathbf{\Pi}$ can be written as

$$[\mathbf{\Pi}]_{i,j} = \sum_{k=0}^{K-1} c_{k,i,j} \lambda_k = \frac{1}{K}\lambda_0 + \sum_{k \neq 0} c_{k,i,j} \lambda_k,$$

where $c_{k,i,j} = \frac{1}{\sqrt{K}}e^{2\pi \mathrm{i}(i-1)k/K} \cdot \frac{1}{\sqrt{K}}e^{2\pi \mathrm{i}(j-1)k/K} = \frac{1}{K}e^{2\pi \mathrm{i}(i+j-2)k/K}$. Then, we can get that

$$[\mathbf{\Pi}^R]_{i,j} = \sum_{k=0}^{K-1} c_{k,i,j} \lambda_k^R = \frac{1}{K} + \sum_{k \neq 0} c_{k,i,j} \lambda_k^R$$

Thus,

$$\begin{aligned}
\left| [\mathbf{\Pi}^R]_{i,j} - \frac{1}{K} \right| &= \left| \sum_{k \neq 0} c_{k,i,j} \lambda_k^R \right| \\
&\leq \sum_{k \neq 0} |c_{k,i,j}||\lambda_k|^R \\
&\leq \left( 1 - \frac{8p(1-p)}{K^2} \right)^R \sum_{k \neq 0} |c_{k,i,j}| \\
&\leq \exp\left( -\frac{8p(1-p)R}{K^2} \right),
\end{aligned}$$

where the second inequality is by the bound of the absolute values of the eigenvalues, and the last inequality is by $(1-t)^R \leq e^{-Rt}$ for any $0 < t < 1$ and $|c_{k,i,j}| = \frac{1}{K}$ for all $k, i, j$. $\qquad \square$

**Lemma E.3.** Assume that $K$ is even. For the case that $R$ is even,

$$\left[ \mathbf{\Pi}^R \right]_{i,j} = 0 \text{ for odd } (j - i);$$

$$\left| \left[ \mathbf{\Pi}^R \right]_{i,j} - \frac{2}{K} \right| \leq \exp\left( -\frac{8p(1-p)R}{K^2} \right) \text{ for even } (j - i).$$

For the case that $R$ is odd,

$$\left| \left[ \mathbf{\Pi}^R \right]_{i,j} - \frac{2}{K} \right| \leq \exp\left( -\frac{8p(1-p)R}{K^2} \right) \text{ for odd } (j - i);$$

$$\left[ \mathbf{\Pi}^R \right]_{i,j} = 0 \text{ for even } (j - i).$$

***Proof of Lemma E.3***. $\mathbf{\Pi}$ has eigenvalues $\lambda_0, ..., \lambda_{K-1}$, where

$$\lambda_k = pe^{-\frac{2\pi \mathrm{i} k}{K}} + (1-p)e^{\frac{2\pi \mathrm{i} k}{K}} = \cos\left( \frac{2\pi k}{K} \right) + \mathrm{i}(1-2p)\sin\left( \frac{2\pi k}{K} \right)$$

with

$$|\lambda_k| = \sqrt{1 - 4p(1-p)\sin^2\left(\frac{2\pi k}{K}\right)} \le \sqrt{1 - \frac{16p(1-p)}{K^2}} \le 1 - \frac{8p(1-p)}{K^2}.$$

for $k \ne 0, K/2$, where the first inequality is by $\sin(2\pi k/K) \ge \sin(\pi/K) \ge 2/K$. The eigendecomposition of each entry in $\mathbf{\Pi}$ can be written as

$$[\mathbf{\Pi}]_{i,j} = \sum_{k=0}^{K-1} c_{k,i,j}\lambda_k = \frac{1}{K}\lambda_0 + \frac{(-1)^{i+j}}{K}\lambda_{K/2} + \sum_{k \ne 0, K/2} c_{k,i,j}\lambda_k,$$

where $c_{k,i,j} = \frac{1}{\sqrt{K}}e^{2\pi i(i-1)k/K} \cdot \frac{1}{\sqrt{K}}e^{2\pi i(j-1)k/K} = \frac{1}{K}e^{2\pi i(i+j-2)k/K}$. Then, we can get that

$$[\mathbf{\Pi}^R]_{i,j} = \sum_{k=0}^{K-1} c_{k,i,j}\lambda_k^R = \frac{1}{K}\lambda_0^R + \frac{(-1)^{i+j}}{K}\lambda_{K/2}^R + \sum_{k \ne 0, K/2} c_{k,i,j}\lambda_k^R$$

$$= \frac{1}{K} + \frac{(-1)^{i+j+R}}{K} + \sum_{k \ne 0, K/2} c_{k,i,j}\lambda_k^R$$

When $i + j + R$ is odd, it is easy to obtain that

$$c_{t,i,j}\lambda_t^R + c_{t+\frac{K}{2},i,j}\lambda_{t+\frac{K}{2}}^R = \frac{1}{K}e^{2\pi i(i+j-2)t/K}\lambda_t^R + \frac{1}{K}(-1)^{i+j-2}e^{2\pi i(i+j-2)t/K}(-\lambda_t)^R$$

$$= \frac{1}{K}e^{2\pi i(i+j-2)t/K}\lambda_t^R + (-1)^{i+j+R} \cdot \frac{1}{K}e^{2\pi i(i+j-2)t/K}\lambda_t^R$$

$$= 0,$$

which indicates that $[\mathbf{\Pi}^R]_{i,j} = 0$.

When $i + j + R$ is even,

$$\left|[\mathbf{\Pi}^R]_{i,j} - \frac{2}{K}\right| = \left|\sum_{k \ne 0, K/2} c_{k,i,j}\lambda_k^R\right|$$

$$\le \sum_{k \ne 0, K/2} |c_{k,i,j}||\lambda_k|^R$$

$$\le \left(1 - \frac{8p(1-p)}{K^2}\right)^R \sum_{k \ne 0, K/2} |c_{k,i,j}|$$

$$\le \exp\left(-\frac{8p(1-p)R}{K^2}\right),$$

where the second inequality is by the bound of the absolute values of the eigenvalues, and the last inequality is by $(1-t)^R \le e^{-Rt}$ for any $0 < t < 1$ and $|c_{k,i,j}| = \frac{1}{K}$ for all $k, i, j$. $\square$

**Lemma E.4.** For $\mathbf{\Pi}$ with $0 < p < 1$, $N = \omega(p^{-1}(1-p)^{-1})$ and $k \in \{2, ..., N-1\}$, it holds that

$$\left[\sum_{i=1}^{N-1} \mathbf{\Pi}^i \cdot \mathbf{\Pi}^\top\right]_{1,1} \ge \left[\sum_{i=1}^{N-1} \mathbf{\Pi}^i \cdot (\mathbf{\Pi}^\top)^k\right]_{1,1} + C_p,$$

where $C_p$ is a positive constant only depending on $p$.

***Proof of Lemma E.4.*** Without loss of generality, we assume $\frac{1}{2} \le p < 1$. We observe that

$$\left[\sum_{i=1}^{N-1} \mathbf{\Pi}^i\mathbf{\Pi}^\top\right]_{1,1} = e_1^\top \sum_{i=1}^{N-1} \mathbf{\Pi}^i\mathbf{\Pi}^\top e_1 = e_1^\top \sum_{i=0}^{N-2} \mathbf{\Pi}^i \cdot \mathbf{\Pi}\mathbf{\Pi}^\top e_1,$$

$$\left[\sum_{i=1}^{N-1} \mathbf{\Pi}^i(\mathbf{\Pi}^\top)^k\right]_{1,1} = \boldsymbol{e}_1^\top \sum_{i=1}^{N-1} \mathbf{\Pi}^i(\mathbf{\Pi}^\top)^k \boldsymbol{e}_1 = \boldsymbol{e}_1^\top \sum_{i=0}^{N-2} \mathbf{\Pi}^i \cdot \mathbf{\Pi}(\mathbf{\Pi}^\top)^k \boldsymbol{e}_1.$$

Denote $\mathbf{\Gamma}(N) = \sum_{i=0}^{N-2} \mathbf{\Pi}^i$. We use induction to prove that

$$\mathbf{\Gamma}(N)_{1,1} \geq \mathbf{\Gamma}(N)_{i,j} + (1-p)^{N-2} \text{ for } i \neq j.$$

It can be easily obtained that for $N = 2$, $\mathbf{\Gamma}(2)_{1,1} = \mathbf{\Gamma}(2)_{i,j} + 1$ for any $i \neq j$. Suppose that the induction hypothesis holds for $\mathbf{\Gamma}(N-1)$. Then, for $\mathbf{\Gamma}(N)$, by the definition of $\mathbf{\Gamma}(N)$, we can get $\mathbf{\Gamma}(N) = \mathbf{\Pi} \cdot \mathbf{\Gamma}(N-1) + \mathbf{I}$. Thus, based on the definition of $\mathbf{\Pi}$, we obtain that for $i \neq j$,

$$\mathbf{\Gamma}(N)_{i,j} = p \cdot \mathbf{\Gamma}(N-1)_{i,j-1} + (1-p) \cdot \mathbf{\Gamma}(N-1)_{i,j+1}.$$

When $|i - j| \geq 2$, by induction, we have

$$\begin{aligned}
\mathbf{\Gamma}(N)_{i,j} &= p \cdot \mathbf{\Gamma}(N-1)_{i,j-1} + (1-p) \cdot \mathbf{\Gamma}(N-1)_{i,j+1} \\
&\leq p(\mathbf{\Gamma}(N-1)_{1,1} - (1-p)^{N-3}) + (1-p)(\mathbf{\Gamma}(N-1)_{1,1} - (1-p)^{N-3}) \\
&= \mathbf{\Gamma}(N-1)_{1,1} - (1-p)^{N-3} \\
&\leq \mathbf{\Gamma}(N-1)_{1,1} - (1-p)^{N-2}.
\end{aligned}$$

When $i = j + 1$, we have

$$\begin{aligned}
\mathbf{\Gamma}(N)_{i,j} &= p \cdot \mathbf{\Gamma}(N-1)_{i,j-1} + (1-p) \cdot \mathbf{\Gamma}(N-1)_{i,j+1} \\
&\leq p(\mathbf{\Gamma}(N-1)_{1,1} - (1-p)^{N-3}) + (1-p)\mathbf{\Gamma}(N-1)_{1,1} \\
&= \mathbf{\Gamma}(N-1)_{1,1} - p(1-p)^{N-3} \\
&\leq \mathbf{\Gamma}(N-1)_{1,1} - (1-p)^{N-2}.
\end{aligned}$$

When $i = j - 1$, we have

$$\begin{aligned}
\mathbf{\Gamma}(N)_{i,j} &= p \cdot \mathbf{\Gamma}(N-1)_{i,j-1} + (1-p) \cdot \mathbf{\Gamma}(N-1)_{i,j+1} \\
&\leq p\mathbf{\Gamma}(N-1)_{1,1} + (1-p)(\mathbf{\Gamma}(N-1)_{1,1} - (1-p)^{N-3}) \\
&= \mathbf{\Gamma}(N-1)_{1,1} - (1-p)^{N-2}.
\end{aligned}$$

Obviously, $\mathbf{\Gamma}(N-1)_{1,1} \leq \mathbf{\Gamma}(N)_{1,1}$ as $\mathbf{\Gamma}(N) = \mathbf{\Gamma}(N-1) + \mathbf{\Pi}^{N-2}$. Therefore, we prove that the induction hypothesis holds for $\mathbf{\Gamma}(N)$, which completes the induction. Thus, for all $N \geq 2$, the diagonal element is the largest entry in $\mathbf{\Gamma}(N)$. We analyze this result from the perspective of random walks. Given that $\mathbf{\Pi}^k$ is the $k$-step transition matrix in the random walk task, the entry $\mathbf{\Gamma}(N)_{i,j} = \sum_{l=0}^{N-2} \mathbf{\Pi}^l$ can be regarded as the expected number of visits to state $j$ within $N-1$ steps, starting from state $i$. Since the largest entry in $\mathbf{\Gamma}(N)$ is found at the diagonal, we can conclude that the expected number of visits back to state $i$ within $N-1$ steps, starting from state $i$, is the largest. From Lemma E.2 and E.3, we know that for any $i, j$,

$$\left|[\mathbf{\Pi}^R]_{i,j} + [\mathbf{\Pi}^{R+1}]_{i,j} - \frac{2}{K}\right| \leq \exp\left(-\frac{8p(1-p)R}{K^2}\right) + \exp\left(-\frac{8p(1-p)(R+1)}{K^2}\right).$$

Hence, we get that for any $i, j$,

$$\left|[\mathbf{\Pi}^R]_{1,1} + [\mathbf{\Pi}^{R+1}]_{1,1} - [\mathbf{\Pi}^R]_{i,j} - [\mathbf{\Pi}^{R+1}]_{i,j}\right| \leq 2\left[\exp\left(-\frac{8p(1-p)R}{K^2}\right) + \exp\left(-\frac{8p(1-p)(R+1)}{K^2}\right)\right].$$

Thus, for a constant integer $N_0 = \omega(p^{-1}(1-p)^{-1})$ and $N > N_0$, we can get for $i \neq j$,

$$\mathbf{\Gamma}(N)_{1,1} - \mathbf{\Gamma}(N)_{i,j} \geq (1-p)^{N_0-2} - c_{p0}\exp(-N_0),$$

where $c_{p0}$ are constants related to $p$. In addition, we can get $|\mathbf{\Gamma}(N)_{1,j_1} - \mathbf{\Gamma}(N)_{1,j_2}| \leq c_{p0}\exp(-N_0)$ for any $j_1 \neq 1, j_2 \neq 1$.

Next, we focus on $\mathbf{\Pi}(\mathbf{\Pi}^\top)^k$ with $k = 1, 2, ..., N-1$. Denote $\mathbf{\Gamma}_2(k) = \mathbf{\Pi}(\mathbf{\Pi}^\top)^k$. Similar to the analysis of induction above, we can use induction to prove that for $i \neq j$,

$$\mathbf{\Gamma}_2(2k-1)_{1,1} + \mathbf{\Gamma}_2(2k)_{1,1} \geq \mathbf{\Gamma}_2(2k-1)_{i,j} + \mathbf{\Gamma}_2(2k)_{i,j} - (1-p^2)^k.$$

The proof can be directly extended from the proof for $\mathbf{\Gamma}(N)$ above. Then, we use this result to prove

$$[\mathbf{\Gamma}_2(1)]_{1,1} + [\mathbf{\Gamma}_2(2)]_{1,1} \geq [\mathbf{\Gamma}_2(2k-1)]_{1,1} + [\mathbf{\Gamma}_2(2k)]_{1,1}$$

for any $k \geq 2$. We also use induction to prove this. It is easily checked that $[\mathbf{\Gamma}_2(1)]_{1,1} + [\mathbf{\Gamma}_2(2)]_{1,1} = p^2 + (1-p)^2 \geq 3p^3(1-p) + 3p(1-p)^3 = [\mathbf{\Gamma}_2(3)]_{1,1} + [\mathbf{\Gamma}_2(4)]_{1,1}$. Suppose that $[\mathbf{\Gamma}_2(2k-1)]_{1,1} + [\mathbf{\Gamma}_2(2k)]_{1,1} \geq [\mathbf{\Gamma}_2(2k+1)]_{1,1} + [\mathbf{\Gamma}_2(2k+2)]_{1,1}$. Then, by $\mathbf{\Gamma}_2(2k+1) + \mathbf{\Gamma}_2(2k+2) = (\mathbf{\Gamma}_2(2k-1) + \mathbf{\Gamma}_2(2k))(\mathbf{\Pi}^\top)^2$, we can get

$$
\begin{aligned}
\mathbf{\Gamma}_2(2k+1)_{1,1} + \mathbf{\Gamma}_2(2k+2)_{1,1} &= p^2[\mathbf{\Gamma}_2(2k-1) + \mathbf{\Gamma}_2(2k)]_{1,K-1} \\
&\quad + 2p(1-p)[\mathbf{\Gamma}_2(2k-1) + \mathbf{\Gamma}_2(2k)]_{1,1} \\
&\quad + (1-p)^2[\mathbf{\Gamma}_2(2k-1) + \mathbf{\Gamma}_2(2k)]_{1,3} \\
&\leq \mathbf{\Gamma}_2(2k-1)_{1,1} + \mathbf{\Gamma}_2(2k)_{1,1},
\end{aligned}
$$

where the inequality is by the result demonstrated before. Therefore, we complete the induction and get that for $k \geq 1$,

$$
\begin{aligned}
\mathbf{\Gamma}_2(1)_{1,1} + \mathbf{\Gamma}_2(2)_{1,1} - \mathbf{\Gamma}_2(2k+1)_{1,1} - \mathbf{\Gamma}_2(2k+2)_{1,1} &\geq \mathbf{\Gamma}_2(1)_{1,1} + \mathbf{\Gamma}_2(2)_{1,1} - \mathbf{\Gamma}_2(3)_{1,1} - \mathbf{\Gamma}_2(4)_{1,1} \\
&= (p^2 + (1-p)^2)(1 - 3p(1-p)).
\end{aligned}
$$

Since $\mathbf{\Gamma}_2(2)_{1,1} = 0$, we obtain for $k \geq 2$,

$$\mathbf{\Gamma}_2(1)_{1,1} - \mathbf{\Gamma}_2(k)_{1,1} \geq (p^2 + (1-p)^2)(1 - 3p(1-p)) := c_{p1}.$$

We provide an intuitive explanation from the perspective of random walks. $\mathbf{\Pi}(\mathbf{\Pi}^\top)^k$ represents first taking a step according to the transition $\mathbf{\Pi}$, followed by $k$ steps according to the transition $\mathbf{\Pi}^\top$. With increasing $k$, the distribution of the possible state after $k+1$ steps is closer to uniform across all states, regardless of the starting state chosen, which can also be shown by Lemma E.2 and E.3. Thus, $\mathbf{\Pi}(\mathbf{\Pi}^\top)^k e_1$ will converge to a vector corresponding uniform distribution, and $\mathbf{\Pi}\mathbf{\Pi}^\top e_1$ represents the most sparse case with the largest value concentrating on the first entry.

Combining all the results obtained above, we can conclude that

$$
\begin{aligned}
&\left[\sum_{i=1}^{N-1} \mathbf{\Pi}^i \cdot \mathbf{\Pi}^\top\right] - \left[\sum_{i=1}^{N-1} \mathbf{\Pi}^i \cdot (\mathbf{\Pi}^\top)^k\right] \\
&= e_1^\top \mathbf{\Gamma}(N) \cdot \mathbf{\Gamma}_2(1) e_1 - e_1^\top \mathbf{\Gamma}(N) \cdot \mathbf{\Gamma}_2(k) e_1 \\
&= \mathbf{\Gamma}(N)_{1,1}\mathbf{\Gamma}_2(1)_{1,1} + \sum_{i=2}^{K} \mathbf{\Gamma}(N)_{1,i}\mathbf{\Gamma}_2(1)_{i,1} - \mathbf{\Gamma}(N)_{1,1}\mathbf{\Gamma}_2(k)_{1,1} - \sum_{i=2}^{K} \mathbf{\Gamma}(N)_{1,i}\mathbf{\Gamma}_2(k)_{i,1} \\
&\geq \mathbf{\Gamma}(N)_{1,1}(\mathbf{\Gamma}_2(1)_{1,1} - \mathbf{\Gamma}_2(k)_{1,1}) + \min_{2 \leq i \leq K}\{\mathbf{\Gamma}(N)_{1,i}\}(1 - \mathbf{\Gamma}_2(1)_{1,1}) - \max_{2 \leq i \leq K}\{\mathbf{\Gamma}(N)_{1,i}\}(1 - \mathbf{\Gamma}_2(k)_{1,1}) \\
&= \mathbf{\Gamma}_2(1)_{1,1}(\mathbf{\Gamma}(N)_{1,1} - \min_{2 \leq i \leq K}\{\mathbf{\Gamma}(N)_{1,i}\}) - \mathbf{\Gamma}_2(k)_{1,1}(\mathbf{\Gamma}(N)_{1,1} - \max_{2 \leq i \leq K}\{\mathbf{\Gamma}(N)_{1,i}\}) \\
&\quad + \min_{2 \leq i \leq K}\{\mathbf{\Gamma}(N)_{1,i}\} - \max_{2 \leq i \leq K}\{\mathbf{\Gamma}(N)_{1,i}\} \\
&\geq (\mathbf{\Gamma}_2(1)_{1,1} - \mathbf{\Gamma}_2(k)_{1,1})(\mathbf{\Gamma}(N)_{1,1} - \min_{2 \leq i \leq K}\{\mathbf{\Gamma}(N)_{1,i}\}) + \min_{2 \leq i \leq K}\{\mathbf{\Gamma}(N)_{1,i}\} - \max_{2 \leq i \leq K}\{\mathbf{\Gamma}(N)_{1,i}\} \\
&\geq c_{p1}\left((1-p)^{N_0-2} - c_{p0}\exp(-N_0)\right) - c_{p0}\exp(-N_0) \\
&\geq C_p,
\end{aligned}
$$

where $C_p$ is a positive constant related to $p$. $\qquad\square$

The following lemma shows the basic property of the positional embedding.

**Lemma E.5.** Assume that

$$p_i = \left[\sin\left(\frac{i\pi}{M+1}\right), \sin\left(\frac{2i\pi}{M+1}\right), \ldots, \sin\left(\frac{Mi\pi}{M+1}\right)\right]^\top$$

for $i \in [M]$. It holds that

$$p_{i_1}^\top p_{i_2} = \begin{cases} \frac{M+1}{2} & \text{for } i_1 = i_2; \\ 0 & \text{for } i_1 \neq i_2. \end{cases}$$

**_Proof of Lemma E.5._** When $i_1 \neq i_2$ and $i_1 + i_2$ are even, we have

$$\boldsymbol{p}_{i_1}^\top \boldsymbol{p}_{i_2} = \sum_{j=1}^M \sin\left(\frac{ji_1\pi}{M+1}\right)\sin\left(\frac{ji_2\pi}{M+1}\right)$$

$$= \sum_{j=0}^M \sin\left(\frac{ji_1\pi}{M+1}\right)\sin\left(\frac{ji_2\pi}{M+1}\right)$$

$$= \frac{1}{4}\sum_{j=0}^M \left[\exp\left(\mathrm{i}\pi\frac{i_1-i_2}{M+1}j\right) + \exp\left(-\mathrm{i}\pi\frac{i_1-i_2}{M+1}j\right)\right.$$

$$\left. - \exp\left(\mathrm{i}\pi\frac{i_1+i_2}{M+1}j\right) - \exp\left(-\mathrm{i}\pi\frac{i_1+i_2}{M+1}j\right)\right]$$

$$= \frac{1}{4}\cdot\frac{\exp\left(\mathrm{i}\pi(i_1-i_2)\right)-1}{\exp\left(\mathrm{i}\pi\frac{i_1-i_2}{M+1}\right)-1} + \frac{1}{4}\cdot\frac{\exp\left(-\mathrm{i}\pi(i_1-i_2)\right)-1}{\exp\left(-\mathrm{i}\pi\frac{i_1-i_2}{M+1}\right)-1}$$

$$- \frac{1}{4}\cdot\frac{\exp\left(\mathrm{i}\pi(i_1+i_2)\right)-1}{\exp\left(\mathrm{i}\pi\frac{i_1+i_2}{M+1}\right)-1} - \frac{1}{4}\cdot\frac{\exp\left(-\mathrm{i}\pi(i_1+i_2)\right)-1}{\exp\left(-\mathrm{i}\pi\frac{i_1+i_2}{M+1}\right)-1}$$

$$= 0,$$

where the third equation is by $\sin(x) = \frac{\exp(\mathrm{i}x)-\exp(-\mathrm{i}x)}{2\mathrm{i}}$, and the last inequality is by $\exp(\mathrm{i}\pi k) = 1$ for even $k$. When $i_1 \neq i_2$ and $i_1 + i_2$ are odd, we have

$$\boldsymbol{p}_{i_1}^\top \boldsymbol{p}_{i_2} = \sum_{j=1}^M \sin\left(\frac{ji_1\pi}{M+1}\right)\sin\left(\frac{ji_2\pi}{M+1}\right)$$

$$= \sum_{j=0}^M \sin\left(\frac{ji_1\pi}{M+1}\right)\sin\left(\frac{ji_2\pi}{M+1}\right)$$

$$= \frac{1}{4}\sum_{j=0}^M \left[\exp\left(\mathrm{i}\pi\frac{i_1-i_2}{M+1}j\right) + \exp\left(-\mathrm{i}\pi\frac{i_1-i_2}{M+1}j\right)\right.$$

$$\left. - \exp\left(\mathrm{i}\pi\frac{i_1+i_2}{M+1}j\right) - \exp\left(-\mathrm{i}\pi\frac{i_1+i_2}{M+1}j\right)\right]$$

$$= \frac{1}{4}\cdot\frac{\exp\left(\mathrm{i}\pi(i_1-i_2)\right)-1}{\exp\left(\mathrm{i}\pi\frac{i_1-i_2}{M+1}\right)-1} + \frac{1}{4}\cdot\frac{\exp\left(-\mathrm{i}\pi(i_1-i_2)\right)-1}{\exp\left(-\mathrm{i}\pi\frac{i_1-i_2}{M+1}\right)-1}$$

$$- \frac{1}{4}\cdot\frac{\exp\left(\mathrm{i}\pi(i_1+i_2)\right)-1}{\exp\left(\mathrm{i}\pi\frac{i_1+i_2}{M+1}\right)-1} - \frac{1}{4}\cdot\frac{\exp\left(-\mathrm{i}\pi(i_1+i_2)\right)-1}{\exp\left(-\mathrm{i}\pi\frac{i_1+i_2}{M+1}\right)-1}$$

$$= -\frac{1}{2}\left(\frac{1}{\exp\left(\mathrm{i}\pi\frac{i_1-i_2}{M+1}\right)-1} + \frac{1}{\exp\left(-\mathrm{i}\pi\frac{i_1-i_2}{M+1}\right)-1}\right)$$

$$+ \frac{1}{2}\left(\frac{1}{\exp\left(\mathrm{i}\pi\frac{i_1+i_2}{M+1}\right)-1} + \frac{1}{\exp\left(-\mathrm{i}\pi\frac{i_1+i_2}{M+1}\right)-1}\right)$$

$$= 0,$$

where the third equation is by $\sin(x) = \frac{\exp(\mathrm{i}x)-\exp(-\mathrm{i}x)}{2\mathrm{i}}$, the fifth inequality is by $\exp(\mathrm{i}\pi k) = -1$ for odd $k$, and the last equation is by $\frac{1}{\exp(x)-1} + \frac{1}{\exp(-x)-1} = -1$. When $i_1 = i_2$, we have

$$\boldsymbol{p}_{i_1}^\top \boldsymbol{p}_{i_2} = \sum_{j=1}^M \sin\left(\frac{ji_1\pi}{M+1}\right)\sin\left(\frac{ji_2\pi}{M+1}\right)$$

$$= \sum_{j=0}^{M} \sin\left(\frac{ji_1\pi}{M+1}\right) \sin\left(\frac{ji_2\pi}{M+1}\right)$$

$$= \frac{1}{4} \sum_{j=0}^{M} \left[ \exp\left(i\pi \frac{i_1 - i_2}{M+1}j\right) + \exp\left(-i\pi \frac{i_1 - i_2}{M+1}j\right) \right.$$

$$\left. - \exp\left(i\pi \frac{i_1 + i_2}{M+1}j\right) - \exp\left(-i\pi \frac{i_1 + i_2}{M+1}j\right) \right]$$

$$= \frac{M+1}{2} - \frac{1}{4} \sum_{j=0}^{M} \left[ \exp\left(i\pi \frac{i_1 + i_2}{M+1}j\right) + \exp\left(-i\pi \frac{i_1 + i_2}{M+1}j\right) \right]$$

$$= \frac{M+1}{2} - \frac{1}{4} \cdot \frac{\exp\left(i\pi(i_1 + i_2)\right) - 1}{\exp\left(i\pi \frac{i_1+i_2}{M+1}\right) - 1} - \frac{1}{4} \cdot \frac{\exp\left(-i\pi(i_1 + i_2)\right) - 1}{\exp\left(-i\pi \frac{i_1+i_2}{M+1}\right) - 1}$$

$$= \frac{M+1}{2},$$

where the third equation is by $\sin(x) = \frac{\exp(ix) - \exp(-ix)}{2i}$, and the last inequality is by $\exp(i\pi k) = 1$ for even $k$. $\qquad\square$

## F  Additional Experiments

### F.1  Additional Experiments on Random/Deterministic Walks

In this subsection, we provide additional experiments on synthetic data with $(K, N) = (20, 101)$. We consider the transformer model introduced in Section 2 with the length of the position embedding M = 1000. To train the model, we utilize gradient descent starting with zero initialization, where the learning rate $\eta = 1$ and the constant $\epsilon$ in the log-loss is set as $\epsilon = 0.1$. And, we run the gradient descent algorithm for T = 50 training epochs. Figure 8 and Figure 9 illustrate the experiments for $p = 0.5$ and $p = 1$ respectively. These experimental results match Theorem 3.1 and Theorem 4.1, which also strongly supports our theoretical results.

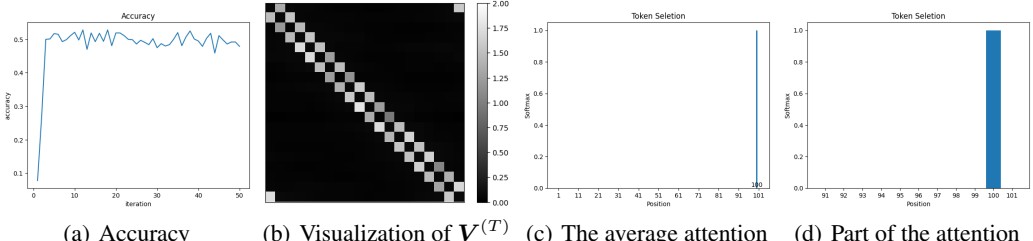

(a) Accuracy     (b) Visualization of $\boldsymbol{V}^{(T)}$     (c) The average attention     (d) Part of the attention

Figure 8: The results of the experiment for $p = 0.5$ with $(K, N) = (20, 101)$: (a) is the test accuracy; (b) is the visualization of $\boldsymbol{V}^{(T)}$; (c) and (d) present the average attention of the test data with x-axis representing the position of the token and y-axis representing the attention score.

### F.2  Additional Experiments on the Question Answering Tasks in Section 6

In this section, we conduct some additional experiments for Task 1 and Task 2 discussed in Section 6. We conduct some additional experiments extending the one-layer transformer model to a more complicated model by adding a fully connected layer with ReLU activation to the transformer model. The new model has the form

$$f_\theta(\boldsymbol{X}) = \boldsymbol{A} \cdot \text{ReLU}\left(\boldsymbol{V}\boldsymbol{X}\text{Softmax}\left(\widetilde{\boldsymbol{X}}^\top \boldsymbol{W}\widetilde{\boldsymbol{x}}_N\right)\right), \tag{F.1}$$

where $\boldsymbol{A} \in \mathbb{R}^{K \times m}$, $\boldsymbol{V} \in \mathbb{R}^{m \times K}$, $\boldsymbol{W} \in \mathbb{R}^{(K+M) \times (K+M)}$ are the trainable parameter matrices, and $m$ is the number of neurons in the fully connected layer. For Task 1 and Task 2, the length of

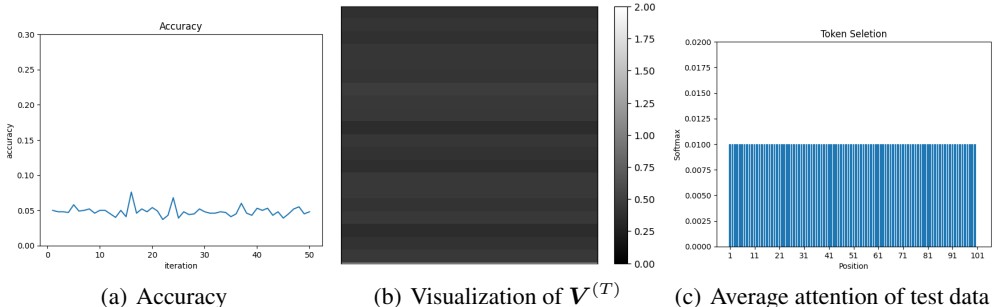

(a) Accuracy        (b) Visualization of $\boldsymbol{V}^{(T)}$        (c) Average attention of test data

Figure 9: The results of the experiment for $p = 1$ with $K = 20, N = 101$. (a) is the prediction accuracy with $x$-axis representing the iteration and $y$-axis representing the accuracy. (b) is the visualization of $\boldsymbol{V}$. (c) is the average attention of the test data with $x$-axis representing the position of the token and $y$-axis representing the attention score.

the vocabulary $K$ and the length of each input sequence $N$ are set as $(K, N) = (19, 17), (19, 19)$ respectively. In addition, we set the positional embedding $M = 1000$ and the number of neurons $m = 19$. To train the model, we consider the Gaussian random initialization $\boldsymbol{A}_{ij}^{(0)}, \boldsymbol{V}_{ij}^{(0)}, \boldsymbol{W}_{ij}^{(0)} \sim N(0, \sigma^2)$ with $\sigma = 0.01$, and use gradient descent with learning rate $\eta = 0.1$. The constant $\epsilon$ in the log-loss is set as $\epsilon = 0.1$. Both experiments are conducted on 1024 training data and 1024 test data. Here, most of the settings remain the same as in the previous experiments in Section 6.

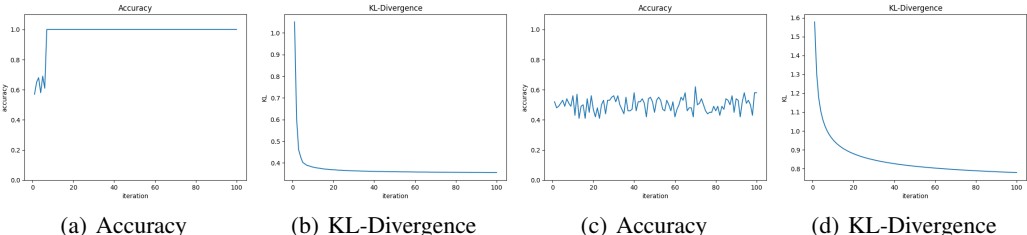

   (a) Accuracy        (b) KL-Divergence        (c) Accuracy        (d) KL-Divergence

Figure 10: The results of the experiment conducted using a more complicated transformer for Task 1 and Task 2: (a) and (b) correspond to the experiment for Task 1; (c) and (d) correspond to the experiment for Task 2.

Figure 10 shows the experiment results using the more complicated transformer in (F.1) to learn Task 1 and Task 2. In Figure 10(a) and Figure 10(c), we present the test accuracy achieved by the transformer model in learning Task 1 and Task 2 respectively. In Figure 10(b) and Figure 10(d), we first normalize the output of the trained transformer model to get a $K$-dimensional vector, representing the prediction distribution of $K$ words. Then, we report the KL-divergence between this prediction distribution and the true distribution of $y | \boldsymbol{x}_1, \boldsymbol{x}_2, ..., \boldsymbol{x}_{N-1}$. The experiment results show a clear difference between the performances of the transformer model in the two tasks. In Task 1, the trained transformer model can successfully approach the optimal accuracy (100%) within 100 iterations. However, in Task 2, the test accuracy always remains around 50%, which is the accuracy of a random guess.

Despite using a more complicated transformer model with an additional feedforward layer of non-linearities compared to the one considered in our theoretical analysis and previous experiments, the experimental results are still similar to those reported in Section 6. These results demonstrate that more complex transformer models may still struggle with the relatively 'simple' Task 2 but excel at the relatively 'difficult' Task 1. This indicates that our findings can be applied to cases involving additional nonlinearities, implying their applicability to more complex and general conditions.

### F.3 Visualizations of the value matrix and Softmax scores corresponding to Figure 6

In this section, we present visualizations of the value matrix and softmax scores corresponding to Figure 6.

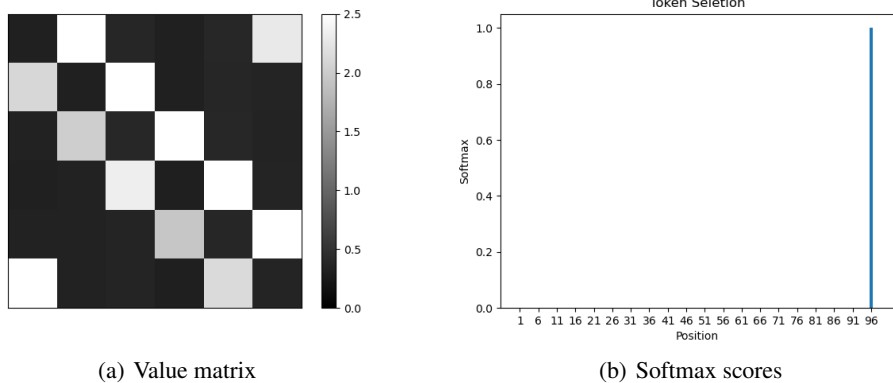

(a) Value matrix              (b) Softmax scores

Figure 11: Visualizations of the trained value matrix and the softmax scores corresponding to Figure 6(a) and Figure 6(b).

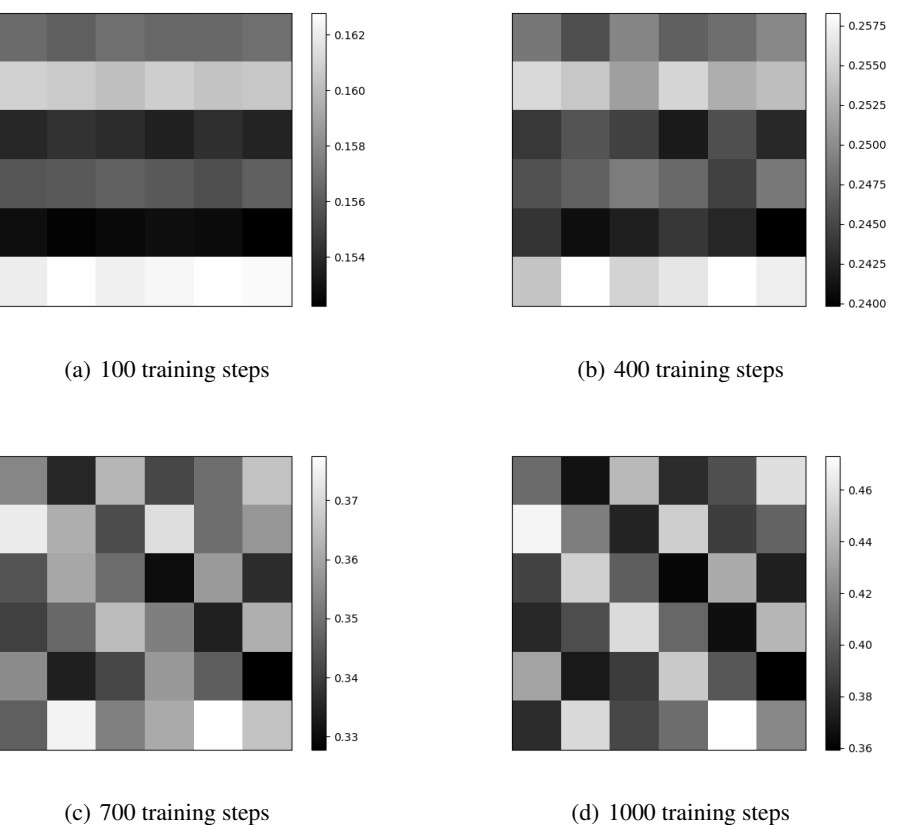

(a) 100 training steps              (b) 400 training steps

(c) 700 training steps              (d) 1000 training steps

Figure 12: Visualizations of the value matrix corresponding to Figure 6(c) and Figure 6(d) at 100, 400, 700, and 1000 training steps, respectively.

Figure 11 visualizes the trained value matrix and the softmax scores attached to all tokens, corresponding to Figure 6(a) and Figure 6(b). It shows that $V$ can recover the transition matrix $\mathbf{\Pi}^{\top}$, and the softmax score attached to the direct parent token is the largest and close to 1. These findings demonstrate that our positive results for learning random walks in Theorem 3.1 are fairly robust to random initialization.

Figure 12 and Figure 13 display the value matrix and the weighted average token $\boldsymbol{X} \cdot \mathcal{S}(\widetilde{\boldsymbol{X}}^{\top} \boldsymbol{W} \widetilde{\boldsymbol{x}}_N)$ corresponding to Figure 6(c) and Figure 6(d) at 100, 400, 700, and 1000 training steps, respectively. In Figure 13, the weighted average token embedding is calculated based on one sequence X as an example, while the variance of each dimension of the weighted average token is calculated across 1000 sequences. We can observe that when considering the case for $p = 1$ with random initialization,

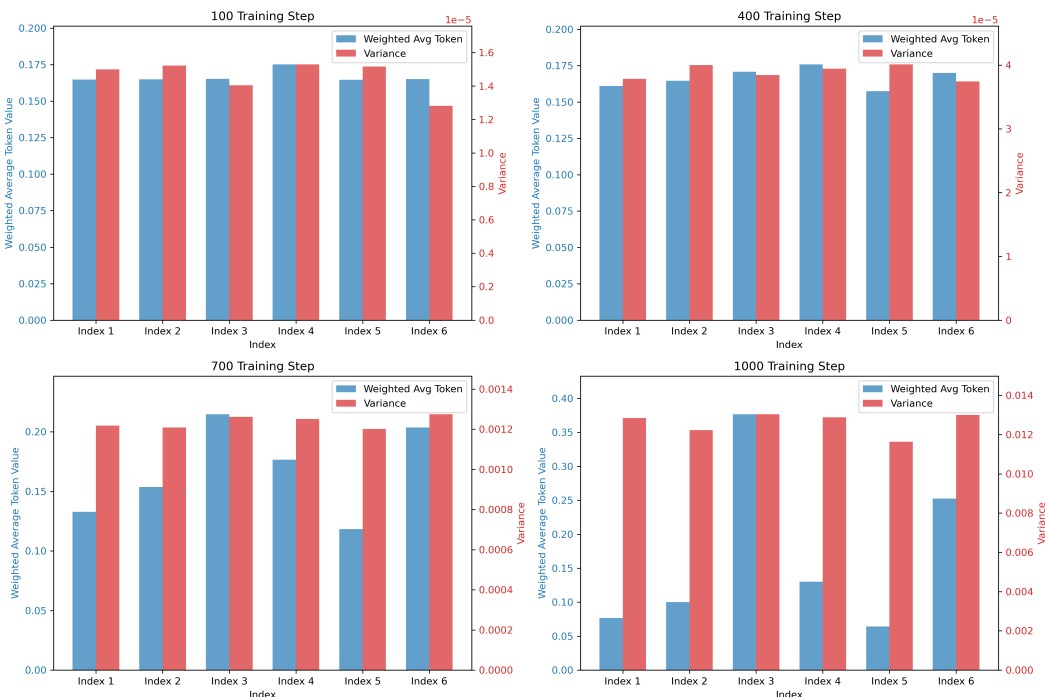

Figure 13: Visualizations of the weighted average token $\boldsymbol{X} \cdot \mathcal{S}(\widetilde{\boldsymbol{X}}^{\top} \boldsymbol{W} \widetilde{\boldsymbol{x}}_N)$ corresponding to Figure 6(c) and Figure 6(d) at 100, 400, 700, and 1000 training steps, respectively.

the value matrix and the weighted average token embedding remain approximately proportional to the all-one matrix and the all-one vector within 700 iterations, aligning with the results stated in Theorem 4.1. Additionally, the variances of all dimensions are close to 0, which shows that this result is general for different sequences. However, after 1000 iterations, $\boldsymbol{V}$ is no longer proportional to an all-one matrix, and the softmax score for one arbitrary token becomes much larger than the others. As discussed in Section 4, the failure case for $p = 0, 1$ is indeed due to an unbreakable symmetry caused by zero initialization. The random initialization may break this symmetry, enabling the transformer to successfully learn the optimal predictor.

