# OpenReview forum: "Towards Understanding Transformers in Learning Random Walks"
_NeurIPS.cc/2025/Conference — NeurIPS 2025 poster_

### Official Review · Reviewer_yFLc · 2025-07-01

**Clarity:** 3
**Significance:** 2
**Originality:** 4
**Rating:** 5
**Confidence:** 4

**Summary:**

The paper considers single-layer 0-initialised attention mechanisms applied to the task of learning random walks on segmented circles. The authors present theoretical results about the ability of such a model to learn random walks when left/right transition probabilities are in $p \\in (0, 1)$. They further present theoretical results arguing that such a model fails to learn when $p \\in \\{0, 1\\}$. Both claims are empirically supported. They present 2 further tasks with small vocabulary and fixed grammar and argue that zero-initialised single-layer attention mechanisms can only learn when the average of tokens is informative for the task output.

**Questions:**

- In the zero initialisation setting, do the results extend to 2-layer attention models? Perhaps with nonlinear activations?
- Could the authors comment further on the impact of their work, and how it can inform design of models / initalisation beyond the synthetic task considered?

**Ethical Concerns:**

["NO or VERY MINOR ethics concerns only"]

**Final Justification:**

The paper presents an interesting method and result for analysing the behavior of single-layer transformers when learning Markov dynamics under 0-init conditions. While open questions and future directions are apparent, it is not necessary for these to be answered in this one paper, which is valuable without these.

**Limitations:**

yes

**Paper Formatting Concerns:**

no concerns

**Quality:**

2

**Strengths And Weaknesses:**

Strengths:
- Convincing results for the zero-init setting for a single attention layer applied to learning random walks on circles
- Interpretable / intuitive explanation of the mechanism the model learns, particularly in figure 3.
- Thorough experiments in the setting considered, and intriguing extensions beyond this.


Weaknesses:
- It is not clear that the zero-initialisation setting is very useful in general. This initialisation is not common when using transformers in practice, for long-established empirical and theoretical reasons. Section 5 extends experiments to the random initialisation setting, and slow convergence is observed to the p=1 solution. It is not immediately apparent to me that this is a consqeuence of the author's prior analysis. I believe the argument here could be strengthened, by:
    1. Showing empirically that even in the presence of random initialisation, the attention model is trained approximately on the "uninformative" averaged token.
    2. Discussing the sudden jumps in accuracy. Are these e.g., because of random noise happening upon an output that is sufficiently far away from the uniform averaging? Though if that were the case, would the zero-init model also stumble on such a point?
- The results in Section 6 appear at best weakly related to the preceding content. The conclusion appears to be that the model is only able to learn when the mean of the tokens is informative for the label. Stated differently, a conjecture this paper seems to make is "Zero initialised models only learn when token averages have discriminative value".  This is an interesting result, which could be strengthened with further experimental support.


Overall, I think this is a good paper that makes a claim backed up by reasonable results. My primary concern is that the impact of the impact of the claim appears limited. I would look for further applications of this result to strengthen this paper.

---

> ### Author Rebuttal · Authors · 2025-07-31
>
> >**Q1**: It is not clear that the zero-initialization setting is very useful in general. Section 5 extends experiments to the random initialisation setting. It is not apparent to me that this is a consequence of the author's prior analysis. Suggestions: Show empirically + Discuss sudden jumps in accuracy.
>
> **A1**: The use of zero initialization in our theoretical analysis is primarily to simplify the dynamics of the model, which allows us to derive rigorous insights and obtain clear theoretical results. Our proof is based on the analyses of the softmax scores on each token during training. The setting with zero initialization ensures a fair start for all tokens, and we can prove that after training, the largest softmax score will be assigned to the direct parent token. When considering non-zero initialization, the initial scores assigned to all the tokens differ, leading to an unfair starting condition for training. This difference can largely influence and complicate the optimization analysis. If the direct parent token receives a relatively small initial score, the larger initial scores assigned to other tokens may cause the value matrix $V$ to learn a k-step transition matrix (other than the one-step transition), which can not achieve the optimal predictor. Additionally, in failure cases, the symmetry preserved under zero initialization is broken with non-zero initialization. This causes the largest softmax score to be assigned to an arbitrary token, which can enable the model to make an optimal prediction. Therefore, zero initialization serves as a useful idealization to facilitate theoretical understanding, while extending the results to non-zero initializations remains an interesting direction for future work. Please note that zero initialization or specific initialization has been considered in most theoretical studies [1,2,3,4].
>
> For the first suggestion, we provide the following results about the weighted average token ($X \cdot \text{softmax}(\tilde{X}W\tilde{x})$) corresponding to Figure 6(c) and 6(d).
>
> |Training steps|Weighted average token $X\text{softmax}(\tilde{X}W\tilde{x})$|Variance|
> |----|----|----|
> |100 steps| [0.1648, 0.1650, 0.1653, 0.1752, 0.1646, 0.1652] |[1.499e-5, 1.522e-5, 1.404e-5, 1.529e-5, 1.516e-5, 1.281e-5]|
> | 300 steps|[0.1633, 0.1649, 0.1676, 0.1753, 0.1617, 0.1672] |[1.912e-5, 2.006e-5, 1.856e-5, 1.974e-5, 2.007e-5, 1.741e-5] |
> | 500 steps|[0.1564, 0.1638, 0.1775, 0.1769, 0.1499, 0.1756] |[1.120e-4, 1.164e-4, 1.166e-4, 1.167e-4, 1.168e-4, 1.162e-4] |
> | 700 steps|[0.1328, 0.1538, 0.2147, 0.1766, 0.1183, 0.2037] |[1.218e-3, 1.209e-3, 1.261e-3, 1.251e-3, 1.201e-3, 1.274e-3] |
> | 900 steps|[0.0766, 0.1001, 0.3765, 0.1300, 0.0642, 0.2526] |[1.284e-2, 1.223e-2, 1.303e-2, 1.288e-2, 1.164e-2, 1.300e-2] |
>
> In the table above, the weighted average token embedding is calculated based on one sequence X as an example, while the variance of the weighted average token is calculated across 1000 sequences. We can observe that when considering the case for p=1 with random initialization, the weighted average token embedding remains approximately proportional to the all-one vector within 500 iterations. This indicates that the average token provides very limited useful information for training V, aligning with the results stated in Theorem 4.1. Additionally, the variances of all dimensions are close to 0, which shows that this result is general for different sequences. Although after 900 steps, some dimensions in the weighted average token embedding are relatively larger than others, it is still not enough to enable the transformer to achieve the optimal prediction, as illustrated in Figure 6(c).
>
> For the second suggestion, as mentioned in the first paragraph, the starting point of the softmax score attached to each token is different. This means that sufficient training steps are needed to update the weight matrix $W$ so that the largest softmax score can be assigned to the direct parent token. Before that, the attention may select another token, which can lead to a lower accuracy than the optimal one. In addition, since we consider 6 nodes in our experiments, the accuracy can only take values from a finite set. Thus, due to the long training steps needed and the discrete nature of accuracy, sudden jumps may occur when using random initialization. In contrast, when starting from zero initialization, the model begins fairly, and the softmax score for the direct parent token can quickly become the highest within the first few steps, which facilitates rapid achievement of the optimal accuracy.
>
> >**Q2**: The results in Section 6 appear at best weakly related to the preceding content.
>
> **A2**: Thank you for your comment. We would like to clarify that the purpose of Section 6 is to demonstrate the value of our findings. In particular, Section 6 demonstrates that our theoretical insights derived from studying random walks can be extended to other learning tasks and help us to successfully identify success and failure cases in training simple transformer models. Specifically, in Task 2, the data has the property that the token averages remain the same across all sequences; thus, we expect that Task 2 is a failure case for the simple transformer model, which is verified in Figure 7.
>
> >**Q3**: 2-layer attention models?
>
> **A3**: We agree that considering more complex models is important and aligns better with the practical setting. However, precise theoretical analysis of more complicated architectures can be extremely complicated. While our theoretical analysis can not be directly extended to 2-layer models, we believe that the theoretical insights derived from studying one-layer transformers may be extended to understand deeper models. In Theorem 3.1, we demonstrate that simple one-layer transformers can already successfully learn random walks. This may indicate the capability of multi-layer transformers in learning the same task as well. In Theorem 4.1, we demonstrate that one-layer transformers fail to learn deterministic walks due to the fact that the average of tokens does not provide useful information for prediction. We believe this insight can also be applied to learning the same task using multi-layer transformers. When the token average is not informative, multi-layer transformers may also struggle with zero initialization since it always takes an average over all tokens in each layer. As a result, the output of multiple layers may lack the necessary information for prediction.
>
> Please note that many recent theoretical studies on the training dynamics of transformers focus on simple one-layer transformer models [3,4,5,6]. In fact, to our knowledge, the setting considered in our paper is already more aligned with the practical transformer architecture compared to many of these recent theoretical studies. For example, [3,4] conduct the theoretical analysis on the transformer with a linear attention (instead of softmax attention). And, in [5,6], the value matrix $V$ is not involved in the training process; instead, the value matrix is held constant while other parameters are trained. In comparison, we construct a transformer architecture with nonlinear softmax attention and analyze the training of the matrices $V$ and $W$ simultaneously, which is a step toward a more practical study.
>
> >**Q4**: the impact of their work
>
> **A4**: To our knowledge, the most related work to ours is [7]. [7] characterizes the loss landscape of a one-layer transformer and identifies global minima and bad local minima based on Markovian data with vocabularies of size two, which can also be interpreted as a random walk over a space of only two states. In contrast, we consider random walk tasks on a circle that allows a large number of states, which can be seen as an extension to the two-state setting studied in [7]. This suggests that our theory represents a step further towards a unified theory for more general cases.
>
> In addition, our work provides a rigorous theoretical framework explaining how transformers learn Markovian data. We demonstrate that the trained transformer model is interpretable and can be decomposed into two parts: the token selector (attention focusing on the direct parent) and the one-step transition model (value matrix recovering $\Pi^T$). This reveals the token selection ability of softmax attention, enhancing our understanding of how softmax attention layers work in more complicated models. Moreover, this capability also addresses the reason why transformers can efficiently handle sequential data in NLP tasks, as softmax attention can capture the underlying structure of the data.
>
> While our work investigates the capability of transformers in learning simple or specific tasks, the findings contribute to a more comprehensive understanding of when and how transformers can succeed. This understanding can guide the design of transformer models for more complicated tasks. We demonstrate that the failure cases for $p=0\text{or}1$ are optimization failures, and are highly due to the zero initialization. This highlights the importance of avoiding unbreakable symmetry in optimization and emphasizes the need to select appropriate initialization strategies.
>
> [1] Wang et al. Transformers Provably Learn Sparse Token Selection While Fully-Connected Nets Cannot. ICML 2024.
>
> [2] Zhang et al. Transformer learns optimal variable selection in group-sparse classification. ICLR 2025.
>
> [3] Zhang et al. Trained transformers learn linear models in-context. JMLR 2024.
>
> [4] Usman et al. Adversarial Robustness of In-Context Learning in Transformers for Linear Regression. arXiv (2024).
>
> [5] Tarzanagh et al. Transformers as support vector machines. In NeurIPS 2023 Workshop on Mathematics of Modern Machine Learning.
>
> [6] Li et al. How do transformers learn topic structure: Towards a mechanistic understanding. ICML 2023.
>
> [7] Makkuva et al. Attention with markov: A curious case of single-layer transformers. ICLR 2024.

---

> > ### Comment · Reviewer_yFLc · 2025-08-05
> >
> > Thank you for your detailed response. The justifications provided are informative, and I believe this is interesting and work.

---

> > > ### Author Response · Authors · 2025-08-06
> > >
> > > Dear Reviewer yFLc,
> > >
> > > Thank you for your positive feedback! Your comments and suggestions greatly help us in revising the paper, and we truly appreciate it. If you have any further suggestions, please let us know.
> > >
> > > Best regards,
> > >
> > > Authors of Submission20613

---

### Official Review · Reviewer_bJ9o · 2025-07-01

**Clarity:** 3
**Significance:** 3
**Originality:** 3
**Rating:** 5
**Confidence:** 2

**Summary:**

The paper investigates how Transformers, specifically softmax attention without the feedforward network, perform on random walk problems. It addresses an interesting and relevant question, offering insightful analysis and well-motivated theoretical contributions. The paper presents clear motivations for the study and provides rigorous proofs regarding both the functionality of attention and its optimization dynamics in the context of learning random walks. The experimental results are supportive of the theoretical claims, and the paper thoughtfully explores various aspects such as initialization, positional embeddings, and architectural design. While I am not an expert in the theoretical analysis of random walks or Transformers in this specific context, I found the paper well-written and engaging in my opinion.

**Questions:**

As I am not very familiar with random walks, I hope these questions are not too basic. Some of them relate to Transformers and would be helpful to clarify for a broader audience.

1. **Figure 2 and Circularity of Random Walks**
   In Figure 2, is the reason that both the top-right and bottom-left corners have probabilities `p` and `1 - p` due to the **circularity** of the random walk? That is, does the random walk wrap around the boundaries in this setup?

2. **Role of Position Embedding**
   The position embedding used in this study appears to be specifically designed to help the Transformer solve the random walk task, effectively acting as an **inductive bias**. I’m curious: how would the **theoretical analysis** hold up **without** this guided position embedding? In practice, position embeddings like **RoPE** are often used, but may not be ideal for this type of structured task. It would be valuable to understand how the analysis extends to the case **without position embeddings**.

3. **Initialization of Attention Weights**
   As far as I understand, there is **no clear theoretical result** regarding random initialization of attention weights in this setting. While the attention computation can be expressed as $S(XᵀWX)$, in practice, for interpretability and implementation, the weight matrix $W$ is typically decomposed as $W_Q @ W_K^T$. Could this decomposition affect or enhance the theoretical results?

4. **Pretrained Initialization for Downstream Tasks**
   Following up on the weakness mentioned earlier, how would a Transformer perform on the tasks in Section 6 if it were first pretrained to **predict the next token** and then fine-tuned on tasks like identifying the preferable fruit? I would expect performance to improve significantly, as **random or zero initialization** is known to be insufficient for downstream tasks in Transformer-based models.

**Ethical Concerns:**

["NO or VERY MINOR ethics concerns only"]

**Final Justification:**

The authors' responses to my questions were convincing and satisfactorily addressed my concerns.

**Limitations:**

Please check the previous sections.

**Paper Formatting Concerns:**

Mostly the format of paper is good just  Figures can be more visible and references [30,31] and [11,12] are repeated.

Lastly, if the major weakness mentioned is addressed I would be happy to increase my score.


# References

[1] Amos, Ido, Jonathan Berant, and Ankit Gupta. "Never train from scratch: Fair comparison of long-sequence models requires data-driven priors."

**Quality:**

3

**Strengths And Weaknesses:**

## Strengths

- The paper clearly states the problem it is aiming to address and understand and the theoretical steps towards it are clear.
- The experimental setup for the problem in the random walks are supporting the claims and well motivated.
- Theoretical Proofs are solid and supporting the claims.

## Weaknesses

Overall, the paper is well-written and presents its ideas clearly. However, there are some presentation and formatting issues in certain sections that could be improved for better readability. In addition, I see one major weakness related to the initialization of the Transformer, and a conceptual gap in the tasks discussed in Section 6 (Beyond Random Walks), which seem to diverge from the core focus of the paper. These are, in my view, the most notable limitations. That said, I acknowledge that I may not be fully familiar with the specific literature in this area.

- **Generalization Beyond Random Walks May Not Be Representative**
Section 6 attempts to extend the findings to more applied settings, such as language-related tasks. However, there is a major conceptual gap, particularly regarding initialization. It is well-established that Transformers — when evaluated on downstream tasks like those mentioned (e.g., identifying a preferred fruit or detecting repetition, as part of copying tasks) — do not perform well with random or zero initialization. Instead, they require pretraining, typically via next-token prediction, followed by fine-tuning for classification or multiple-choice tasks, as highlighted in [1].

   This is consistent across all large language models: they are pretrained on large-scale language modeling tasks before being capable of handling such downstream problems. The use of zero initialization in the current study is, therefore, quite far from how Transformers are applied in practice.

   In this sense, random walks are more analogous to pretraining tasks (e.g., next-token prediction), whereas the tasks in Section 6 resemble downstream evaluation tasks, which inherently require pretraining to be solvable. Thus, the theoretical results derived under the zero initialization regime may not generalize well to the tasks presented in Section 6. As such, the section’s title, "Beyond Random Walks," may be misleading — it is unclear how far beyond and in what direction these extensions go, given the fundamental differences in task structure and training setup.

- **Readability of Figures** Figures in general specifically from 4 to 7 are very hard to read and very small in font it would be better to enlarge font size ore resize for better readability.

---

> ### Author Rebuttal · Authors · 2025-07-31
>
> >**Q1**: Generalization Beyond Random Walks May Not Be Representative + Pretrained Initialization for Downstream Tasks
>
> **A1**: Thank you for your detailed feedback. We agree that modern large language models are typically trained on large-scale datasets, and the simple question-answering tasks we consider in Section 6 can definitely be easily handled by well-pretrained language models.
>
> However, we believe that this cannot deny the importance of studying the capability of transformers in learning simpler or more specific tasks, as such findings can still help us gain a more comprehensive understanding of when and how transformers can succeed. Our results show that direct training of transformers fails in some of the seemingly simpler tasks while succeeds in other tasks, which we believe can contribute to a better understanding of the nature of transformer models and their training processes.
>
> In addition, we would like to clarify that the purpose of Section 6 is to demonstrate the value of our findings. In particular, Section 6 demonstrates that our theoretical insights derived from studying random walks can be extended to other learning tasks and help us to successfully identify success and failure cases in training simple transformer models. While it is an important question to theoretically study the performance of transformers under the setting of pretraining and fine-tuning, we feel that this is beyond the scope of our work.
>
> We will change the title of Section 6 to “Similar Observations in Two Simple Question Answering Tasks” to clarify the goal of this section and be more precise about the type of tasks we are going to consider.
>
> >**Q2**: Readability of Figures
>
> **A2**: Thank you for your suggestion. We will enlarge the font size for better readability in the revised version.
>
> >**Q3**: Figure 2 and Circularity of Random Walks
>
> **A3**: Yes, you are right. Since we study random walks on a circle with a finite number of nodes, the boundary states 1 and $K$ are actually neighbours. This means the walker can move from node 1 to node $K$ with probability $1-p$, and from node $K$ to node 1 with probability $p$. This circular structure is why the top-right and bottom-left corners in the transition matrix have value $p$ and $1 - p$.
>
> >**Q4**: Role of Position Embedding
>
> **A4**: Thanks for your comment. We would like to clarify that the specific definitions of the position embeddings are not important. The definition in the equation between lines 108 and 109 is mainly to mimic the classic positional embeddings based on sine and cosine functions. Our analysis only relies on the property that embeddings of different positions are orthogonal to each other. As long as orthogonality is maintained, changes to the embedding matrix will not impact the validity of our analysis. Please note that orthogonal positional embeddings are common in existing theoretical studies: [1] employs positional embeddings based on orthogonal sine basis functions; [2] and [3] adopt one-hot positional embeddings that are inherently orthogonal; [4] considers near-orthogonal positional embeddings.
>
> In our analysis, positional embeddings play a crucial role. In learning random walks, the key for a transformer model is to successfully identify the “direct parent token”. Positional embeddings can serve as identifiers of the direct parent. If positional embeddings are removed, the model may lack crucial information to solve the task optimally. In fact, we can show that if the input matrix does not consist of positional embeddings, any one-layer transformer cannot achieve a prediction accuracy better than a random guess. We will add this result and discussions about it in the revision.
>
> Regarding RoPE, since our current analysis relies on orthogonal positional embeddings, our analysis may not be directly applicable to RoPE. However, as we have mentioned, the setting with orthogonal positional embeddings is commonly considered in existing theoretical studies on transformers, and we believe that our current setting helps us to perform relatively clean theoretical analysis and therefore serves our purpose well. We will add comments that studying the case with RoPE is an interesting future work direction.
>
> >**Q5.1**: Initialization of Attention Weights: As far as I understand, there is no clear theoretical result regarding random initialization of attention weights in this setting.
>
> **A5.1**: The use of zero initialization in our theoretical analysis is primarily to simplify the dynamics of the model, which allows us to derive rigorous insights and obtain clear theoretical results. Our proof is based on the analyses of the softmax scores on each token during training. The setting with zero initialization ensures a fair start for all tokens, and we can prove that after training, the largest softmax score will be assigned to the direct parent token. When considering non-zero initialization, the initial scores assigned to all the tokens differ, leading to an unfair starting condition for training. This difference can largely influence and complicate the optimization analysis. If the direct parent token receives a relatively small initial score, the larger initial scores assigned to other tokens may cause the value matrix $V$ to learn a k-step transition matrix (other than the one-step transition), which can not achieve the optimal predictor. Additionally, in failure cases, the symmetry preserved under zero initialization is broken with non-zero initialization. This causes the largest softmax score to be assigned to an arbitrary token, which can enable the model to make an optimal prediction. Therefore, zero initialization serves as a useful idealization to facilitate theoretical understanding, while extending the results to non-zero initializations remains an interesting direction for future work. Please note that zero initialization or specific initialization has been considered in most theoretical studies [1,2,3,4,5,6].
>
> >**Q5.2**: While the attention computation can be expressed as $S(X^TWX)$, in practice, for interpretability and implementation, the weight matrix $W$ is typically decomposed as $W_QW_K^T$. Could this decomposition affect or enhance the theoretical results?
>
> **A5.2**: In our theoretical analysis, we consider a single weight matrix $W$ to denote $W_QW_K^T$ in self-attention, mainly for the simplicity of the theoretical analysis of the training dynamics. If we utilize the decomposition $W_QW_K^T$, it would require a more complex and careful analysis, as we need to deal with the simultaneous training of three parameter matrices $V$, $W_Q$, and $W_K$. In fact, we can no longer use zero initialization in this setting due to the interdependence of the training dynamics of $W_Q$ and $W_K$. The gradient of $W_Q$ involves $W_K$, and the gradient of $W_K$ also involves $W_Q$, which results in both gradients remaining zero with zero initialization. In this case, we have to consider random initialization. However, as discussed in **A5.1**, random initialization can largely complicate the analysis and may not yield clear theoretical results. There, we consider the reparameterization of a single weight matrix in our work. Please note that such reparameterization is widely considered in theoretical studies of transformer models [5,6,7,8].
>
> [1] Zhang et al. Transformer learns optimal variable selection in group-sparse classification. ICLR 2025.
>
> [2] Wang et al. Transformers Provably Learn Sparse Token Selection While Fully-Connected Nets Cannot. ICML 2024.
>
> [3] Eshaan Nichani et al. How Transformers Learn Causal Structure with Gradient Descent. ICML 2024.
>
> [4] Bietti et al. Birth of a transformer: A memory viewpoint. NeurIPS 2024.
>
> [5] Zhang et al. Trained transformers learn linear models in-context. JMLR 2024.
>
> [6] Usman et al. Adversarial Robustness of In-Context Learning in Transformers for Linear Regression. arXiv preprint arXiv:2411.05189 (2024).
>
> [7] Huang et al. In-context convergence of transformers. ICML 2024.
>
> [8] Ildiz et al. From Self-Attention to Markov Models: Unveiling the Dynamics of Generative Transformers. ICML 2024.
>
> [9] Jelassi et al. Vision transformers provably learn spatial structure. NeurIPS 2022.
>
> [10] Li et al. Mechanics of next token prediction with self-attention. AISTATS 2024.

---

> > ### Comment · Reviewer_bJ9o · 2025-08-02
> > **Thanks**
> >
> > I thank the authors for addressing my remaining questions. Overall, I find the paper valuable and addressing an interesting problem.
> >
> > To further improve the clarity and completeness of the work, I strongly encourage the authors to consider including discussions on the following points:
> >
> > * **Position embeddings:** Particularly, the issue of **orthogonality** in position embeddings and its implications.
> > * **Query/Key weight separation:** While a minor detail, it's worth noting—perhaps in a footnote—that separating query and key weights can affect initialization.
> > * **Figure readability:** It is important to improve **font size and clarity** across figures, as their current form is difficult to read.
> >
> > These suggestions aim to enhance the paper’s clarity and usability. That said, my main concerns have been resolved, and I will be happy to accept the paper even though I am not very familiar with this literature.

---

> > > ### Author Response · Authors · 2025-08-03
> > >
> > > Dear Reviewer bJ9o,
> > >
> > > Thank you for your support and feedback! We appreciate your valuable suggestions. In the revision, we will include additional discussions on positional embeddings and Query/Key weight separation, as well as adjust the font size to improve figure readability. If you have any further suggestions, please let us know.
> > >
> > > Best regards,
> > >
> > > Authors of Submission20613

---

### Official Review · Reviewer_Z1xn · 2025-07-02

**Clarity:** 4
**Significance:** 3
**Originality:** 3
**Rating:** 4
**Confidence:** 4

**Summary:**

This paper provides a theoretical and empirical analysis of a one-layer transformer's learning random walks on a circle. The authors prove that for non-deterministic walks (where the transition probability p is between 0 and 1), a transformer trained with gradient descent from zero initialization can achieve optimal prediction accuracy. Their analysis reveals an interpretable mechanism: the self-attention layer learns to function as a token selector, focusing almost exclusively on the direct parent state, while the value matrix learns to approximate the transition matrix. Conversely, the authors demonstrate that for deterministic walks (p=0 or p=1), the same training setup fails, resulting in performance no better than a random guess. They attribute this to an optimization failure caused by unbreakable symmetries under zero initialization. The findings are supported by experiments on the random walk task and extended to simple question-answering tasks to show the broader relevance of the identified failure mode.

**Questions:**

1. **On the Role of Random Initialization:** The analysis of the failure case for `p=0,1` is nice, but relies heavily on zero initialization. Your experiments with random initialization (Figure 6) are interesting; they show that learning remains much harder for `p=1` than for `p=0.5`. Could you elaborate on the theoretical challenges of analyzing dynamics under random initialization? Does it simply break the symmetry, allowing gradients to find useful directions, or is there a more complex dynamic?

2. **Generalizing Beyond Circle Graphs:** The random walk on a circle possesses a convenient Toeplitz/circulant structure. How do you expect results to change for random walks on general (undirected) graphs? Specifically, would the core mechanism of "attention selects parent, value matrix learns transition" still hold? I suspect the attention component might remain robust, but the clean convergence of `V(t)` to `Π^T` might be complicated by the absence of structural regularity.

3. **Absence of Softmax Scaling:** The standard transformer architecture includes a `1/sqrt(d_k)` scaling factor inside the softmax, which your model in Eq. (2.1) omits. Was this omission for analytical convenience? How would its inclusion affect the dynamics? This scaling factor could influence the softmax temperature, potentially impacting the speed and sharpness with which attention concentrates on the parent token, particularly during early training stages.

4. **Circle Structure Justification:** Could you elaborate on the choice to constrain the random walk to a circle? Specifically, could you clarify in the main text how the circle structure is fundamental to your analysis and why this regular structure is necessary for proof tractability? This would help readers better understand the scope and assumptions of your theoretical contribution early on.

5. **Residual Connection Impact:** A residual connection would pass token embeddings directly to the output, effectively 'giving away' the parent token x_{N-1} to the prediction head, making parent token selection trivial. How would your analysis change with the inclusion of a residual stream? Would the failure case for deterministic walks (p=0,1) still occur, or would direct access to the parent token resolve the 'uninformative average' issue even under zero initialization?

6. **Alternative Token Dependencies:** To address the triviality introduced by skip connections, it would be interesting to see generalization to Markov chains that still depend on a single token but at different positions rather than the previous one (e.g., x_{N-2}), as explored in [2,5].

7. **Higher-Order Markov Chains:** Have you considered extensions to higher-order Markov chains where the attention mechanism must learn to attend to multiple tokens simultaneously? Even if multiple token attention is possible using multiple heads (see [1] above), do you have insights into how a mechanism similar to your Value matrix implementation would function in such cases?

**Ethical Concerns:**

["NO or VERY MINOR ethics concerns only"]

**Final Justification:**

I recommend acceptance of this work. This paper presents a theoretically rigorous and insightful analysis of a one-layer transformer learning a specific, structured task. The authors' rebuttal was thorough and successfully clarified my questions. The paper is interesting and technically solid, but its core findings rely on a simplified and highly structured setting, which limits their generality. My final score reflects this balance between the paper's technical strength and its structured setup.

**Resolved Issues:**

The authors' rebuttal addressed all my specific questions.
*   **Impact of Residual Connections:** The authors corrected my misunderstanding, explaining why, in their specific prediction setup, a residual connection would not alter the model's output.
*   **Justification for Simplifications:** The authors provided justifications for their modeling choices, i.e., the use of a circle, zero initialization, and the omission of softmax scaling.
*   **Contribution:** They effectively positioned their work relative to prior related papers, highlighting that their contribution lies in providing rigorous guarantees for the training dynamics, not just the expressive power of the learned mechanism.

**Unresolved Issues:**

While my questions were answered, the core limitations that I raised in my initial review remain. These are not unresolved issues in the sense of a flawed rebuttal, but rather inherent constraints on the paper that the authors acknowledged:

*   **Limited Generality:** The analysis is fundamentally tied to the highly symmetric structure of a random walk on a circle and the use of a simplified one-layer transformer. How these results generalize to less structured tasks or more complex architectures remains a significant open question.
*   **Dependence on Zero Initialization:** The studied failure case for walks is an artifact of zero initialization. As the authors' own experiments show, this specific failure mode is less pronounced with random initialization, which is standard practice. The theory does not cover this more practical case.

**Limitations:**

yes

**Quality:**

3

**Strengths And Weaknesses:**

### Strengths

- **Presentation:** The paper is well written, I enjoyed reading it, and I learned something from it.

- **Theoretical Rigor:** The paper's primary strength is its rigorous theoretical analysis. The authors prove that a one-layer transformer trained with gradient descent learns the optimal predictor (for p∈(0,1)) and its convergence rate. They also show how in the deterministic cases (p=0,1) learning fails. This complete success/failure analysis is insightful and provides a clear picture of the model's capabilities and its training under the specified conditions.

- **Interpretability of the Learned Mechanism:** The paper provides a clear and interpretable explanation of how the transformer solves the task. While the finding that attention mechanisms learn to select parent tokens and value matrices encode transition probabilities is not entirely novel, seeing how this structured solution emerges naturally through gradient descent training also in this setting offers valuable insights.

- **Empirical Validation:** The experiments, while simple, are well-designed to directly validate the theoretical claims even beyond Markov Chains.


**Weaknesses:**

- **Simplified Model:** The analysis is restricted to a one-layer transformer without an MLP block, layer normalization, or residual connections. While this is a standard and necessary simplification for theoretical tractability, it represents a significant departure from practical, multi-layer transformer architectures. Although the authors are transparent about this limitation, it naturally constrains the generalization of their results.

- **Interpretability Contribution:** The novelty of the interpretable construction remains unclear. Previous works relating to n-gram learning and Markov chains have already established how attention is responsible for learning to attend to the correct token (implementing a token selector) while the transition matrix is stored in linear or MLP layers (see, for example, [1,2,3,4]).

- **Dependence on Zero Initialization:** The sharp failure case for deterministic walks critically depends on zero initialization. As the authors' own experiments in Figure 6 demonstrate, this failure can be partially mitigated through random initialization. While the zero-initialization analysis is theoretically clean and insightful, it raises questions about the practical relevance of this specific failure mode, given that random initialization is standard practice. Notably, the theory does not cover the random initialization case.

- **Task Specificity:** The random walk on a circle exhibits a highly regular structure, resulting in a Toeplitz transition matrix. This structure is heavily exploited throughout the proofs (e.g., Lemma C.2). It remains unclear how the analysis would extend to random walks on more general, less symmetric graphs, where the properties of the transition matrix might differ substantially.

- **Circle Constraint Motivation:**  The motivation for constraining the random walk to a circle is not immediately clear from the main text and could be made more explicit. This choice is crucial, as it induces a 'Toeplitz property' in the transition matrix.

[1] Anej Svete and Ryan Cotterell. 2024. Transformers Can Represent n-gram Language Models. In _Proceedings of the 2024 Conference of the North American Chapter of the Association for Computational Linguistics: Human Language Technologies (Volume 1: Long Papers)_, pages 6845–6881, Mexico City, Mexico. Association for Computational Linguistics.

[2] D'Angelo, F., Croce, F., & Flammarion, N. (2025). Selective induction heads: How transformers select causal structures in context. In _The Thirteenth International Conference on Learning Representations_.

[3] Nichani, Eshaan, Alex Damian, and Jason D. Lee. "How transformers learn causal structure with gradient descent." arXiv preprint arXiv:2402.14735 (2024).

[4] Edelman, Ezra, et al. "The evolution of statistical induction heads: In-context learning markov chains." Advances in neural information processing systems 37 (2024): 64273-64311.

[5] Makkuva, Ashok Vardhan, et al. "Attention with markov: A framework for principled analysis of transformers via markov chains." arXiv preprint arXiv:2402.04161 (2024).

---

> ### Author Rebuttal · Authors · 2025-07-31
>
> >**Q1**: Simplified Model
>
> **A1**: We agree that considering more complex models is important and aligns better with the practical setting. However, providing precise theoretical analysis for such more complicated architectures can be extremely challenging. Most existing theoretical studies on the training dynamics of transformers focus on simple one-layer transformer models [1,2,3,4,5]. In fact, to our knowledge, the setting considered in our paper is already more aligned with the practical transformer architecture compared to many of these recent theoretical studies. For example, [1,2] analyze transformers with linear attention instead of softmax attention. And, in [3,4,5], the value matrix is held constant during training, with only parameters inside the softmax matrix being trained. In comparison, our work constructs a transformer architecture with nonlinear softmax attention and analyzes the training of V and W simultaneously, which is a step toward a more practical study.
> >**Q2**: Interpretability Contribution
>
> **A2**: We acknowledge that previous works have explored how attention mechanisms facilitate token selection. We believe that one of our contributions is to provide further theoretical evidence on the successful cases where transformers can be trained to perform token selection. Below, we give detailed comparisons between our work and existing results in [6,7,8,9]. [5] and [6] show that transformers with specific constructions can represent any n-gram language model, but they primarily focus on the expressive power of transformers and do not delve into the training dynamics. [7,8] investigate transformers' in-context learning capabilities with Markovian data, both considering two-layer transformers. [7] sets a specific initialization designed to assign distinct roles to each layer, thereby facilitating the analysis of how different layers contribute during training. Meanwhile, [8] simplifies the model to a single linear attention layer making the analysis of training dynamics more tractable. In contrast, our work focuses on directly learning Markov chains through a single-layer transformer model. We provide theoretical guarantees that standard gradient descent, without any specialized initialization or assumptions, naturally converges to an interpretable solution capturing the inherent sequential structure (Theorem 3.1).
> >**Q3**: Zero Initialization
>
> **A3**: The use of zero initialization in our theoretical analysis is primarily to simplify the dynamics of the model, which allows us to derive rigorous insights and obtain clear theoretical results. Our proof is based on the analyses of the softmax scores on each token during training. The setting with zero initialization ensures a fair start for all tokens, and we can prove that after training, the largest softmax score will be assigned to the direct parent token. When considering non-zero initialization, the initial scores assigned to all the tokens differ, leading to an unfair starting condition for training. This difference can largely influence and complicate the optimization analysis. If the direct parent token receives a relatively small initial score, the larger initial scores assigned to other tokens may cause the value matrix V to learn a k-step transition matrix (other than the one-step transition), which can not achieve the optimal predictor. Additionally, in failure cases, the symmetry preserved under zero initialization is broken with non-zero initialization. This causes the largest softmax score to be assigned to an arbitrary token, which can enable the model to make an optimal prediction. Therefore, zero initialization serves as a useful idealization to facilitate theoretical understanding, while extending the results to non-zero initializations remains an interesting direction for future work. Please note that zero initialization or specific initialization has been considered in most theoretical studies [1,2,3,4].
> >**Q4**: Task Specificity
>
> **A4**: Our result demonstrates different performances of transformers when learning specific random or deterministic walks. Existing work [10] also points out the existence of success and failure cases in learning Markov chains with two states. These results together indicate that it may be very challenging to develop a general and unified theory for arbitrary transition matrices. We believe that such challenges in developing a more unified theory also, to some extent, demonstrate the difficulty and importance of obtaining our results. In fact, our setting is already more general than the previous work [10]. [10] characterizes the loss landscape of a one-layer transformer and identifies global minima and bad local minima based on Markovian data with vocabularies of size two, which can also be interpreted as a random walk over a space of only two states. In contrast, our framework accommodates a large number of states, suggesting that our theory represents a step further towards a unified theory for more general cases.
> >**Q5**: Circle Constraint Motivation
>
> **A5**: Thank you for your suggestion. Random walks in circles enjoy better symmetry properties that greatly facilitate the calculations of gradients, which are fundamental to the training analysis. For instance, due to the symmetry, the stationary distribution is uniform across all nodes, which significantly simplifies the calculation of some expected values like the term $E[x_ix_i^T]$ during the calculation of expected gradients. While it is possible to implement our proof techniques to study other types of random walks, the calculations may be significantly more complicated and tedious. This is the main reason that we choose to study random walks in circles. We will add some discussions in the revision.
> >**Q6**: Absence of Softmax Scaling
>
> **A6**: Thanks for your question. The reason we omit the softmax scaling factor is mainly for simplicity, as such omission has been considered in most of the theoretical studies [3,4,5,6,7,8]. If we add the scaling factor, we expect that the main theoretical conclusions will remain unchanged. Intuitively, even if the value matrix V is trained much faster than the matrix W, the softmax function will predict a mixture distribution before W is sufficiently trained to assign an almost-one softmax score to the direct parent token, and the predicted distribution cannot be close to optimal. Therefore, in such cases, the loss will not be sufficiently minimized, and the training will not stop (gradients of W will not be small). Therefore, we can still expect that the value matrix and the softmax matrix will eventually be trained as the transition matrix and the direct parent selector, respectively. However, when such a scaling factor is added, it is likely that the training will take more iterations to converge to the desired model. We will add a detailed discussion in the revision.
> >**Q7**: Residual Connection Impact
>
> **A7**: Thank you for your question. We believe this is a misunderstanding. We consider the setting where the prediction is made by reading the output of the last column of the transformer output, while the corresponding input part of this column is set to zero (see equation (2.1)). When using the residual connection, the transformer model can be expressed as $f(X)=X+VXS(...)$. Since the last column of X is zero, the last column of the output is determined solely by the second term. Thus, when reading the last column as the prediction, this model's output is the same as ours. Therefore, adding the residual connection will not make parent token selection trivial. In fact, our current result can already be equivalently applied to the setting with residual connection, because the effective model output, the gradients of parameters, and thus the training dynamics, are actually the same with or without residual connection. We will clarify this point in the revision.
> >**Q8**: Alternative Token Dependencies+Higher-Order Markov Chains
>
> **A8**: Thank you for your suggestions. We believe that our theory can be extended to Markov chains that depend on a different position. We take the case of $x_{N-2}$ as an example. At the first step, $V^{(1)}$ is proportional to $\sum_{i=1}^{[N/2]}(\Pi^T)^i$, similar to Lemma C.1. Due to the Toeplitz property of the transition matrix, we can find that $V^{(1)}$ is also a Toeplitz matrix whose largest entries appear on the locations of the largest entries of $P_i^T$. At the next step, $V^{(1)}$ will encourage $W$ to assign the highest score to the parent $x_{N-2}$. The remaining is the same as that stated in our work. Therefore, we can extend our results to this specific task.
>
> Regarding high-order models, it may require multiple layers or multiple heads, since it is likely that one-layer transformers may not have enough expressive power to handle such cases. The softmax attention may only select one token, and the value matrix can not store too much information about the transition probabilities.
>
> [1] Zhang et al. Trained transformers learn linear models in-context. JMLR 2024.
>
> [2] Usman et al. Adversarial Robustness of In-Context Learning in Transformers for Linear Regression. arXiv (2024).
>
> [3] Tarzanagh et al. Transformers as support vector machines. In NeurIPS 2023 Workshop.
>
> [4] Li et al. How do transformers learn topic structure: Towards a mechanistic understanding. ICML 2023.
>
> [5] Li et al. One-Layer Transformer Provably Learns One-Nearest Neighbor In Context. NeurIPS 2024.
>
> [6] Anej et al. Transformers Can Represent n-gram Language Models. NAACL 2024.
>
> [7] D'Angelo et al. Selective induction heads: How transformers select causal structures in context. ICLR 2025.
>
> [8] Nichani et al. How transformers learn causal structure with gradient descent. ICML 2024.
>
> [9] Edelman et al. The evolution of statistical induction heads: In-context learning markov chains. NeurIPS 2024.
>
> [10] Makkuva et al. Attention with Markov: A curious case of single-layer transformers. ICLR 2024.

---

> > ### Comment · Reviewer_Z1xn · 2025-08-09
> >
> > I would like to thank the authors for their thorough and thoughtful rebuttal. Your responses have successfully clarified my doubts, and the explanation regarding the residual connection was useful. I have no further questions.
> >
> > I find the paper's results to be an interesting and valid contribution, and I therefore recommend acceptance. However, as highlighted by other reviewers, the analysis relies on a simplified setup with specific structural assumptions, which are fundamental to the results. This limits the generality of the findings, which remains an important open question.
> >
> > I believe my current score reflects this balance, and I will be maintaining it.

---

> > > ### Author Response · Authors · 2025-08-09
> > >
> > > Dear Reviewer Z1xn,
> > >
> > > Thank you for your support and valuable suggestions. We agree that exploring more complicated settings remains an important direction for future work, and we will include a discussion of this in the revision. If you have any further suggestions, please let us know.
> > >
> > > Best regards,
> > >
> > > Authors of Submission20613

---

### Official Review · Reviewer_5AGb · 2025-07-03

**Clarity:** 4
**Significance:** 2
**Originality:** 2
**Rating:** 4
**Confidence:** 3

**Summary:**

This paper theoretically characterizes the learning dynamics of a one-layer transformer on the task of predicting a circular random walk.
The authors prove that for transition probabilities p in (0,1), gradient descent from zero initialization enables the model to achieve optimal prediction accuracy.
This is accomplished via an interpretable mechanism where the self-attention layer learns to act as a token selector, while the value matrix converges to the transition matrix.
In contrast, for deterministic walks (p=0,1), the paper demonstrates a failure case, caused by initial symmetries, where the model's performance is comparable to random guess.

**Questions:**

- The theory relies heavily on zero initialization. The experiments with random initialization are interesting but could be explored further. Could the authors comment on whether the interpretability of the V and S matrices is preserved (numerically) with larger, non-zero random initializations? What are the key obstacles to extending the theoretical analysis beyond the zero-init case?
- How would the failure mechanism for deterministic walks (p=0,1) be affected by the presence of multiple transformer layers? Would deeper models be able to break the symmetry, or would the 'uninformative' signal propagate and cause a similar failure?
- The appendix includes an experiment with an additional fully connected layer (F.2), which is a valuable addition. It would strengthen the paper to reference and briefly discuss this result in the main text. On a related note, what would be the expected effect of adding bias terms to the model's layers?
- Do similar conclusions about interpretability hold for the case of Markov chains with more complex transition matrices?

**Ethical Concerns:**

["NO or VERY MINOR ethics concerns only"]

**Final Justification:**

The paper presents a clear analysis of a simplified model and task. While I still believe that the architecture and setting are too idealized to have immediate practical relevance, and that the extendibility of the results to more realistic scenarios may be limited, the theoretical analysis is well written and instructive. The authors’ rebuttal addressed most of my concerns regarding the modeling assumptions. For these reasons, I consider this a borderline accept.

**Limitations:**

Yes

**Quality:**

3

**Strengths And Weaknesses:**

**Strengths:**
- The paper's main strength is its well-chosen toy problem that allows for a clear and mathematically tractable analysis.
- The work presents both a success case and a failure case within the same framework. The authors identify precisely which conditions lead to successful learning and which lead to optimization failure, and why.
- The findings offer a clear, interpretable story for how the parameters of a transformer can learn that specific simple task and the experiments are convincing.

**Weaknesses:**
- The primary weakness is that the paper's analysis relies on a heavily simplified model and task. The theoretical analysis is limited to a single attention layer and omits standard transformer components like MLPs, residual connections, and layer normalization.
- Because the proofs rely on explicit calculations made possible by these simplifications, it is  unclear whether the core insights or mathematical techniques would generalize to more realistic transformer architectures. This raises questions about the potential impact of these findings for understanding the transformers used in practice.

---

> ### Author Rebuttal · Authors · 2025-07-31
>
> >**Q1**: The primary weakness is that the paper's analysis relies on a heavily simplified model and task. The theoretical analysis is limited to a single attention layer and omits standard transformer components like MLPs, residual connections, and layer normalization.
>
> **A1**: Thank you for your suggestion. We agree that considering more complex models is important and aligns better with the practical setting. However, providing precise theoretical analysis for such more complicated architectures can be extremely challenging. Most existing theoretical studies on the training dynamics of transformers focus on simple one-layer transformer models [1,2,3,4,5]. In fact, to our knowledge, the setting considered in our paper is already more aligned with the practical transformer architecture compared to many of these recent theoretical studies. For example, [1] and [2] analyze transformers with linear attention instead of softmax attention. And, in [3], [4], and [5], the value matrix $V$ is held constant during training, with only parameters inside the softmax matrix being trained. In comparison, our work constructs a transformer architecture with nonlinear softmax attention and analyzes the training of the matrices $V$ and $W$ simultaneously, which is a step toward a more practical study.
>
> >**Q2**: Because the proofs rely on explicit calculations made possible by these simplifications, it is unclear whether the core insights or mathematical techniques would generalize to more realistic transformer architectures. This raises questions about the potential impact of these findings for understanding the transformers used in practice.
>
> **A2**: As mentioned in **A1**, our work has already analyzed a setting that is closer to practice scenarios than many existing theoretical studies. In addition, while the precise training procedure may differ, we believe that our study can still provide insights into more practical settings. Specifically, our study proves that a one-layer transformer can learn random walks by ‘selecting the parent token’ with softmax attention. The insight that softmax attention can serve as token selectors aligns with many existing theoretical and empirical studies [6,7,8].
>
> Additionally, we believe the fact that our theory can successfully motivate us to predict the performance of transformers in two question-answering tasks demonstrates the practical value of our theory. We believe that conducting rigorous theoretical analysis in a clean setting and providing insights applicable to other settings is precisely what a theory paper should aim to do, and our paper accomplishes this.
>
> >**Q3**: The theory relies heavily on zero initialization. The experiments with random initialization are interesting but could be explored further. Could the authors comment on whether the interpretability of the V and S matrices is preserved (numerically) with larger, non-zero random initializations? What are the key obstacles to extending the theoretical analysis beyond the zero-init case?
>
> **A3**: Thanks for your question. In our experiments with random initialization, the visualizations of V and S corresponding to Figure 6(a) and 6(b) show that the trained value matrix $V$ can recover the transition matrix $\Pi^{\top}$ and the softmax score attached to the direct parent token is the largest and close to 1, demonstrating that our positive results for learning random walks in Theorem 3.1 are fairly robust to random initialization.
>
> In the case for p=0,1, with random initialization, the visualizations of V and S corresponding to Figure 6(c) and 6(d) show that within 600 iterations, the value matrix is approximately proportional to an all-one matrix, and the softmax attention fails to select any token, aligning with the negative results stated in Theorem 4.1. However, after 1000 iterations, V is no longer proportional to an all-one matrix, and the softmax score for one arbitrary token becomes much larger than the others. As we have discussed in the paper, the failure case for p=0,1 is indeed due to an unbreakable symmetry caused by zero initialization. The random initialization may break this symmetry, enabling the transformer to successfully learn the optimal predictor.
>
> We will add these visualizations in the revision.
>
> Moreover, the use of zero initialization in our theoretical analysis is primarily to simplify the dynamics of the model, which allows us to derive rigorous insights and obtain clear theoretical results. Our proof is based on the analyses of the softmax scores on each token during training. The setting with zero initialization ensures a fair start for all tokens, and we can prove that after training, the largest softmax score will be assigned to the direct parent token. When considering non-zero initialization, the initial scores assigned to all the tokens differ, leading to an unfair starting condition for training. This difference can largely influence and complicate the optimization analysis. If the direct parent token receives a relatively small initial score, the larger initial scores assigned to other tokens may cause the value matrix $V$ to learn a k-step transition matrix (other than the one-step transition), which can not achieve the optimal predictor. Additionally, in failure cases, the symmetry preserved under zero initialization is broken with non-zero initialization. This causes the largest softmax score to be assigned to an arbitrary token, which can enable the model to make an optimal prediction. Therefore, zero initialization serves as a useful idealization to facilitate theoretical understanding, while extending the results to non-zero initializations remains an interesting direction for future work. Please note that zero initialization or specific initialization has been considered in most theoretical studies [1,2,3,4].
>
> >**Q4**: How would the failure mechanism for deterministic walks (p=0,1) be affected by the presence of multiple transformer layers? Would deeper models be able to break the symmetry, or would the 'uninformative' signal propagate and cause a similar failure?
>
> **A4**: We demonstrate that one-layer transformers fail to learn deterministic walks due to the fact that the average of tokens does not provide useful information for prediction. We believe this insight can also be applied to learning the same task using multi-layer transformers. When the token average is not informative, multi-layer transformers may struggle since it always takes an average over all tokens in each layer. As a result, the output of multiple layers may lack the necessary information for prediction.
>
> >**Q5**: The appendix includes an experiment with an additional fully connected layer (F.2), which is a valuable addition. It would strengthen the paper to reference and briefly discuss this result in the main text. On a related note, what would be the expected effect of adding bias terms to the model's layers?
>
> **A5**: Thank you for your suggestion. We will include discussions about this experiment in our revised version. Regarding the bias terms, we conduct the experiment with the following model: $f(X) = A\cdot\text{ReLU}((VX+b_V)\text{Softmax}(\tilde{X}^{\top} W \tilde{x}_{N})) + b_A$. The learning tasks and the settings are the same as the experiments in Section F.2. The results are similar to Figure 10 in the paper. In task 1, the transformer achieves the optimal accuracy (100%) after 20 iterations. In task 2, the test accuracy remains around 50%, which is just the accuracy of a random guess. This indicates that adding bias terms may not significantly influence the results.
>
> >**Q6**: Do similar conclusions about interpretability hold for the case of Markov chains with more complex transition matrices?
>
> **A6**: Our results demonstrate different performances of transformers when learning specific random or deterministic walks. Existing work [5] also points out the existence of success and failure cases in learning Markov chains with two states. These results together indicate that it may be very challenging to develop a general and unified theory for more complex transition matrices. We believe that such challenges in developing a more unified theory also, to some extent, demonstrate the difficulty and importance of obtaining our results.
>
> In fact, our setting is already more general than the previous work [5]. [5] characterizes the loss landscape of a one-layer transformer and identifies global minima and bad local minima based on Markovian data with vocabularies of size two, which can also be interpreted as a random walk over a space of only two states. In contrast, our framework accommodates a large number of states, suggesting that our theory represents a step further towards a unified theory for more general cases.
>
> [1] Zhang et al. Trained transformers learn linear models in-context. JMLR 2024.
>
> [2] Usman et al. Adversarial Robustness of In-Context Learning in Transformers for Linear Regression. arXiv preprint arXiv:2411.05189 (2024).
>
> [3] Li et al. How do transformers learn topic structure: Towards a mechanistic understanding. ICML 2023.
>
> [4] Li et al. One-Layer Transformer Provably Learns One-Nearest Neighbor In Context. NeurIPS 2024.
>
> [5] Tarzanagh et al. Transformers as support vector machines. In NeurIPS 2023 Workshop on Mathematics of Modern Machine Learning.
>
> [6] Zhang et al. Anchor function: a type of benchmark functions for studying language models. arXiv preprint arXiv:2401.08309 (2024).
>
> [7] Song et al. Out-of-distribution generalization via composition: a lens through induction heads in transformers. Proceedings of the National Academy of Sciences 122.6 (2025).
>
> [8] D'Angelo et al. Selective induction heads: How transformers select causal structures in context. ICLR 2025.

---

> > ### Comment · Reviewer_5AGb · 2025-08-06
> >
> > I would like to thank the authors for their responses to my questions. I still believe that the setting is quite simplified, both in terms of the model architecture (though the rebuttal addressed some of my doubts) and the task considered. However, the analysis is clear and instructive. For this reason, I am increasing my score to 4.

---

> > > ### Author Response · Authors · 2025-08-06
> > >
> > > Dear Reviewer 5AGb,
> > >
> > > Thank you for your valuable comments and for raising your score! We appreciate your insightful suggestions, which greatly help us improve our work. If you have any further suggestions, please let us know.
> > >
> > > Best regards,
> > >
> > > Authors of Submission20613

---

### Official Review · Reviewer_ki3b · 2025-07-05

**Clarity:** 3
**Significance:** 1
**Originality:** 2
**Rating:** 3
**Confidence:** 3

**Summary:**

This paper studies the dynamics of gradient descent on a 1 layer Transformer trained on a specific Markov chain arising from a biased random walk. It shows that gradient descent successfully solves this task in a constant number of iterations.

**Questions:**

- Would it be possible to extend the proof to the empirical loss rather than the population loss? It would be interesting to compare the sample complexity of the simple autoregressive model mentioned above with the depth 1 transformer considered in the paper. In particular, unlike the simple autoregressive model, the sample complexity of the depth 1 transformer would probably scale with the sequence length as it is forced to identify the parent tokens to emulate this autoregressive process.
- Why is it important for the analysis to specialize to this specific class of Markov chains (random walks on a circle) rather than allowing for arbitrary Markov chains (under some non-degeneracy assumption like $\pi(s|s') > 0$ for all $s,s'$)?

**Ethical Concerns:**

["NO or VERY MINOR ethics concerns only"]

**Final Justification:**

After the discussion, my main concern remains that this paper takes a very simple task (learning a Markov chain) which a standard Transformer can very easily learn, and then uses an unnatural input encoding and non-standard architecture which forces the model to learn a more complicated solution using the self-attention block.

**Limitations:**

yes

**Paper Formatting Concerns:**

I did not notice any major formatting issues in this paper.

**Quality:**

2

**Strengths And Weaknesses:**

Strengths:
- The analysis successfully handles Softmax attention rather than studying the simplified linear attention.
- The paper is generally well written and both the proof sketch and the discussion for the optimization failure when $p \in \{0,1\}$ are very clear.

Weaknesses:
- The setting is very simple and can be solved by much simpler models. For example, a simple autoregressive model (a "depth 0" Transformer) $f_\theta(\boldsymbol{X}) = \boldsymbol{V} \boldsymbol{x}_N$ would trivially solve this task with $\boldsymbol{V} \to \boldsymbol{\Pi}$ and the corresponding optimization problem would be convex. This could also be achieved by adding a residual connection to the transformer. The difficulty, both in the analysis and in the optimization dynamics, therefore arise from the fact that the architecture is written in such a way that the attention layer is forced to simulate an autoregressive process.
- A minor comment is that while the use of log-loss is now fairly common in theoretical analyses of Transformers, it should still be noted that this is non-standard as the final softmax after the value matrix is omitted. The paper therefore relies on the fact that $\mathcal{S}(\ldots) \ge 0$, $X \ge 0$ (because of the one-hot positional embeddings), and $V \ge 0$ (from the dynamics) to ensure that $f \ge 0$ so that the log-loss doesn't blow up.

---

> ### Author Rebuttal · Authors · 2025-07-31
>
> >**Q1**: The setting is very simple and can be solved by much simpler models. For example, a simple autoregressive model (a "depth 0" Transformer) $f(X)=Vx_N$ would trivially solve this task with $V \to \Pi$ and the corresponding optimization problem would be convex.
>
> **A1**: We agree that a simple autoregressive model $f(X)=Vx_N$ can also solve the classic random walk task. However, please note that our goal is not to identify the optimal model for solving random walk tasks but to understand and analyze the capability and interpretability of transformers when learning such classic statistical tasks from scratch.
>
> Importantly, autoregressive models may have an advantage in learning random walks because they assume Markovian property by definition. In comparison, the one-layer transformer model we consider follows the standard definition and is not designed with any prior knowledge of the Markovian property of the data. Therefore, investigating whether transformers can also learn random walks is more challenging. Moreover, by showing that self-attention can successfully be trained to learn the Markovian property by attending to the direct parent token, our results demonstrate how a simple transformer model can capture underlying data dependency structures and interpretably learn classic statistical tasks.
>
> We would like to mention that there exists a line of work studying the capabilities of transformers in learning Markov chains [1,2]. The tasks considered in these studies could also be addressed by some similar autoregressive models. [1] characterizes the loss landscape of a one-layer transformer and identifies global minima and bad local minima based on Markovian data. [2] shows that transformers with tokenization can efficiently learn k-th order Markov chains, and demonstrates that the failure cases observed in [1] can be overcome with appropriate tokenization. However, these studies only focus on the Markov chains with only two states. In contrast, our framework allows a large number of states, suggesting that our setting is already more general and complex.
>
> >**Q2**: A minor comment is that while the use of log-loss is now fairly common in theoretical analyses of Transformers, it should still be noted that this is non-standard as the final softmax after the value matrix is omitted. The paper therefore relies on the fact that $S(...) \geq 0, X \geq 0$ (because of the one-hot positional embeddings), and $V \geq 0$ (from the dynamics) to ensure that $f \geq 0$ so that the log-loss doesn't blow up.
>
> **A2**: Thank you for your comment. As you mentioned, log-loss is fairly common in theoretical analyses [1,3,4], and the setting with log-loss can already provide valuable insights into the underlying mechanisms and behaviors of transformers. In the revision, we will add discussions on the reasons that the loss does not blow up.
>
> >**Q3**: Would it be possible to extend the proof to the empirical loss rather than the population loss? It would be interesting to compare the sample complexity of the simple autoregressive model mentioned above with the depth 1 transformer considered in the paper.
>
> **A3**: We appreciate your suggestion regarding the comparison of sample complexity between transformers and the simple autoregressive model. Our current analysis focuses on the population loss, as this setting can help provide clean theoretical insights and guarantees about the transformer’s capabilities and interpretability. Training on the empirical loss may require significantly more complicated perturbation analyses to handle the gaps between population and empirical gradients. Please note that most of the existing theoretical studies on the training dynamics of transformers are based on the population loss [1,3,5,6].
>
> It is also likely that more carefully designed autoregressive models may have better sample complexity. However, as discussed in **A1**, the transformer model is designed with no prior knowledge of the Markov property, and therefore, such a comparison may not be fair. Our goal is not to optimally learn random works with the best-designed model, but to understand and demonstrate the capabilities of transformers.
>
> >**Q4**: Why is it important for the analysis to specialize to this specific class of Markov chains (random walks on a circle) rather than allowing for arbitrary Markov chains (under some non-degeneracy assumption like $\pi(s|s’)>0$ for all $s,s’$?
>
> **A4**: Our result demonstrates different performances of transformers when learning specific random or deterministic walks. Existing work [1] also points out the existence of success and failure cases in learning Markov chains with two states. These results together indicate that it may be very challenging to develop a general and unified theory for arbitrary transition matrices. We believe that such challenges in developing a more unified theory also, to some extent, demonstrate the difficulty and importance of obtaining our results.
>
> In fact, our setting is already more general than the previous work [1]. As mentioned in **A1**, [1] focuses on the first-order Markovian chains with vocabularies of size two, which can also be interpreted as a random walk over a space of only two states. In comparison, we consider random walk tasks on circles that allow a large number of states, indicating that our theory represents a step further towards a unified theory for more general cases.
>
> [1] Makkuva et al. Attention with Markov: A curious case of single-layer transformers. ICLR 2024.
>
> [2] Rajaraman et al. An analysis of tokenization: Transformers under markov data. NeurIPS 2024.
>
> [3] Ildiz et al. From self-attention to Markov models: Unveiling the dynamics of generative transformers. ICML 2024.
>
> [4] Li et al. Mechanics of next token prediction with self-attention. International Conference on Artificial Intelligence and Statistics. AISTATS, 2024.
>
> [5] Nichani et al. How transformers learn causal structure with gradient descent. ICML 2024.
>
> [6] Edelman, Ezra, et al. The evolution of statistical induction heads: In-context learning Markov chains. NeurIPS 2024.

---

> > ### Comment · Reviewer_ki3b · 2025-08-04
> >
> > Thank you for the clarifications. I still feel that the setting is rather artificial as you are heavily constraining the transformer by removing the residual connection (see below) and have thus decided to keep my score.
> >
> > > However, please note that our goal is not to identify the optimal model [...] the one-layer transformer model we consider follows the standard definition and is not designed with any prior knowledge of the Markovian property of the data.
> >
> > My point was that the transformer model is **non-standard** because there are no residual connections. A standard one-layer attention only transformer is of the form $x \to V(x + \text{attn}(x))_N = Vx_N + V\text{attn}(x)_N$. A standard transformer can easily learn Markov chains using the first term. This paper is therefore effectively asking whether the attention can emulate the identity map to mimic a residual connection, which seems a bit artificial.

---

> > > ### Author Response · Authors · 2025-08-05
> > >
> > > Dear Reviewer ki3b,
> > >
> > > Thank you for your further comments. We believe this is a misunderstanding. In our setting, the goal is to predict the location at the N-th step, and therefore $x_N$ is unknown and is set as zero (see the equation between lines 112 and 113), i.e. $x_N=0$. Because of this, in order to make good predictions in our setting, the model must learn to attend to the direct parent token.
> > >
> > > Importantly, our current setting can already cover the case with residual connection. When considering the residual connection, the transformer model can be expressed as
> > >
> > >  $f(X) = P[X + VX \cdot S(\tilde{X}^{\top} W \tilde{X})]_N = Px_N+PVX \cdot S(\tilde{X}^{\top} W \tilde{x}_N)$.
> > >
> > > Since $x_N=0$, the output is determined solely by the second term above. Thus, when including the residual connection, the output is the same as ours by combining $PV$ as a single parameter matrix. In this case, our current results can already be equivalently applied to the setting with residual connection, because the effective model output, the gradients of parameters, and thus the training dynamics, are actually the same with or without residual connection. We will clarify this point in the revision.
> > >
> > > Best regards,
> > >
> > > Authors of Submission20613

---

> ### Author Response · Authors · 2025-08-08
>
> Dear Reviewer ki3b,
>
> As the author-reviewer discussion period will end soon, we would like to kindly follow up to see if our responses have adequately addressed your concerns.
>
> Regarding your concern about the residual connection, we have clarified in our previous reply that this is a misunderstanding. Our setting already covers scenarios involving the residual connection, and even with the residual connection, the model still relies on softmax attention to select the direct parent token for optimal prediction. For more details, please refer to our previous response.
>
> Thank you for your time and effort in reviewing our work. If your concerns are addressed, we would sincerely appreciate your reconsideration of the evaluation.
>
> Best regards,
>
> Authors of Submission20613

---

> ### Author Response · Authors · 2025-08-09
>
> Dear Reviewer ki3b,
>
> We would like to clarify again that your concern about the residual connection is actually a misunderstanding. The transformer model with the residual connection cannot handle the task more easily.
>
> Additionally, we would like to reemphasize that our goal is to understand the capability and interpretability of transformers when learning classic statistical tasks. Although other models (especially those designed with prior knowledge of Markovian property) can also deal with the random walk tasks, this cannot deny the importance of studying the capability of transformers in learning such tasks. We believe our findings can inspire a more comprehensive understanding of when and how transformers can succeed.
>
> We sincerely appreciate your time and effort in reviewing our work. If your concerns are addressed, we would be grateful for your reconsideration of the evaluation.
>
> Best regards,
>
> Authors of Submission20613

---

### Decision · Program_Chairs · 2025-09-17

**Decision:**

Accept (poster)

**Comment:**

This paper studies the dynamics of gradient descent on a one-layer Transformer trained on a specific Markov chain arising from a biased random walk. The authors show that gradient descent successfully solves this task in a constant number of iterations. Specifically, the authors prove that for transition probabilities p∈(0,1), gradient descent from zero initialization achieves optimal prediction accuracy, through an interpretable mechanism in which the self-attention layer learns to act as a token selector, while the value matrix converges to the transition matrix. In contrast, for deterministic walks (p=0,1), the paper demonstrates a failure case caused by initial symmetries, where the model’s performance is no better than random guessing.

The reviewers found the paper’s results to be an interesting and valid contribution. They appreciated that the analysis is conducted on a real softmax Transformer and noted that the paper is well written, with clear explanations of the proof ideas and optimization bottlenecks. However, as highlighted in the reviews, the analysis relies on a simplified setting with strong structural assumptions that are fundamental to the results, which limits the generality of the findings.

Overall, I recommend acceptance and encourage the authors to incorporate the reviewers’ feedback in the final version.